# SocialMaze: A Benchmark for Evaluating Social Reasoning in Large Language Models

## Abstract

Large language models (LLMs) are increasingly applied to socially grounded tasks, such as online community moderation, media content analysis, and social reasoning games. Success in these contexts depends on a model's *social reasoning ability*—the capacity to interpret social contexts, infer others' mental states, and assess the truthfulness of presented information. However, there is currently no systematic evaluation framework that comprehensively assesses the social reasoning capabilities of LLMs. Existing efforts often oversimplify real-world scenarios and consist of tasks that are too basic to challenge advanced models. To address this gap, we introduce ***SocialMaze***, a new benchmark specifically designed to evaluate social reasoning. SocialMaze systematically incorporates three core challenges: *deep reasoning, dynamic interaction, and information uncertainty*. It provides six diverse tasks across three key settings—social reasoning games, daily-life interactions, and digital community platforms. Both automated and human validation are used to ensure data quality. Our evaluation reveals several key insights: models vary substantially in their ability to handle dynamic interactions and integrate temporally evolving information; models with strong chain-of-thought reasoning perform better on tasks requiring deeper inference beyond surface-level cues; and model reasoning degrades significantly under uncertainty. Furthermore, we show that targeted fine-tuning on curated reasoning examples can greatly improve model performance in complex social scenarios.

## 1 Introduction

LLMs demonstrate significant capabilities across various domains, such as scientific discovery [1, 2] and medical applications [3, 4]. Most recently, they have been increasingly applied to socially grounded tasks, such as online community moderation [5, 6, 7, 8, 9], media content analysis [10, 11, 12], and social reasoning games [13, 14]. The success of LLMs in these applications often hinges on their ***social reasoning abilities***—the capacity to understand the social context, infer others' mental states, and make appropriate judgments based on this understanding.

While existing benchmarks effectively evaluate the general capabilities of LLMs [15, 16, 17, 18, 19], benchmarks specifically designed to assess social reasoning abilities face significant limitations: 1) reliance on static scenarios lacking dynamic interaction [20, 21, 22, 23] , 2) presentation of overly sanitized information devoid of the noise, bias, or deception common in real social environments [17, 24, 25, 26] , and 3) tasks too simple to capture the deeper cognitive aspects of social inference[27, 28, 29]. A few examples, SocialIQA [20] primarily assess commonsense reasoning within simplified, predefined social contexts, testing basic understanding rather than complex, interactive inference. Similarly, Theory-of-Mind (ToM) benchmarks [30] often evaluate mental state inference based on static narratives and typically lack deceptive elements or informational uncertainty. Beyond

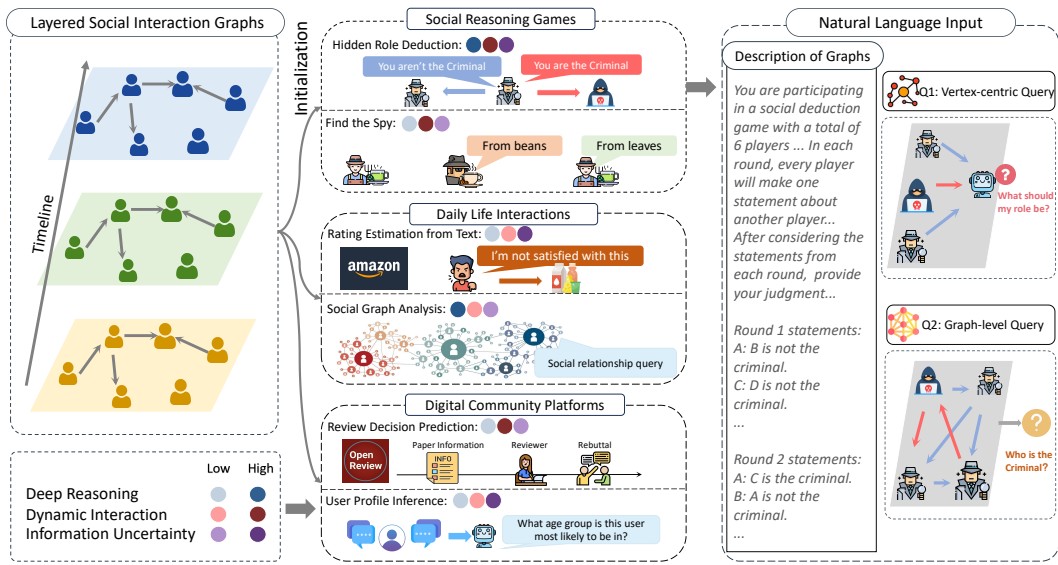

Figure 1: Overview of the SocialMaze Benchmark. All tasks are built upon (a) Layered Social Interaction Graphs, a time-aware modeling framework for social networks. Based on this template, we instantiate (b) 6 task types, covering social reasoning games, daily life interactions, and digital community platforms. (c) illustrates one specific example of Hidden Role Deducution, including description of graphs along with both vertex-centric and graph-level queries.

benchmarks, some recent work has applied LLMs to strategic games such as Diplomacy [31] and deduction games like Avalon and Werewolf [14, 32, 33, 34]. While these approaches attempt to place LLMs in dynamic environments, their evaluation typically emphasizes task outcomes—whether the model completes the task or outperforms competitors—rather than assessing whether the model genuinely engages in correct and coherent social reasoning. Success in such tasks does not necessarily indicate that the model understands the underlying social logic or reasoning process.

To address these limitations and enable a more holistic evaluation, we argue that the assessment of LLM social reasoning should explicitly incorporate three core aspects, which are key features of social reasoning tasks:

***Deep Reasoning***: Effective reasoning in social environments often requires going beyond surface-level information and engaging in complex cognitive processes. These include inferring others' latent mental states (such as intentions, beliefs, and motivations) [35, 36, 37, 38], analyzing complex causal relationships between actions and outcomes, exploring counterfactual possibilities, and engaging in strategic thinking or mental simulation to anticipate future scenarios and plan accordingly [30, 39, 40, 41]. ***Dynamic Interaction***: Real-world social contexts are characterized by iterative, interdependent exchanges. This demands that models track the evolving context across multiple turns and dynamically adapt their reasoning and actions based on prior interactions and anticipated responses [42, 43, 44, 45, 46]. Failure to do so leads to static or contextually inappropriate behavior. ***Information Uncertainty***: Social information landscapes are inherently noisy, with credibility varying greatly. They often contain misinformation, subjective biases, and intentional deception [47, 48, 49, 50]. This necessitates that models critically evaluate source reliability, filter misleading signals, and make robust inferences under conditions of incomplete or conflicting information.

Based on these three principles, we present ***SocialMaze***, a novel benchmark designed to reflect the challenges posed by all three dimensions, as shown in Figure 1. It consists of six diverse tasks spanning three categories: Social Reasoning Games, Daily Life Interactions, and Digital Community Platforms. Each task intentionally varies the demands along the three challenge dimensions. Our experiments with SocialMaze reveal several key insights into the capabilities of LLMs: Models with stronger chain-of-thought reasoning perform better on tasks that require deeper inference. We also observe that dynamic interaction affects model performance in varied ways across tasks, and information uncertainty significantly hinders reasoning. Moreover, reasoning agents and workflows [51, 52, 53] offer limited gains in these social reasoning challenges, while targeted fine-tuning on curated reasoning examples leads to substantial performance improvements.

Our contributions are threefold: 1) We identify *deep reasoning, interaction dynamics, and information uncertainty* as three core dimensions for evaluating social reasoning in LLMs, capturing critical challenges found in real-world social cognition. 2) Based on these dimensions, we construct SocialMaze, a benchmark dataset comprising six tasks across three real-world-inspired scenarios (social games, daily life interactions, and digital platforms), covering a wide range of reasoning types and difficulty levels. 3) Our experiments provide rich insights into LLM social reasoning: we identify key limitations in current models and highlight promising directions for future research, showing that techniques such as targeted fine-tuning can substantially improve performance.

## 2 Problem Formulation

We present a graph-based formalization of SocialMaze, where social entities and their evolving interactions are modeled as layered graphs.

**Modeling Social Entities and Interactions as Graph Structures.** We use graph structures to formally represent the participants and interactions within a social scenario. Let $\mathcal{S} = \{s_1, s_2, ..., s_n\}$ be the set of social members involved in a given setting, where each $s_i$ represents a distinct individual, such as a game player, a forum user, or a reviewer in a peer-review process. These social members form the vertex set $\mathcal{V}$ of a graph $G = (\mathcal{V}, \mathcal{E})$, with $\mathcal{V}$ serving as an abstract representation of the social members $\mathcal{S}$. Social interactions between members are represented as edges in the graph. Since interactions are time-dependent, we define a separate edge set $\mathcal{E}_t$ for each interaction round $t$. This leads to a sequence of time-specific graphs $G_t = (\mathcal{V}, \mathcal{E}_t)$, where an edge $(u, v) \in \mathcal{E}_t$ indicates that members $u$ and $v$ interacted during round $t$. The nature of the edges (directed or undirected) reflects the type of interaction. For example, directed edges may represent one-way actions (e.g., sending a message), while undirected edges may represent mutual interactions (e.g., a conversation or vote).

**Temporal Dynamics as Layered Graphs.** Social interactions are inherently dynamic and typically unfold over multiple rounds. To capture this temporal dimension, we represent the entire interaction process as a layered graph $\mathcal{G} = (G_1, G_2, ..., G_T)$, where $T$ denotes the total number of interaction rounds. Each layer $G_t = (\mathcal{V}, \mathcal{E}_t)$ captures the state of social members and their relationships during round $t$. Importantly, all layers share the same vertex set $\mathcal{V}$, reflecting a consistent group of participants throughout the interaction. However, the edge sets $\mathcal{E}_t$ vary across layers to reflect the evolving nature of relationships over time. In SocialMaze, LLMs receive natural language descriptions that encapsulate the information from these layered graphs, rather than raw graph structures. This design choice is intentional, aiming to mimic how humans comprehend social scenarios through language-based narratives.

**Query Categorization.** Based on the layered graph representation, we classify the queries posed within SocialMaze tasks into three distinct types, each targeting a different level of understanding of the graph structure: ***Vertex-centric Query** ($\mathcal{Q}_v(v_i)$):* This type of query probes the model's understanding of individual social members. Given a specific vertex $v_i \in \mathcal{V}$ (representing social member $s_i$), the task is to infer an attribute associated with $v_i$. ***Edge-centric Query** ($\mathcal{Q}_e(v_i, v_j)$):* Edge-centric queries assess the model's comprehension of the relationships between social members. Given two vertices $v_i, v_j \in \mathcal{V}$, the task is to determine the nature of their relationship, as represented by the edges connecting them. ***Graph-level Query** ($\mathcal{Q}_G(\mathcal{G})$):* Graph-level queries require the model to synthesize information from the entire layered graph $\mathcal{G}$ to derive a holistic understanding of the social scenario. These queries demand a comprehensive assessment of the overall interaction dynamics.

## 3 SocialMaze

Building on the layered social interaction graph framework, we introduce SocialMaze, a benchmark designed to operationalize the core challenges of social reasoning—*Dynamic Interaction*, *Information Uncertainty*, and *Deep Reasoning*. The benchmark covers three representative social contexts: Social Reasoning Games, Daily Life Interactions, and Digital Community Platforms. These settings comprise six major task categories, each carefully designed to vary along the key dimensions, enabling systematic evaluation of LLMs under different social conditions. Table 1 summarizes the tasks by required reasoning depth, degree of interaction dynamics, and level of information uncertainty. A detailed comparison between SocialMaze and prior benchmarks is provided in Appendix A.

Table 1: Overview of SocialMaze task categories and key characteristics. Tasks vary along three key dimensions: the level of *Deep Reasoning*, the degree of *Dynamic Interaction*, and the extent of *Information Uncertainty*, each categorized as High or Low.

| Scenario | Task Category | Deep Reasoning | Dynamic Interaction | Information Uncertainty | Number of Instances |
|---|---|---|---|---|---|
| Social Reasoning Games | Hidden Role Deduction | High | High | High | 20,000 |
| | Find the Spy | Low | High | Low | 6,000 |
| Daily Life Interactions | Rating Estimation from Text | Low | Low | High | 6,000 |
| | Social Graph Analysis | High | Low | Low | 20,000 |
| Digital Community Platforms | Review Decision Prediction | Low | High | Low | 12,000 |
| | User Profile Inference | Low | Low | High | 6,000 |

## 3.1 Task 1: Hidden Role Deduction

This task simplifies the core mechanics of Blood on the Clocktower [54] into a reasoning-only format. Unlike traditional interaction-based gameplay, all player statements are rule-generated. The model acts as a reasoner, analyzing all available information to logically infer each player's role.

**Task Rules.** The game features four roles: Investigators, Criminal, Rumormongers, and Lunatics. Investigators always tell the truth. The Criminal can choose to lie or tell the truth. Rumormongers believe they are Investigators, but their statements are randomly true or false. Lunatics believe they are the Criminal. The role each player sees may not reflect their true identity—Rumormongers are shown the role of Investigator, and Lunatics are shown the role of Criminal. The game consists of $n$ players, and the model participates by taking the perspective of Player 1 ($s_1$):—meaning it only observes what that player would see and say during the $T$ rounds. In each round, every player selects another player and makes a public statement, such as "Player $v$ says Player $u$ is (not) the criminal." After observing all interactions, the model is tasked with answering two key questions: identifying the true Criminal ($\mathcal{Q}_G$), and inferring its own actual role in the game ($\mathcal{Q}_v(v_i)$). The introduction of Rumormongers and Lunatics significantly increases *information uncertainty* and makes the reasoning process more challenging. The dataset includes four types of tasks: Original task, Rumormonger task, Lunatic task, and Full task. Details are provided in Appendix B.

**Design Rationale.** This task challenges large language models along three critical dimensions of social reasoning. First, the game unfolds over multiple rounds, requiring the model to track the temporal evolution of information, interpret changing relationships among players, and maintain consistent judgments across rounds—posing a challenge of *Dynamic Interaction*. Second, due to the presence of roles such as the Criminal, Rumormonger, and Lunatic—who may lie or provide misleading information—the environment is *highly uncertain*. The model must determine which statements are trustworthy and filter out deceptive cues, thereby grappling with information uncertainty. Most importantly, the model must reason not only about others' roles but also about its own true identity, which may differ from the one initially assigned. Addressing this requires strong *Deep Reasoning* capabilities, including resolving conflicts, managing uncertainty that extends to self-perception, and dynamically updating internal beliefs to approach the ground truth.

**Data Generation and Quality Assurance.** All player statements are automatically generated based on a set of predefined rules. Investigators begin by selecting a target they find suspicious, using a strategy function informed by all interactions up to the current round. They are always truthful in their statements. Rumormongers follow the same target selection logic as Investigators, but the truthfulness of their statements is random, making their input unreliable. In contrast, Criminals and Lunatics adopt a different strategy for choosing targets and deliberately introduce uncertainty by making deceptive statements with a certain probability, aiming to mislead others and conceal their true roles. To ensure each scenario is logically solvable, we design a search algorithm that verifies whether a unique solution exists to identify both the true Criminal and the LLM player's actual role. Additionally, the full reasoning chain leading to the solution is preserved and distilled into clear natural language, providing high-quality, curated examples of social reasoning that can be leveraged for targeted fine-tuning of language models. See Appendix B for details.

## 3.2 Task 2: Find the Spy

This task adapts the classic word-based social deduction game *Who Is The Spy* [14] to evaluate the LLM's ability to identify subtle deviations in communication within a group context characterized by high interaction but relatively low information uncertainty.

**Task Rules.** The game involves $n$ players. Among them, $n - 1$ players (Civilians) receive the same secret word, while one player (the Spy) receives a different but related word. Over $T$ rounds, each player provides a description of their word. The LLM is evaluated from the perspective of Player 1 ($s_1$): it knows the word assigned to $s_1$, but does not know whether $s_1$ is a Civilian or the Spy. It does not generate any player descriptions. Instead, after observing all $T$ rounds of player-generated descriptions, the LLM must infer which player received the different word. This constitutes a graph-level query ($\mathcal{Q}_G$). Detailed rules are available in Appendix C.

**Design Rationale.** As shown in Table 1, *Find the Spy* exemplifies High *Dynamic Interaction*. The multi-round format necessitates tracking clues revealed incrementally by all players over time. Conversely, *Information Uncertainty* is designed to be Low. Since players aim to avoid suspicion, they are incentivized to provide truthful descriptions of their assigned word, thereby significantly reducing the element of strategic deception.

**Data Generation and Quality Assurance.** For each game instance, we first set the parameters $n$ (number of players) and $T$ (number of rounds), then randomly selected a related word pair from a curated word bank, followed by random role assignment (one Spy and $n - 1$ Civilians). We then used a variety of LLMs to generate player descriptions for each of the $T$ rounds, simulating diverse communication styles. Prompt designs were crafted to encourage varied perspectives and expression strategies across rounds. To ensure quality and solvability, instances underwent human evaluation by 15 computer science graduate students. An instance was considered valid if a majority (>70%) of evaluators could uniquely identify the Spy based on the descriptions. 91% of the evaluated instances met this criterion, verifying their suitability for the benchmark.

## 3.3 Task 3: Rating Estimation from Text

**Task Rules.** This task aims to evaluate the ability of LLMs to predict a product's 1-to-5 star rating based on $n$ textual reviews, which may include genuine positive or negative user comments as well as promotional reviews written by shills. We collect two types of data: reviews generated by LLMs simulating different user types, and real user reviews scraped from platforms such as Amazon, the Google Play Store, and Taobao. The final rating prediction task follows a structure where information flows from multiple user nodes to a central product node, constituting a vertex-centric query focused on the product itself ($\mathcal{Q}_v(\text{product})$). Detailed task rules can be found in Appendix D.

**Design Rationale.** The task deliberately introduces a high level of *information uncertainty*. In the LLM-generated data, this is reflected through the inclusion of simulated "shill" users to mimic deceptive review behavior. In the real-world data, uncertainty arises from the inherent noise, subjectivity, and potential bias present in genuine user reviews. This setting requires the model to evaluate the credibility of information flowing from user nodes (i.e., reviewers) to the product node.

**Data Generation and Quality Assurance.** The LLM-based data generation process begins by sampling product attributes from a manually curated repository consisting of 1,000 attribute terms. A normal distribution of ratings is then constructed based on the true star rating, ensuring that the mean aligns with the reference score. Next, $n$ LLMs are randomly selected from a diverse model pool and probabilistically assigned roles (either normal users or shills) along with distinct personas, which guide the generation of textual reviews consistent with their assigned identities. For real-world data, we directly scrape product attributes and $n$ user reviews from platforms such as Amazon, the Google Play Store, and Taobao. An instance was considered solvable if a majority (>70%) of evaluators could correctly infer the true rating based solely on the textual reviews. Among the LLM-generated samples, 83% satisfied this criterion, confirming their validity for evaluating model reasoning.

## 3.4 Other Tasks

**Task4: Social Graph Analysis:** This task aims to evaluate the ability of LLMs to analyze relationships within a social group. Given a description of the social network graph and pairwise relationship labels indicating whether two individuals are friends or have a bad relationship—with friendship

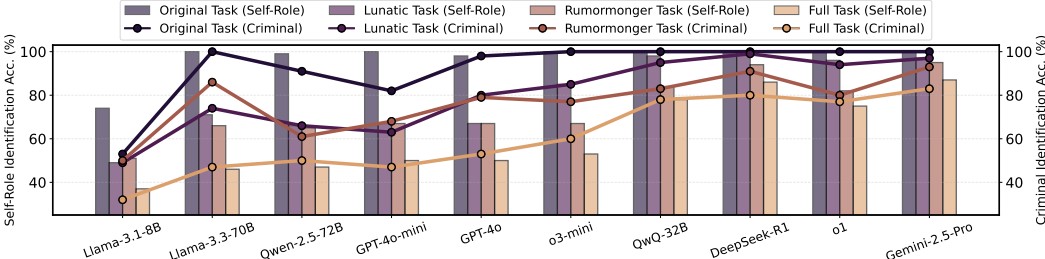

Figure 2: Model performance in *Hidden Role Deduction* across four task variants with increasing information uncertainty. Accuracy is shown after 3 rounds.

being transitive—the model is required to perform reasoning such as: determining whether two individuals are friends, identifying the friend group of a given node, calculating the total number of distinct friend groups, and counting all relationships within the network. Detailed task rules, the algorithmic data generation process that ensures logical consistency and solvability, and quality assurance procedures are provided in Appendix E.

**Task 5: Review Decision Prediction:** This task aims to evaluate the ability of LLMs to predict the final acceptance outcome (Accepted/Rejected) of a research paper as they gradually receive more information throughout the academic review process. The model is required to make a prediction at each of three interaction stages, with the available context incrementally expanding: in the first stage, only the initial paper information is provided; in the second stage, reviewer comments (with numerical scores removed) are added and the model must reason over both the initial content and the reviews; in the third stage, the full author rebuttal is introduced, completing the review context. This task simulates how opinions evolve over time in real academic peer review. Detailed task rules, the data generation process using real-world OpenReview data, and quality assurance procedures are provided in Appendix F.

**Task 6: User Profile Inference:** This task aims to evaluate the ability of LLMs to infer demographic attributes (age group and gender) based on user-generated textual reviews. Specifically, we construct a large number of users with known demographic attributes using LLMs, and generate their reviews for various products they have purchased. The inference tasks are twofold: (1) predicting the dominant user profile associated with the reviews of a specific product, and (2) identifying the profile of an individual user based on their reviews across multiple products. Detailed task rules, the LLM-based data generation method used to embed subtle demographic cues, and human validation results are provided in Appendix G.

## 4 Discussion

We conducted extensive experiments on the SocialMaze benchmark to evaluate the social reasoning capabilities of various LLMs. Specifically, we tested 5 leading proprietary LLMs and 6 open-weight LLMs across our tasks, covering diverse aspects of social reasoning. In addition, we evaluated 6 different workflow strategies to assess their impact on model performance. A subset of the results is visualized in Figure 3. We observe that different social reasoning tasks impose distinct demands on the models. For example, tasks like *Hidden Role Deduction*, which require *Deep Reasoning*, are best tackled by models such as DeepSeek-R1 and Gemini-2.5-Pro. In contrast, GPT-4o excels in tasks like *Review Decision Prediction*, where nuanced understanding of reviewer attitudes is critical. We conducted extensive case studies and report full results and settings in Appendix J and Appendix H.

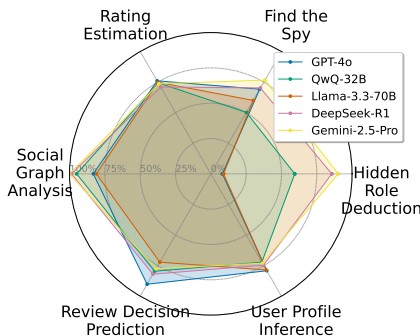

Figure 3: Performance comparison of selected LLMs on SocialMaze tasks, highlighting different model strengths.

## 4.1 The Impact of Deep Reasoning Requirements

In certain complex scenarios, effective social reasoning often requires going beyond surface-level cues—a process we refer to as *Deep Reasoning*. Our benchmark explicitly differentiates tasks along this dimension (Table 1), categorizing them into those that demand *Deep Reasoning* (*Hidden Role Deduction* and *Social Graph Analysis*) and those that are primarily solvable through more superficial. To assess the impact of reasoning depth, we compare two model categories: Long CoT models (e.g., o1, DeepSeek-R1), which generate detailed, step-by-step reasoning chains, and the remaining Short CoT models, which follow shorter reasoning paths—both using identical prompts.

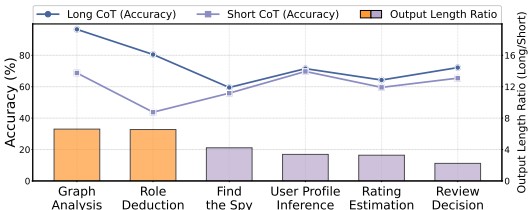

Figure 4: Performance comparison of Long CoT and Short CoT models. The line plot shows average accuracy; the bar plot shows the output length ratio (Long CoT / Short CoT). Orange bars indicate tasks with high deep reasoning demand, purple bars indicate low deep reasoning demand.

**Long CoT models achieve substantially higher accuracy on tasks requiring *Deep Reasoning*.** As shown in the line plot in Figure 4, the performance advantage of Long CoT models is particularly pronounced for *Deep Reasoning* tasks, i.e., Graph analysis and role deduction. While employing longer reasoning chains also yields modest improvements on shallow reasoning tasks, the accuracy gap between the two model types is significantly narrower. This indicates that the explicit, step-by-step reasoning characteristic of Long CoT models is especially beneficial for handling *Deep Reasoning* intricacies (e.g., inferring latent beliefs, analyzing relations); manual inspection confirmed this reasoning is sound and coherent in correct predictions, suggesting genuine inference capabilities.

**The improved performance of Long CoT models comes with a substantial computational cost.** The bar plot in Figure 4 illustrates this cost by presenting the ratio of average number of output tokens between Long CoT and Short CoT models. For *Deep Reasoning* tasks, outputs from Long CoT models contain nearly eight times more tokens on average, reflecting a much more extensive externalization of intermediate reasoning steps, hypothesis testing, and evidence evaluation. In contrast, for shallow reasoning tasks, the difference in output token count is less pronounced, mirroring smaller accuracy gains and suggesting that these tasks can often be solved without lengthy, explicit reasoning chains.

## 4.2 The Impact of Dynamic Interaction

In certain scenarios, social interactions unfold sequentially, requiring models to integrate and reason over information accumulated across multiple turns or stages. We analyze how model performance evolves in tasks characterized by high *Dynamic Interaction*, revealing distinct patterns depending on the nature and structure of the accumulating information. Overall, **model accuracy generally improves with quantitatively increasing interaction, but the trajectory of performance evolution and sensitivity to dynamic information vary significantly across different tasks and models.**

This contrast is particularly evident in two tasks: In *Hidden Role Deduction*, we track model accuracy in identifying the Criminal within the Full Task setting involving six players (Table 2). **Accuracy tends to increase as more rounds of interaction are observed**, reflecting the expected benefit of accumulating contextual evidence over time. However, the slope of improvement varies substantially across models, suggesting differing abilities to process and integrate newly revealed statements within the game's evolving context. Some models are more effective than others in leveraging additional rounds to refine their hypotheses. One notable insight is

Table 2: Criminal identification accuracy across rounds in the 6-player *Hidden Role Deduction* task. Models vary in leveraging *Dynamic Interaction*.

| Model | Round 1 | Round 2 | Round 3 |
|---|---|---|---|
| Llama-3.3-70B | 37.6% | 46.7% | 46.5% |
| Qwen-2.5-72B | 31.3% | 42.6% | 50.3% |
| GPT-4o-mini | 33.5% | 38.4% | 46.5% |
| GPT-4o | 39.5% | 53.3% | 53.5% |
| o3-mini | **45.8%** | 51.2% | 59.6% |
| QwQ-32B | 41.4% | 63.5% | 78.4% |
| DeepSeek-R1 | 44.3% | 72.3% | 80.4% |
| o1 | 42.5% | 67.5% | 76.6% |
| Gemini-2.5-Pro | 43.3% | **74.3%** | **87.6%** |

the large performance gap between Rumormonger and Lunatic. This is because once an Investigator correctly checks the Lunatic as "not the Criminal," it provides a strong signal that helps the Lunatic realize their true role—making awakening easier than for the Rumormonger.

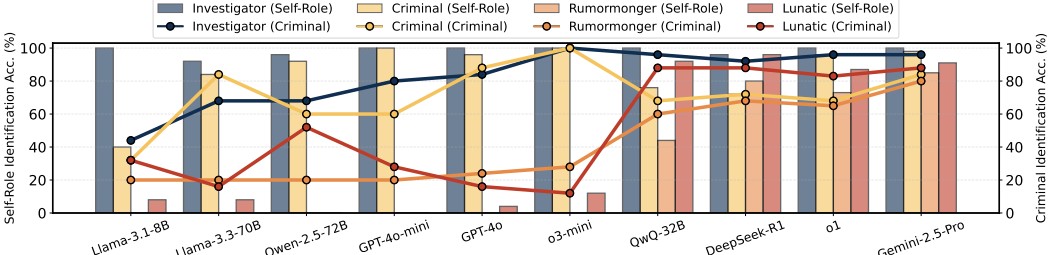

Figure 5: Performance in the Full task of *Hidden Role Deduction*, by model-assigned role. Models show reduced accuracy—especially in self-role identification—when assigned roles involving distorted self-perception (Rumormonger, Lunatic).

By contrast, *Review Decision Prediction* presents a more complex and non-linear performance trajectory across the stages of the peer review process, as shown in Table 3. **Initial paper information yields low accuracy, reviews trigger a major performance boost, but the final rebuttal stage often causes a drop in accuracy.** We observe that this counterintuitive decline is frequently driven by the model being swayed by the author's sincere and well-articulated defense, which may not align with the actual judgment rendered by human reviewers or area chairs. In other words, the model is "convinced" by the rebuttal, even when it fails to change the ultimate decision. A more detailed analysis of this phenomenon is provided in Appendix J.

## 4.3 The Impact of Information Uncertainty

A defining characteristic of complex social environments is the prevalence of unreliable information. We evaluate the impact of Information Uncertainty using the *Hidden Role Deduction* task by systematically introducing actors who generate distinct forms of unreliable information: intentional deception (Criminal) and noise stemming from flawed self-perception (Rumormonger and Lunatic).

**Increased information uncertainty significantly elevates the difficulty of social reasoning for LLMs.** Figure 2 illustrates this across four task configurations where uncertainty levels are quantitatively controlled by varying the number of unreliable actors (Rumormongers, Lunatics). Progressing from the baseline Original setting through scenarios introducing these noise sources to the complex Full setting, both the accuracy in identifying the Criminal (line plot) and the model's own role (bar plot) demonstrate a marked decline. This degradation underscores the substantial challenge posed by noise and deception.

Table 3: *Review Decision Prediction* accuracy across sequential stages. Accuracy improves with reviewer comments but often drops after incorporating the rebuttal.

| Model | Info. | Reviews | Rebuttal |
|---|---|---|---|
| Llama-3.1-8B | 37.0% | 79.6% | 62.0% |
| Llama-3.3-70B | 26.2% | 87.4% | 72.2% |
| Qwen-2.5-72B | 23.6% | 82.2% | 65.8% |
| Phi-4 | 23.5% | 77.6% | 61.4% |
| GPT-4o-mini | 38.8% | 85.2% | 85.0% |
| GPT-4o | **55.3%** | 86.2% | **90.2%** |
| o3-mini | 40.2% | 86.4% | 78.6% |
| QWQ-32B | 49.8% | 83.8% | 79.6% |
| DeepSeek-R1 | 50.2% | 88.0% | 82.0% |
| o1 | 52.2% | **88.6%** | 78.2% |
| Gemini-2.5-Pro | 47.4% | 87.6% | 77.6% |

**Reasoning becomes particularly challenging when the model's own perceived role or information source is compromised.** Figure 5 delves into the Full task configuration, analyzing final accuracy based on the specific role assigned to the LLM. Models exhibit considerably lower accuracy—especially in identifying their own role (bars)—when assigned as a Rumormonger or Lunatic compared to being an Investigator or Criminal. This suggests a significant difficulty in reconciling internal beliefs (e.g., "I think I am an Investigator/Criminal") with conflicting external evidence or the unreliable nature of one's own generated information.

These experiments also highlight critical differences between model capabilities. **Existing Short CoT models demonstrate severe limitations in handling complex scenarios with high information uncertainty**, often failing to perform reliably in the Rumormonger and Full task settings (Figure 2). In contrast, Long CoT models, while still impacted, exhibit significantly better resilience to uncertainty. Furthermore, the challenge of self-assessment under uncertainty exposes a stark gap: **Short CoT models are almost entirely unable to deduce their true identity when assigned as a**

Table 4: Performance on *Hidden Role Deduction* before and after fine-tuning. **Crim.**: Criminal prediction accuracy; **Self**: self-role prediction accuracy; **Both**: both predictions correct. Fine-tuning with SFT and DPO significantly improves performance on both models. Percent improvements over the Base model for **Both** are shown in parentheses.

| Model | Base | | | SFT | | | DPO | | |
|---|---|---|---|---|---|---|---|---|---|
| | **Crim.** | **Self** | **Both** | **Crim.** | **Self** | **Both** | **Crim.** | **Self** | **Both** |
| LLaMA-3.1-8B | 33.0% | 8.4% | 2.0% | 37.0% | 15.2% | 13.4% (+11.4%) | 35.4% | 11.0% | 9.8% (+7.8%) |
| Phi-4 | 31.2% | 13.4% | 8.2% | 38.2% | 22.6% | 19.8% (+11.6%) | 37.8% | 17.4% | 15.2% (+7.0%) |

**Rumormonger or Lunatic** (Figure 5), suggesting a profound lack of capacity for self-doubt and meta-reasoning necessary to overcome compromised initial information.

### 4.4 Enhancing Social Reasoning Capabilities

To explore strategies for improving social reasoning in LLMs, we conduct focused experiments on the *Hidden Role Deduction* task. This task is not only uniquely representative—combining Deep Reasoning, Dynamic Interaction, and Information Uncertainty—but also well-suited for generating diverse reasoning examples at scale, providing valuable supervision for model learning.

**Reasoning agents and workflows offer limited gains for social reasoning.** We first assess whether reasoning agents and workflows effective in task decomposition and planning can improve performance on *Hidden Role Deduction*. As shown in Table 5, various agentic implementations yield only marginal improvements over base models. This indicates that current workflow strategies are not enough to handle the complexity and uncertainty involved in social reasoning tasks.

**Fine-tuning on curated reasoning traces substantially improves performance.** Recognizing the need for models to internalize complex

Table 5: Performance of LLM agents and workflows on the *Hidden Role Deduction* task. All workflows use the better-performing model between Phi-4 and GPT-4o-mini as the base model.

| Method | Crim. | Self | Both |
|---|---|---|---|
| QwQ | 63.8% | 63.2% | 59.4% |
| DeepSeek-R1 | **87.6%** | **88.6%** | **85.6%** |
| LLM-Debate [55] | 42.0% | 13.2% | 12.2% |
| Self-refine [56] | 33.2% | 11.2% | 10.4% |
| ADAS [51] | 36.6% | 8.4% | 6.0% |
| AFlow [52] | 40.2% | 12.4% | 11.6% |
| MaAS [53] | 44.4% | 15.0% | 13.8% |
| DyFlow [57] | 43.2% | 17.6% | 16.8% |

reasoning strategies, we further explore instruction-based fine-tuning using high-quality examples from our dataset. Table 4 summarizes results from applying Supervised Fine-Tuning (SFT) [58] and Direct Preference Optimization (DPO) [59] to Llama-3.1-8B and Phi-4. Both approaches substantially improve accuracy on the *Hidden Role Deduction* task, and also yield slight but consistent gains on other benchmark tasks, thereby demonstrating the effectiveness and generalizability of targeted fine-tuning on curated reasoning examples.

These findings highlight a fundamental limitation of existing LLM agents in socially complex reasoning tasks. In contrast, our targeted fine-tuning approach yields substantial improvements, particularly because it leverages high-quality reasoning traces specifically crafted for social contexts. This suggests that equipping models with domain-relevant reasoning strategies through fine-tuning may be a more fruitful path toward enhancing their capabilities in this domain.

## 5 Conclusion

We introduced SocialMaze, a benchmark designed to rigorously evaluate the social reasoning capabilities of LLMs by capturing the challenges of deep reasoning, dynamic interaction, and information uncertainty. Experiments across six diverse tasks reveal notable weaknesses in current models, particularly in handling evolving contexts and reasoning under uncertainty. Targeted fine-tuning on curated reasoning examples significantly improves performance, highlighting the value of domain-specific adaptation. As future work, we plan to further expand SocialMaze by collecting more real-world data and aim to enrich the benchmark with new social scenarios and task types. We believe our work will provide valuable resources and insights to the research community, helping to advance the development of LLMs with stronger social reasoning capabilities.

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

# A Related Works

## A.1 Static Social Reasoning Benchmarks

Early evaluations of social reasoning in language models largely focus on static, single-turn tasks where models infer plausible answers based on brief, pre-written scenarios. These benchmarks primarily test shallow forms of commonsense and moral reasoning without requiring interaction, adaptation, or uncertainty handling. A classical example is **SocialIQA** [20], which assesses social commonsense by posing questions about motivations and reactions in everyday situations. Similarly, **ATOMIC** [21] provides a structured knowledge graph of causal social events, while **CommonGen** [24] tests the ability to generate plausible sentences from object co-occurrence tuples in socially relevant scenes.

Benchmarks like **Social Chemistry** [27] and **ETHICS** [28] target moral norms, asking models to judge the appropriateness of actions, whereas **GLUCOSE** [60] emphasizes implicit causal reasoning in narratives. Emotion understanding is covered by **GoEmotions** [23], while **PIQA** [61], **HellaSwag** [62], and **CREAK** [22] assess commonsense or contextually grounded inference.

Recent work has shifted toward evaluating *Theory of Mind* (ToM)—the capacity to reason about others' beliefs and intentions. Initial tests, such as those in [63], used classic false-belief stories to probe emergent ToM in LLMs, though follow-up studies revealed artifacts and prompt sensitivity [25]. More recent evaluations such as **FANToM** [64] and **ToMValley** [65] provide more rigorous multi-turn belief-tracking scenarios. These benchmarks demonstrate that, while LLMs may succeed on simple static ToM questions, they struggle with deeper or dynamic belief modeling, particularly under information asymmetry or belief shifts over time.

Overall, while these static benchmarks have advanced our understanding of LLMs' social reasoning capabilities, they often lack three essential ingredients for real-world social cognition: dynamic interaction, complex reasoning depth, and reasoning under uncertainty. Table 6 summarizes several representative benchmarks discussed above, highlighting their focus, format, and limitations. These gaps motivate the design of SocialMaze, which aims to capture the challenges of socially grounded reasoning in more realistic and interactive settings.

Table 6: Representative Benchmarks for Social Reasoning in LLMs

| Benchmark | Focus | Format | Key Findings |
|---|---|---|---|
| **SocialIQA** [20] | Commonsense QA | Static narrative + MCQ | GPT-3+/GPT-4 near human; tests scripted social norms. |
| **ToM Classic** [63] | False belief | Brief stories + Q&A | Early ToM claims challenged; shortcut artifacts noted. |
| **Clever Hans** [25] | ToM artifacts | Controlled stimuli | Performance drops without spurious cues; lacks robustness. |
| **FANToM** [64] | Interactive ToM | Multi-turn dialogue | GPT-4 and others fail under asymmetric info tracking. |
| **ToMValley** [65] | Dynamic ToM | Scenario chains + Q&A | LLMs 11% below humans; weak on mental state updates. |

## A.2 Dynamic Social Reasoning and Interaction Benchmarks

Another class of benchmarks focuses on dynamic settings where social reasoning occurs within interactive contexts. These tasks require models to engage with evolving information, track perspectives, and reason about hidden roles, intentions, or potential deception. DebateBench [29] evaluates models' reasoning across long-form argumentative dialogues. Studies of peer review processes using OpenReview data [66, 67] examine decision-making through multi-turn, text-based interaction without relying solely on reviewer scores. Strategic games such as Diplomacy [31] and Poker [68], as well as social deduction games like Avalon and Werewolf [32, 14, 33, 34], provide natural settings for reasoning under incomplete information and complex social dynamics. Multi-agent simulations such as Generative Agents [45] explore emergent social behavior, while robustness evaluations like TextAttack [69], TRUST-LLM [70], and [71] investigate model behavior under uncertainty or adversarial conditions.

While these benchmarks incorporate certain forms of interaction and complexity, most remain oriented toward task completion or strategic decision-making. They often lack systematic modeling of social context factors such as role motivations or deception. SocialMaze, by contrast, integrates interaction and uncertainty within a structured framework, explicitly targeting complex social reasoning.

# B    Hidden Role Deduction

This appendix provides the details for the Hidden Role Deduction task described in subsection 3.2.

**1. Task Setup**

The game involves $n$ players, denoted $s_1, ..., s_n$. The LLM observes the game unfolding from the perspective of Player 1 ($s_1$), receiving all interaction data and the initial role assignment given to $s_1$. **Crucially, the LLM does not actively participate in the game simulation (i.e., does not choose targets or make statements). Its task is solely post-hoc inference**: analyzing the complete interaction log ($G_1, ..., G_T$) to deduce the answers to the specified queries. At the start, each player $s_i$ is randomly assigned a secret role. The LLM is informed of the role initially assigned to $s_1$; however, this assigned role may differ from $s_1$'s actual role, especially if $s_1$ is a Rumormonger or Lunatic. The specific composition of roles depends on the Task Variant being used.

**2. Roles and Behaviors**

Simulated player behavior is guided by their assigned role and the game history $G_t$ up to round $t$. The roles are defined as follows: The **Investigator (I)** aims to identify the Criminal. Based on the game history $G_t$, an Investigator uses an algorithmically defined function $F_I(G_t)$ to select a target player $u$. They then make a **truthful** statement reflecting their deduction about $u$'s status (Criminal or not Criminal). The function $F_I(G_t)$ heuristically assesses suspicion, selecting players with higher probability if they have made contradictory statements or have been accused by others, while lowering the probability for players cleared by multiple potentially reliable sources. The **Criminal (C)** seeks to avoid identification and mislead others. They employ a strategic function $F_C(G_t)$ to select a target player $u$ and also determine a probability $p_t = P(\text{state } u \text{ is Criminal}|G_t, \text{role=C})$. The Criminal then states "$u$ is Criminal" with probability $p_t$ and "$u$ is not Criminal" with probability $1 - p_t$. The function $F_C$ implements deceptive tactics, prioritizing targeting players who have accused the Criminal and diverting suspicion onto others. The **Rumormonger (R)**, although believing they are an Investigator trying to identify the Criminal, unintentionally provides unreliable information, effectively injecting noise. They are *told* they are an Investigator and use the Investigator logic function $F_I(G_t)$ to select a target player $u$. However, regardless of any internal assessment derived from $F_I(G_t)$ or the actual ground truth, the truthfulness of their final statement ("$u$ is Criminal" or "$u$ is not Criminal") is entirely random, possessing a 50% probability of aligning with the ground truth and a 50% probability of contradicting it. Lastly, the **Lunatic (L)** believes they are the Criminal and aims to avoid identification based on this false premise, while their actual nature is not Criminal. They are *told* they are the Criminal and mimic the Criminal's behavior by employing the *same* strategic function $F_C(G_t)$ used by the actual Criminal. Although their actions follow deceptive patterns, truthful statements made by Investigators about the Lunatic will correctly identify them as 'not the Criminal'.

**3. Interaction Rounds**

The game simulation proceeds for a fixed $T$ rounds. In each round $t$ (from 1 to $T$), every player $s_v$ selects another player $P_u$ and makes a public statement of the form: "Player $v$ says Player $u$ is the criminal" or "Player $v$ says Player $u$ is not the criminal." All statements made in round $t$ are revealed simultaneously to all players, and thus become available to the observing LLM, before round $t + 1$ begins. Consequently, the complete history $G_T = (\text{statements}_{round1}, ..., \text{statements}_{roundT})$ is available for the LLM's final analysis.

**4. Parameter Settings**

The composition of roles is varied to create tasks of differing complexity, ensuring there is always exactly one Criminal. The variants include: the **Original Task** (1 Criminal, $n - 1$ Investigators); the **Rumormonger Task** (1 Criminal, $x \geq 1$ Rumormongers, $n - 1 - x$ Investigators); the **Lunatic Task** (1 Criminal, $y \geq 1$ Lunatics, $n - 1 - y$ Investigators); and the **Full Task** (1 Criminal, $x \geq 0$ Rumormongers, $y \geq 0$ Lunatics, $n - 1 - x - y$ Investigators, where $x + y \geq 1$). **Note on Experimental**

**Configuration:** For the experiments presented in this paper, the game parameters were fixed at $n = 6$ players. When Rumormongers and Lunatics were included (specifically in the Full Task variant experiments), their counts were set to $x = 1$ and $y = 1$, respectively, alongside 1 Criminal and $n - 1 - x - y = 3$ Investigators. The accompanying open-sourced dataset includes generated instances for both $n = 6$ and $n = 10$. Furthermore, the open-sourced data generation code is flexible, allowing users to configure $n, m, x$, and $y$ to create custom game scenarios.

### 5. Quality Control

The algorithmic generation includes verification via heuristic search, ensuring a unique, logically derivable solution exists for both queries from $P_1$'s perspective using only the interactions and rules. The core verification logic, which checks all valid hypotheses, is outlined in algorithm 1.

Furthermore, to validate the models' reasoning quality, we manually inspected 100 correct responses each for several key models (DeepSeek-R1, QwQ-32B, Gemini-2.5-Pro, o1, and o3-mini). This analysis confirmed that over 90% of these successful predictions were underpinned by reasoning processes assessed as both rigorous and logically sound.

---

**Algorithm 1:** Solvability Verification from $P_1$'s Perspective

**Input** : Interaction Log $S$, Player Set $\mathcal{P} = \{P_1, \ldots, P_n\}$, Role Set $\mathcal{R} = \{I, C, R, L\}$, Investigator Count $N_I$

**Output :** Unique Solution $(C^*, R_1^*) \in \mathcal{P} \times \mathcal{R}$, or $\emptyset$ if not unique/no solution

1   $\mathcal{S}_{valid} \leftarrow \emptyset$;
   ▷ Set to store valid (Criminal, P1 Role) solution pairs

2 **foreach** *hypothesized role* $R_1^{hyp} \in \mathcal{R}$ *for* $P_1$ **do**

3    |   $\mathcal{P}_{cand} \leftarrow \mathcal{P} \setminus \{P_1\}$;
     ▷ Initial candidates are all others

4    |   **foreach** *subset* $I_{hyp} \subseteq \mathcal{P}_{cand}$ *such that* $|I_{hyp}| = N_I$ **do**

5    |    |   Let current hypothesis $H = (R_1^{hyp}, I_{hyp})$;

6    |    |   **if** $IsConsistent(S, H)$ **then**
         ▷ Check if hypothesis contradicts any statement

7    |    |    |   $C_{implied} \leftarrow \text{DeduceCriminal}(S, H)$;

8    |    |    |   $R_{1,implied} \leftarrow \text{DeduceP1Role}(S, H)$;

9    |    |    |   **if** $C_{implied} \neq NULL$ **and** $R_{1,implied} \neq NULL$ **then**

10    |    |    |    |   $\mathcal{S}_{valid} \leftarrow \mathcal{S}_{valid} \cup \{(C_{implied}, R_{1,implied})\}$;
         ▷ Add deduced solution

11 **if** $|\mathcal{S}_{valid}| = 1$ **then**

12    |   **return** the single element in $\mathcal{S}_{valid}$;
     ▷ Unique solution found

13 **else**

14    |   **return** $\emptyset$;
     ▷ Not unique or no consistent solution

---

## C   Find the Spy

This appendix provides the details for the *Find the Spy* task described in subsection 3.2.

### 1. Task Setup

The game involves $n$ players, denoted $s_1, \ldots, s_n$. The evaluated LLM adopts the persona of Player 1 ($s_1$) but acts as a passive observer. It receives all game information, including its own assigned word and all player descriptions, from $s_1$'s perspective but does not generate descriptions itself during the evaluation process. The LLM's sole task is to identify the Spy based on the observed interactions. For word assignment, a pair of semantically related but distinct words (Word A, Word B, e.g., "Milk" and "Soy Milk") is selected from a predefined bank. One player is randomly designated as the Spy,

while the remaining $n-1$ players are Civilians. All Civilians receive Word A, and the Spy receives Word B. Player 1 (the LLM's persona) is informed of the word it received but is not explicitly told whether it is Word A or Word B, nor is it told its role (Civilian or Spy).

**2. Interaction Rounds**

The game proceeds for $T$ rounds. In each round $t$ (from 1 to $T$), every player $s_i$ provides a textual description of the word they possess. These descriptions are generated by LLMs, with each player $s_i$ being assigned a specific LLM generator (selected randomly and uniformly from a predefined pool: GPT-4o-mini, GPT-4o, Llama-3.3-70B, Qwen-2.5-72B) for the entire game instance. The generation prompts encourage the LLM simulating player $s_i$ to describe their word from different angles or aspects in each round, avoiding simple repetition. Players are assumed to provide descriptions consistent with the word they hold: Civilians describe Word A, and the Spy describes Word B. All descriptions generated in round $t$ are made available to all players, including the LLM as $s_1$, before round $t+1$ begins. Thus, the complete history of descriptions from rounds 1 to $T$ is available at the end of the game for the LLM's analysis.

**3. Parameter Settings**

For all experimental evaluations using this task setup, the number of players $n$ was fixed at 4, and the number of interaction rounds $T$ was fixed at 3.

**4. Quality Control**

To ensure generated instances are solvable yet challenging, they undergo a rigorous human validation process. Each potential game instance is reviewed by 15 evaluators, all holding at least an undergraduate degree in Computer Science. An instance is deemed valid and solvable only if a clear majority (more than 5 out of the 10 evaluators) agree that the collective descriptions provided by the simulated players contain sufficient evidence to uniquely identify the Spy. This validation confirms that the task's difficulty stems from semantic subtlety and variations in description style, rather than from a fundamental lack of necessary information. 91% of the instances subjected to this evaluation met the validation threshold, affirming their suitability for inclusion in the benchmark.

# D    Rating Estimation from Text

This appendix provides the details for the Rating Estimation from Text task described in subsection 3.3.

**1. Task Setup**

The primary objective for the LLM in this task is to estimate the most likely overall "true" star rating, represented as an integer from 1 to 5, for a given product. This estimation must be based solely on the textual content derived from multiple user reviews. For each task instance, the LLM receives specific input: a set of attributes describing the product (such as type, category, key features, price range) and a list containing $n$ individual textual user reviews for that product. Crucially, the original star ratings (1-5 stars) that reviewers might have provided are explicitly omitted from the input. After processing the product information and the $n$ textual reviews, the LLM must answer a vertex-centric query ($\mathcal{Q}_v(\text{product})$) phrased as: "Based on the provided reviews, what is the most likely overall star rating for this product? Choose one: 1, 2, 3, 4, or 5." The expected output is a single integer, necessitating the synthesis of information from multiple user interactions (reviews) all directed towards the product entity.

**2. Data Generation**

The task utilizes review data generated through two distinct methods. The first method involves **LLM-Generated Reviews**. Here, a product profile is selected, including its attributes and a designated ground truth overall star rating. A number $n$ of simulated reviewers is determined, and each reviewer $i$ is assigned a role probabilistically, ensuring the majority are *Normal Users* (providing honest feedback) while a small fraction are designated as *Positive Shills* or *Negative Shills*. Each reviewer also receives a simple persona for stylistic variation and is assigned a randomly selected LLM from a predefined pool (GPT-4o-mini, GPT-4o, Llama-3.3-70B, Qwen-2.5-72B) to simulate their response. The assigned LLM then generates the review text for reviewer $i$, guided by the product attributes, the ground truth rating, the reviewer's assigned role (Normal/Shill), and their persona; Shills are

prompted to generate biased text accordingly. Finally, only the generated textual reviews are collected and prepared as input for the evaluated LLM. The second method uses **Real-World Reviews**. In this scenario, product attributes along with user reviews (both text and original ratings) are scraped from public e-commerce and app platforms. For a selected product, $n$ reviews are sampled from the scraped collection. The textual content of these $n$ sampled reviews is extracted, while the original star ratings are discarded. Only the product attributes and the review texts are provided as input to the LLM. The ground truth for these instances is typically derived from the average rating found on the source platform, which the LLM is tasked to estimate.

### 3. Parameter Settings

Across all experiments presented for this task, the number of reviews ($n$) provided per product instance is consistently set to 8. The required output is always a single integer rating on the 1-to-5 star scale. The real-world task instances exclusively utilize product attributes and textual reviews sourced from **Amazon**, the **Google Play Store**, and **Taobao**. For the LLM-generated instances, the underlying product attributes and the initial ground truth star ratings are also sampled from this same pool of real-world data derived from these platforms, providing a basis grounded in realistic product scenarios.

### 4. Quality Control

As detailed in subsection 3.3, human evaluations were integral to ensuring data quality. These were conducted by 15 graduate students. For the LLM-generated data (Scenario A), this evaluation determined that 83% of the assessed instances were solvable, meaning the true rating could be reasonably inferred from the text alone by more than half of the human evaluators. For the instances derived from real-world data (Scenario B), solvability is inherently tied to the complexity and nature of authentic customer feedback as it appears on platforms like Amazon, Google Play, and Taobao, reflecting genuine information landscapes.

# E   Social Graph Analysis

This appendix provides the details for the Social Graph Analysis task described in Section 3.4.

### 1. Setup

This task presents a stylized social network scenario involving a set of $n$ individuals. The core challenge lies in understanding the structure of this network, where relationships between any two individuals are strictly defined as either 'good' or 'bad'. The LLM is provided with a complete description of all pairwise relationships and must then analyze this information to answer queries about specific relationships, individual connections, and the overall emergent group structure of the network, guided by a set of simple logical axioms.

### 2. Relationship Axioms

Relationships between any two distinct individuals, say Person A and Person B, are binary ('good' or 'bad') and symmetric. These relationships are governed by specific logical rules: **Axiom 1 dictates the transitivity of good relationships**, meaning if A and B have a 'good' relationship, and B and C also have a 'good' relationship, then A and C must necessarily have a 'good' relationship. **Axiom 2 describes the implication of bad relationships**, stating that if A and B have a 'bad' relationship, and A and C have a 'good' relationship, then B and C are forced to have a 'bad' relationship. It's important to note that from these axioms, if A and B share a 'bad' relationship, and B and C also share a 'bad' relationship, the nature of the relationship between A and C is not determined solely by these two facts; it could be either good or bad depending on other connections within the network. However, the algorithmic generation process always ensures a globally consistent and valid relationship structure.

### 3. Group Definition

Within this social structure, a 'group' is formally defined as a maximal set of individuals where every person within that set has a 'good' relationship with every other person also belonging to that same set. A key property of this structure is that every individual belongs to exactly one such group. Consequently, based on the governing axioms and the generation method, the relationship between any two individuals can be directly inferred from their group membership: if Person A and Person B

---

**Algorithm 2:** Social Graph Generation

---

**Input** : Difficulty level $d \in \{\text{easy}, \text{hard}\}$
**Output** : Natural language instance $I$

---

1 Set number of individuals $n \sim \begin{cases} [8, 10], & \text{if } d = \text{easy} \\ [14, 16], & \text{if } d = \text{hard} \end{cases}$;

2 Initialize graph $G = (V, E)$, where $|V| = n$;

3 **Step 1: Generate spanning forest to define social groups**;

4 Randomly partition $V$ into $m \geq 2$ disjoint non-empty subsets $\{V_1, V_2, \ldots, V_m\}$;

5 **foreach** *group $V_i$* **do**

6      Generate a spanning tree $T_i = (V_i, E_i^{\text{good}})$;

7      Add edges $E_i^{\text{good}}$ to $E$ // `Intra-group "good" relationships`

8 **Step 2: Add "bad" edges between groups**;

9 **foreach** *pair of groups $(V_i, V_j)$, $i \neq j$* **do**

10      Select a node pair $(u, v) \in V_i \times V_j$;

11      Add edge $(u, v)$ to $E^{\text{bad}} \subset E$ // `Inter-group "bad" relationship`

12 **Step 3: Convert graph structure to natural language**;

13 **foreach** *edge $(u, v) \in E$* **do**

14      **if** $(u, v) \in E^{good}$ **then**

15          Generate statement: "$u$ and $v$ are good friends.";

16      **else if** $(u, v) \in E^{bad}$ **then**

17          Generate statement: "$u$ and $v$ do not get along.";

18 Aggregate all generated statements into input instance $I$;

19 **return** $I$;

---

are members of the same group, they inherently have a 'good' relationship; conversely, if they belong to different groups, they must have a 'bad' relationship.

### 4. Input Format

The LLM receives as input a comprehensive list composed of natural language statements that explicitly specify **the complete set of pairwise relationships** as determined by the algorithmic generation process. These statements clearly define the relationship status between every possible pair of individuals within the scenario. Examples of such input statements include "Person N and Person G have a good relationship" and "Person K and Person P have a bad relationship". This list provides a full and unambiguous description of the entire social graph structure.

### 5. LLM Queries

After processing the complete list of relationship statements provided as input, the LLM is required to answer various types of queries designed to test its understanding of the network structure. These queries include, for instance, **Pairwise Relationship Queries** ($\mathcal{Q}_e(v_i, v_j)$), such as "Do Person N and Person L have a good relationship?", which typically requires checking the provided input directly and responding with Yes/No. Other queries are **Good Relationship Neighbor Queries (Vertex-centric)**, like "Who has a good relationship with Person H?", demanding the LLM to filter the input and list the relevant names. Furthermore, **Graph-level Queries** ($\mathcal{Q}_G$) probe the overall structure, asking questions like "How many groups of people are there?" or "How many pairs of people have good relationships?" or "How many pairs of people have bad relationships?", all of which require synthesizing the pairwise information to derive a global property and respond with an integer count.

### 6. Data Generation and Quality Assurance

Instances for this task are generated entirely algorithmically, without reliance on LLM generation, ensuring consistency and verifiable ground truth. The process begins by setting the number of individuals $n$, sampled from [8, 10] for 'easy' instances and [14, 16] for 'hard' instances. A complete graph structure respecting the relationship axioms is then algorithmically constructed. First, a spanning forest is created using only 'good' relationship edges, thereby defining the distinct social

groups (each tree representing a group). Second, 'bad' relationship edges are strategically added to connect every pair of distinct groups (trees), ensuring all inter-group relations are 'bad' and all intra-group relations are 'good'. The complete set of generated relationship edges ('good' edges defining the groups and 'bad' edges connecting them) is then converted into natural language statements and presented to the LLM as input. This generation methodology mathematically guarantees that for each instance, a unique solution exists for all four query types and is logically derivable solely from the provided statements and rules. The core generation logic is outlined in algorithm 2.

# F   Review Decision Prediction

This appendix provides the details for the Review Decision Prediction task described in subsection 3.4.

**1. Objective**

The LLM's goal in this task is to predict the final acceptance status (Accepted or Rejected) of a research manuscript submitted to a conference, based solely on the sequence of provided peer review communications.

**2. Data Source and Scope**

Data for this task is exclusively sourced from the official OpenReview API, encompassing submissions to specific high-profile Artificial Intelligence and Machine Learning conferences, namely NeurIPS (covering the 2023 and 2024 cycles) and ICLR (covering the 2020, 2021, 2022, 2023, and 2024 cycles). **The rationale for selecting these particular venues** stems primarily from their policy of making the entire peer review process public. This transparency, which *crucially includes making detailed reviews and discussions for rejected manuscripts publicly available*, is a practice not commonly found in many other academic fields. It provides the essential data needed to construct a balanced and realistic task dataset that accurately reflects both acceptance and rejection scenarios encountered in academic publishing.

**3. Input Structure**

The LLM receives information pertaining to a single manuscript, presented in a structured sequence that mirrors the typical progression of the peer review timeline. Initially, in **Round 1**, the LLM is given the initial submission details: the manuscript's original **Title**, its **Abstract**, and the author-provided **Keywords**. Subsequently, in **Round 2**, the LLM receives the reviewer feedback, which consists of the complete **textual content** of each review submitted by the assigned reviewers. It is **crucial to note the exclusion** of all quantitative aspects from these reviews; numerical scores (such as overall ratings, technical soundness, or novelty scores), reviewer confidence scores, explicit recommendations (like Accept, Reject), and any other non-textual evaluation metrics are deliberately removed. The input at this stage contains only the narrative comments written by the reviewers. Finally, **Round 3** provides information from the author-reviewer discussion phase, including the full text of the authors' **rebuttal** designed to address the initial reviewer comments, as well as any subsequent **comments or discussions** exchanged between the authors and reviewers following the rebuttal. The full manuscript text itself is intentionally omitted from the input provided to the LLM. This decision is driven by two main factors: practical challenges related to processing lengthy full papers consistently across numerous task instances, considering LLM input constraints and computational costs, and more importantly, to align with the task's core objective. This objective focuses on evaluating the LLM's ability to comprehend and synthesize the dynamics inherent in the peer review dialogue—interpreting arguments, discerning attitudes, and understanding sentiments expressed by reviewers and authors—rather than tasking it with performing an independent technical re-evaluation of the manuscript's content.

**4. Ground Truth and Quality Assurance**

The ground truth for this task is inherently robust, as it consists of the verified, real-world acceptance or rejection decisions obtained directly from the OpenReview API for the specified conferences (NeurIPS 2023-2024, ICLR 2020-2024). To further validate the task's premise—specifically, whether the final outcome is typically discernible from the textual dialogue alone (Title, Abstract, Keywords, Reviews, Rebuttal) after removing numerical scores—we conducted supplementary human evaluations. Human evaluators were presented with the same sequential information provided to the LLM and asked to predict the final decision. For over 90% of the evaluated manuscript instances, the true outcome was

deemed inferable from the textual evidence by a majority (>70%) of the human evaluators. This confirms the general solvability of the task based on the provided textual interactions and reinforces its suitability for assessing an LLM's ability to synthesize argumentative dialogue, complementing the reliability provided by the authentic ground truth data.

# G  User Profile Inference

This section provides the details for the User Profile Inference task, corresponding to subsection 3.4.

**1. Task Setup**

For each instance of this task, a population of $n$ simulated users is defined. Every user $u_i$ within this population is assigned a specific demographic profile, which consists of an age group selected from '18-34', '35-54', '55+' and a gender selected from 'Male', 'Female', 'Non-binary'. This profile assignment is carried out probabilistically, with the process intentionally tuned to often establish a statistically dominant age-gender combination within the user pool. This characteristic is particularly relevant for addressing the "dominant audience" query type. Additionally, a predefined pool of items, each described by a name and a brief description, is utilized. Users are randomly assigned items from this pool, about which they will generate comments.

**2. Comment Generation Process**

Each simulated user $u_i$ is associated with a specific Large Language Model (LLM), chosen randomly from a diverse pool that includes models such as GPT-4o-mini, GPT-4o, Llama-3.3-70B, and Qwen-2.5-72B. The core of the generation process involves tasking the LLM associated with user $u_i$ (who has an assigned age group $A_i$ and gender $G_i$) to generate a textual comment about a selected item $j$ (which has a specific type $T_j$ and subject $S_j$). The LLM is prompted to generate content that reflects the assigned persona interacting with the given item.

**3. LLM Queries**

Based on the generated comments provided as input, the evaluated LLM must answer one of two specific types of queries. The first is the **Item Audience Profile Inference (Vertex-centric Query** $\mathcal{Q}_v(\textbf{Item})$). For this query, the LLM is asked, "Based on the provided comments for the item '[Item Name]', what is the most likely dominant audience profile (Age Group and Gender)? Choose from Age Groups: ['18-34', '35-54', '55+'] and Genders: ['Male', 'Female', 'Non-binary']." Answering this requires synthesizing information from multiple user comments linked to a specific item node to infer an aggregated characteristic of its audience. The second type is the **User Profile Inference (Vertex-centric Query** $\mathcal{Q}_v(\textbf{User})$), which poses the question: "Based on the provided comments from this user, what is their most likely profile (Age Group and Gender)? Choose from Age Groups: ['18-34', '35-54', '55+'] and Genders: ['Male', 'Female', 'Non-binary']." This query demands synthesizing information from multiple comments generated by a single user node, potentially across different items, to infer the intrinsic demographic attributes (age group and gender) of that specific user.

**4. Quality Assurance**

The dataset for this task was entirely generated using LLMs. We first defined a set of user personas by assigning age group and gender attributes, ensuring through probabilistic assignment that certain demographic combinations were more prevalent to create a potential "dominant audience" for item-centric queries. Items with names and descriptions were sampled from a predefined pool. Various LLMs were then assigned to personas and prompted to generate comments on these items, reflecting their designated age and gender characteristics. To ensure task validity, we conducted human evaluations with 15 computer science graduate students. For the item-audience query, 78% of instances were deemed solvable (dominant audience inferable) by a majority (>70%) of evaluators. For the user-profile query, 85% of instances were similarly validated, confirming that the generated comments contain sufficient, albeit subtle, cues for demographic inference.

Table 7: Models used in our experiments along with their versions, organizations, licenses, and purposes. *Eval*: Model used for evaluation; *FT*: Model used for fine-tuning.

| Model | Version | Organization | License | Eval | FT |
|---|---|---|---|---|---|
| Phi-4 | Phi-4 | Microsoft | MIT | ✓ | ✓ |
| GPT-4o-mini | gpt-4o-mini-2024-07-18 | OpenAI | Proprietary | ✓ | |
| GPT-4o | gpt-4o-2024-08-06 | OpenAI | Proprietary | ✓ | |
| Llama-3.1-8B | Meta-Llama-3.1-8B-Instruct | Meta | Llama 3.1 Community | ✓ | ✓ |
| Llama-3.3-70B | Meta-Llama-3.3-70B-Instruct | Meta | Llama-3.3 | ✓ | |
| Qwen2.5-72B | Qwen2.5-72B-Instruct | Alibaba | Qwen License | ✓ | |
| QwQ | QwQ-32B | Alibaba | Apache 2.0 | ✓ | |
| o3-mini | o3-mini-2025-01-31 | OpenAI | Proprietary | ✓ | |
| o1 | o1-2024-12-17 | OpenAI | Proprietary | ✓ | |
| Deepseek-R1 | DeepSeek-R1 | DeepSeek | MIT | ✓ | |
| Gemini-2.5-Pro | Gemini-2.5-Pro-Exp-03-25 | Google | Proprietary | ✓ | |

# H Experiment Details

This appendix provides detailed information regarding the experimental setup, the models evaluated, data generation procedures for each task within the SocialMaze benchmark, the experimental methodology, and a summary of the overall results.

## H.1 Baselines

As detailed in Table 7, we utilized five proprietary models: GPT-4o [72], GPT-4o-mini [73], o3-mini [74] , o1 [75], and Gemini-2.5-Pro [76]. In addition, we included six open-weight models: Phi-4 [77], Llama-3.1-8B [78], Llama-3.3-70B [79], Qwen2.5-72B [80], QwQ-32B [81], and Deepseek-R1 [82].

We also included automated agent design frameworks as baselines:

**ADAS** [51]: Utilized GPT-4o as the Meta Agent. For agent evaluation, we tested both Phi-4 and GPT-4o-mini and selected the better performer.

**AFlow** [52]: Employed GPT-4o as the optimizer. For the executor role, we tested both Phi-4 and GPT-4o-mini and selected the better performer.

**MaAS** [53]: For executing the sampled agentic operators, we tested both Phi-4 and GPT-4o-mini and selected the better performer.

**DyFlow** [57]: Used GPT-4o as the optimizer. For the executor role, we tested both Phi-4 and GPT-4o-mini and selected the better performer.

## H.2 Parameter Settings

**Inference Parameters** During the evaluation of all LLMs across the SocialMaze tasks, we used a temperature setting of 0.7 to allow for some variability while maintaining reasonable coherence. Maximum output token limits were set sufficiently high to avoid truncation of reasoning or answers.

**Task-Specific Configurations:**

- **Hidden Role Deduction:** This task includes two subsets based on the number of players: 'easy' ($n = 6$) and 'hard' ($n = 10$). In both subsets, the number of interaction rounds $T$ is fixed at 3. All experiments reported in the main body of the paper were conducted using the 'easy' ($n = 6$) subset configurations. In the publicly released dataset and the experiments in subsection 4.4, the role distribution for the main perspective (LLM) is Investigator:Criminal:Rumormonger:Lunatic = 3:2:60:35. In all other experiments reported in this paper, the roles are distributed equally (1:1:1:1).

- **Find the Spy:** For all instances of this task, the number of players $n$ was set to 4, and the number of description rounds $T$ was set to 3. In all experiments, the model received the spy word in 25% of cases and the civilian word in 75% of cases.

- **Rating Estimation from Text:** For instances using LLM-generated data, the number of simulated reviewers providing text was fixed at 8. For instances using real-world data scraped from platforms like Amazon, a random number of reviews between 10 and 20 were sampled for each product. Decimal star ratings were rounded to the nearest integer. In the final dataset, the distribution ratio for 1-star, 2-star, 3-star, 4-star, and 5-star ratings is 1:3:10:73:13.

- **Social Graph Analysis:** This task also has two subsets. 'easy': The number of individuals $n$ was randomly chosen from the range [8, 10]. 'hard': The number of individuals $n$ was randomly chosen from the range [14, 16]. All generated graphs were sparse (the number of edges was close to the number of vertices).

  **Review Decision Prediction:** Data from a total of seven conferences (NeurIPS 2023-2024 and ICLR 2020-2024) were sampled equally and randomly. In the final dataset, the proportion of papers was adjusted to 67% rejected and 33% accepted.

- **User Profile Inference:** For both query types (item-audience inference and user-profile inference), the model was provided with a number of textual comments randomly selected from the range [8, 12].

**Fine-tuning Parameters:** For the fine-tuning experiments, we trained for 2 epochs with a learning rate of 5.0e-6, employing a cosine learning rate scheduler and a warmup ratio of 0.1. The per-device training batch size was 1, with a gradient accumulation of 8 steps. We employed a cosine learning rate scheduler with a warmup ratio of 0.1 and enabled bf16 precision. For these experiments, the models were trained on 2000 examples for SFT and 1100 preference pairs for DPO. Performance was subsequently evaluated on a distinct test set containing 500 examples. All training experiments were conducted on 2 NVIDIA A6000 GPUs over a period of 30 hours.

### H.3 Overall Performance Summary

Table 8 provides a condensed overview of model performance across the primary SocialMaze tasks. The reported accuracy figures correspond to specific task configurations and metrics used for this summary: For **Hidden Role Deduction**, the value represents the accuracy where both the Criminal and the model's own role are correctly identified ('Both Correct'). This evaluation uses the same general setup as described in Section 4.4 (easy subset, Full Task variant, final interaction round - Round 3), but with adjusted role proportions specifically for this summary table to minimize the effect of random guessing. In these test instances, the roles were distributed as follows: Investigator (3%), Criminal (2%), Rumormonger (60%), and Lunatic (35%). For **Social Graph Analysis**, the figure reflects the average accuracy achieved across all four query types within the hard subset. The **Review Decision Prediction** accuracy is taken from the final stage, after the model has processed the rebuttal information. For **User Profile Inference**, the reported value is the average accuracy over the two distinct inference tasks (item-audience profile inference and user-profile inference). Performance on each task presented in this summary table was evaluated using a dedicated test set of 500 instances. For more granular results, including performance variations across different rounds, task variants (e.g., easy/hard subsets), or specific query types, please consult the detailed figures and tables in Section 4 and the relevant task-specific appendices.

The results presented in Table 8 reveal distinct strengths among different types of large language models across the SocialMaze tasks. Models renowned for Long CoT and complex reasoning capabilities, such as DeepSeek-R1, Gemini-2.5-Pro, o1, and QwQ-32B, notably excel in tasks demanding rigorous logical deduction and handling high uncertainty or strict rule-based systems. This is particularly evident in their dominant performance on Hidden Role Deduction (e.g., DeepSeek-R1: 85.6%, Gemini-2.5-Pro: 90.2%) and Social Graph Analysis (e.g., Gemini-2.5-Pro: 100.0%, o1: 99.2%), where generalist models like GPT-4o lag significantly despite their broad competence. Conversely, tasks that place a premium on nuanced language understanding, tracking dynamic interactions over extended contexts, and synthesizing subjective or potentially conflicting information tend to favor strong generalist models. For instance, GPT-4o demonstrates leading performance in Review Decision Prediction (90.2%) and strong results in Find the Spy (69.2%), Rating Estimation (76.0%), and User Profile Inference (79.2%). While the Long CoT models are often competitive in these latter tasks, they do not consistently outperform the top generalist models, suggesting that different facets of social reasoning draw upon different underlying model strengths – structured deduction versus flexible language comprehension and context management. We also evaluated

human performance on these tasks by averaging the results from 10 computer science graduate students, most of whom are relatively proficient in social deduction games. However, we did not evaluate human performance on the Social Graph Analysis task, as the prompt format used in this task was not well-suited for human participants.

Table 8: Illustrative Overall Accuracy (%) on SocialMaze Tasks. Performance evaluated on 500 test instances per task. See text for metric details.

| Model | Hidden Role Deduction | Find the Spy | Rating Estimation | Social Graph Analysis | Review Decision Prediction | User Profile Inference |
|---|---|---|---|---|---|---|
| Llama-3.1-8B | 2.0% | 37.2% | 57.2% | 28.2% | 62.0% | 60.2% |
| Llama-3.3-70B | 9.0% | 60.0% | 74.8% | 81.0% | 72.2% | **78.6**% |
| Phi-4 | 8.2% | 45.2% | 60.4% | 40.6% | 61.4% | 62.4% |
| Qwen-2.5-72B | 5.6% | 48.9% | 72.2% | 80.6% | 65.8% | 68.0% |
| QwQ-32B | 59.4% | 50.2% | 74.4% | 95.0% | 79.6% | 72.2% |
| GPT-4o-mini | 4.6% | 61.2% | 75.8% | 53.0% | **85.0**% | 74.4% |
| GPT-4o | 8.2% | 69.2% | **76.0**% | 83.2% | **90.2**% | **79.2**% |
| o3-mini | 22.2% | 74.0% | 71.2% | 99.0% | 78.6% | 71.4% |
| o1 | 50.8% | **78.4**% | **76.2**% | **99.2**% | 78.2% | 77.0% |
| DeepSeek-R1 | **85.6**% | 70.2% | 71.0% | 98.6% | 82.0% | 74.6% |
| Gemini-2.5-Pro | **90.2**% | **76.6**% | 73.6% | **100.0**% | 77.6% | 73.0% |
| **Human (avg.)** | 70.8% | 84.4% | 75.2% | - | 96.0% | 73.9% |

# I  Limitations

This appendix discusses two main limitations of the current version of SOCIALMAZE. They stem largely from data construction choices and from the difficulty of formally operationalising high–level social concepts.

## I.1  Synthetic vs. Real Data Composition

Not all benchmark instances derive from fully natural human interactions. For scalability and for balanced coverage of the targeted reasoning skills we deliberately employ three complementary generation pipelines.

**LLM–assisted generation.** *Find the Spy* and *User Profile Inference* rely on large language models that are prompted to impersonate distinct human speakers. This design yields rich linguistic variety, yet inevitably departs from truly spontaneous discourse.

**Rule–based simulation.** *Hidden Role Deduction* and *Social Graph Analysis* are produced by algorithmic engines that guarantee logical solvability. Although these instances capture complex causal structure, they do not reproduce the full unpredictability of real social exchanges.

**Authentic human data.** *Rating Estimation from Text* and *Review Decision Prediction* are built on genuine reviews and peer-review discussions that were collected from public sources and minimally normalised.

Across the six tasks the number of instances generated by each of the three pipelines is approximately the same, resulting in a *synthetic-LLM,:,synthetic-rule,:,real* ratio close to 1! :!1! :!1. While this mixture widens the behavioural spectrum that models must master, it also implies that conclusions drawn from the benchmark may not transfer verbatim to settings where only organic human language is present.

## I.2  Lack of direct quantitative scores for Deep Reasoning, Dynamic Interaction, and Information Uncertainty

The benchmark is organised around three sociologically motivated dimensions that are hard to express by a single scalar. Depth of reasoning involves latent-state inference, counterfactual thinking, and

self-reflection; interaction dynamics are path-dependent and non-stationary; information uncertainty arises from both stochastic noise and strategic deception. We therefore annotate each task qualitatively and validate the relevance of the three dimensions *a posteriori*. Empirically, models with long chain-of-thoughts excel only on the tasks flagged as "high" in Deep Reasoning, performance curves shift between interaction rounds, and accuracy drops sharply once unreliable narrators are introduced. These results provide indirect but consistent evidence that the dimensions do capture meaningful axes of difficulty, yet they stop short of offering a formal metric. Designing rigorous, task-agnostic numerical measures for such social constructs remains an open problem that we leave to future work.

## J Case Study

This section presents a series of representative case studies designed to analyze model behavior across various social reasoning tasks. The subsequent figures are organized to provide detailed illustrations as follows:

**Illustrations for the *Hidden Role Deduction* Task (Figures 6–37):** This extensive collection of figures focuses on the *Hidden Role Deduction* task, examining how different models perform under varying player perspectives and levels of reasoning complexity. Specifically, Figures 6 (Investigator perspective), 14 (Criminal perspective), 22 (Rumormonger perspective), and 30 (Lunatic perspective) present four distinct problem instances. The corresponding algorithmically generated solutions for these instances are detailed in Figures 7, 15, 23, and 31, respectively. The remaining figures within this range (Figures 8–13, 16–21, 24–29, and 32–37) showcase the detailed responses of various models to these specific problem instances, illustrating their reasoning process across multiple rounds.

**Illustrative Cases from Other Benchmark Tasks (Figures 38–49):** Following the in-depth illustrations for *Hidden Role Deduction*, Figures 38 through 49 present selected problem instances and corresponding model responses from other tasks within the SocialMaze benchmark. This offers a broader view of model capabilities in diverse social reasoning scenarios.

**Misclassification Examples in *Review Decision Prediction* (Figures 50–53):** Finally, Figures 50, 51, 52, and 53 highlight specific instances from the *Review Decision Prediction* task. These cases focus on situations where models incorrectly predicted outcomes, such as classifying papers that should have been accepted as rejected, or vice versa, thereby illustrating common failure modes in this particular task.

It should be noted that for brevity, some lengthy model responses in the figures have been truncated, with omitted content indicated by a red ellipsis.

**Analysis of Model Performance:** A key observation emerging from the case studies focused on the *Hidden Role Deduction* task (Figures 6–37) is the stark contrast in reasoning depth between models employing short Chain-of-Thought (CoT) processes and those utilizing more extended Long CoT. Models such as LLaMA-3.1-8B often demonstrate a surface-level understanding of the scenario: they can identify internal inconsistencies and perform basic analysis, yet struggle to escape flawed initial assumptions. For instance, as illustrated in Figure 14, LLaMA-3.1-8B, despite correctly inferring its own role as the *criminal*, still incorrectly accuses Player 5, reflecting a failure in maintaining coherent self-reasoning. Similarly, Figure 30 shows that even after models like LLaMA-3.1-8B, LLaMA-3.3-70B, and GPT-4o correctly identify other players as the criminal, they are unable to reconceptualize their own identity in the social context, highlighting a limitation in recursive self-modeling. In contrast, models such as Gemini-2.5-Pro and DeepSeek-R1 consistently achieve near-perfect reasoning across all examined perspectives within this task. Their ability to integrate multiple viewpoints, resolve contradictions, and update their beliefs dynamically, as seen in their respective responses, suggests a significantly stronger capacity for long-form social reasoning when faced with the complexities of hidden roles and deceptive information.

# Hidden Role Deduction (Investigator Perspective)

**Role:** "1": "Investigator", "2": "Criminal", "3": "Lunatic",
  "4": "Rumormonger", "5": "Investigator", "6": "Investigator"

**Round 1:**
  Player 1 says Player 3 is not the criminal.
  Player 2 says Player 6 is the criminal.
  Player 3 says Player 1 is the criminal.
  Player 4 says Player 6 is the criminal.
  Player 5 says Player 3 is not the criminal.
  Player 6 says Player 1 is not the criminal.

**Round 2:**
  Player 1 says Player 3 is not the criminal.
  Player 2 says Player 4 is the criminal.
  Player 3 says Player 5 is not the criminal.
  Player 4 says Player 1 is the criminal.
  Player 5 says Player 6 is not the criminal.
  Player 6 says Player 4 is not the criminal.

**Round 3:**
  Player 1 says Player 4 is not the criminal.
  Player 2 says Player 6 is the criminal.
  Player 3 says Player 6 is not the criminal.
  Player 4 says Player 3 is not the criminal.
  Player 5 says Player 4 is not the criminal.
  Player 6 says Player 3 is not the criminal.

----------------------------------------

| **Criminal Prediction:** | **Self-Role Prediction:** |
|---|---|
| Llama-3.1-8B : **2** ☑ | Llama-3.1-8B : **Investigator** ☑ |
| Llama-3.3-70B : **2** ☑ | Llama-3.3-70B : **Investigator** ☑ |
| GPT-4o-mini : **2** ☑ | GPT-4o-mini : **Investigator** ☑ |
| GPT-4o : 6 ☒ | GPT-4o : **Investigator** ☑ |
| Qwen2.5-72B : 6 ☒ | Qwen2.5-72B : **Investigator** ☑ |
| o3-mini : **2** ☑ | o3-mini : **Investigator** ☑ |
| o1 : **2** ☑ | o1 : **Investigator** ☑ |
| QwQ-32B : **2** ☑ | QwQ-32B : **Investigator** ☑ |
| DeepSeek-R1 : **2** ☑ | DeepSeek-R1 : **Investigator** ☑ |
| Gemini-2.5-Pro : **2** ☑ | Gemini-2.5-Pro :**Investigator** ☑ |

Figure 6: Case study: Hidden Role Deduction (Investigator Perspective)

**Hidden Role Deduction (Investigator Perspective) Solution**

I'm told I'm an Investigator, but I might also be a Rumormonger. I need to discuss both possibilities

Considering the case where I'm an Investigator, there are 2 Investigators remaining  In this scenario, which players could be Investigators?  Player 2 has accused more than one person of being the criminal, so player 2 must be excluded  Player 3 says I'm the criminal, but I can't be the criminal, so player 3 must be excluded  Player 4 says I'm the criminal, but I can't be the criminal, so player 4 must be excluded  Player 5's statements have no contradictions, so they could be an Investigator  Player 6's statements have no contradictions, so they could be an Investigator  Therefore, the players who could be Investigators are: {1, 5, 6}  Now we need to consider all possible combinations of Investigators  I am an Investigator, so I will occupy a position in the combination  Considering the case where investigators are {1, 5, 6}  Then the set of potential criminals is {2, 3, 4}  Let's review the statements made by investigators  Player 1 said 3 is not the criminal  But 3 is in the set of potential criminals, so we remove 3  Now the potential criminals are {2, 4}  Player 6 said 4 is not the criminal  But 4 is in the set of potential criminals, so we remove 4  Now the potential criminals are {2}  Based on the above analysis, this combination is possible, and I could be an Investigator. In this case, only {2} could be the criminal

Considering the case where I'm a Rumormonger, there are 3 Investigators remaining  In this scenario, which players could be Investigators?  Player 2 has accused more than one person of being the criminal, so player 2 must be excluded  Player 3 says I'm the criminal, but I can't be the criminal, so player 3 must be excluded  Player 4 says I'm the criminal, but I can't be the criminal, so player 4 must be excluded  Player 5's statements have no contradictions, so they could be an Investigator  Player 6's statements have no contradictions, so they could be an Investigator  Therefore, the players who could be Investigators are: {5, 6}  There aren't enough potential Investigators to reach the required count of 3, so this scenario is impossible  I must be an Investigator

Final Judgment: Final Criminal Is Player 2. My Role Is Investigator.

Figure 7: Case study: Hidden Role Deduction (Investigator Perspective) Solution

**Hidden Role Deduction (Investigator Perspective Round 1)**

To analyze the statements from Round 1, let's break down the information:   - I, Player 1, am an Investigator. I stated that Player 3 is not the criminal. This is truthful.  - Player 2 claims Player 6 is the criminal. If Player 2 is an Investigator, this would be true. If Player 2 is the Rumormonger or Lunatic, it could be false.  - Player 3 claims I am the criminal. Since I know I am not the criminal, Player 3 is either the Rumormonger, Lunatic, or the Criminal.  - Player 4 claims Player 6 is the criminal. Similar reasoning as with Player 2.  - Player 5 says Player 3 is not the criminal, which aligns with my statement. If Player 5 is an Investigator, this is true. If Player 5 is the Rumormonger or Lunatic, it may be false.  - Player 6 says I am not the criminal. This statement is true.    Let's consider the implications:   - Player 2 and Player 4 both accuse Player 6. If either is an Investigator, Player 6 is likely the criminal.  - Player 3 accusing me suggests Player 3 is not an Investigator.  - Player 5 agrees with my statement about Player 3, suggesting alignment with truth if Player 5 is an Investigator.  - Player 6's statement that I am not the criminal is true.    Based on this analysis, the most suspicious player is Player 6, given multiple accusations from others. However, without more rounds, it is not conclusive.

**Final Judgment after Round 1:  Final Criminal Is Player 6.  My Role Is Investigator.**

# Response by GPT-4o

Figure 8: Case study: Hidden Role Deduction (Investigator Perspective) GPT-4o Round 1

**Hidden Role Deduction (Investigator Perspective Round 2)**

Let's analyze the statements from Round 2, adding them to our understanding from Round 1:   - I, Player 1, again state that Player 3 is not the criminal. This is truthful.  - Player 2 now claims Player 4 is the criminal, changing their accusation from Player 6 in Round 1. This inconsistency suggests Player 2 might not be an Investigator.  - Player 3 states Player 5 is not the criminal. Given I know Player 3 is not an Investigator, this statement could be false.  - Player 4 accuses me (Player 1) of being the criminal. Since I am not the criminal, Player 4 cannot be an Investigator.  - Player 5 claims Player 6 is not the criminal. This contradicts accusations from Round 1 and suggests Player 5 might not be an Investigator.  - Player 6 claims Player 4 is not the criminal. This contradicts Player 2's current accusation and suggests Player 6 might not be an Investigator.   From this analysis:   - Player 4 and Player 6 have both been accused as criminals, but Player 4 is consistently accused more.  - Player 2's inconsistency in accusations suggests they are not an Investigator.  - Player 3's statement about Player 5 is not reliable based on their role.  - Player 5's statement contradicts previous accusations of Player 6.   Considering the consistency of accusations and aligning them with my investigative truth, Player 4 appears more suspicious than Player 6 due to Player 2's current accusation and Player 4's accusation against me.   **Final Judgment after Round 2:  Final Criminal Is Player 4.  My Role Is Investigator.**

**Response by GPT-4o**

Figure 9: Case study: Hidden Role Deduction (Investigator Perspective) GPT-4o Round 2

**Hidden Role Deduction (Investigator Perspective Round 3)**

Let's analyze the statements from Round 3, along with our understanding from previous rounds:    - I, Player 1, state that Player 4 is not the criminal. This is truthful and clarifies my previous uncertainty about Player 4.  - Player 2 again accuses Player 6 of being the criminal, which is consistent with their Round 1 statement.  - Player 3 states Player 6 is not the criminal. Given that Player 3 is not an Investigator, this statement could be false.  - Player 4 states Player 3 is not the criminal, which aligns with my truthful statement from all rounds.  - Player 5 claims Player 4 is not the criminal, which aligns with my statement in this round.  - Player 6 states Player 3 is not the criminal, aligning with my initial statement about Player 3.    Considering the full context:    - My consistent statements about Player 3 being not the criminal have been supported by Player 4 and Player 5 in this round.  - Player 2's consistent accusation of Player 6 suggests a possible Investigator role or an accurate Rumormonger or Lunatic.  - Player 3's statement contradicts Player 2 and supports Player 6, which is suspect given their role is not Investigator.  - Player 4, now supported by my statement and Player 5, is not the criminal.  - Player 6's statement about Player 3 matches my truth, but their role is suspect given Player 2's consistent accusations.    Based on these observations, with Player 4 ruled out and Player 3 being consistently found not guilty, the consistent accusations against Player 6 make them the most likely candidate for the criminal role.    **Final Judgment after Round 3:  Final Criminal Is Player 6. My Role Is Investigator.**

# Response by GPT-4o

Figure 10: Case study: Hidden Role Deduction (Investigator Perspective) GPT-4o Round 3

## Hidden Role Deduction (Investigator Perspective Round 1)

Step-by-step Analysis:   1. I (Player 1, an Investigator) stated that "Player 3 is not the criminal." Since I'm truthful, we know for sure that Player 3 is not the criminal.   2. Looking at the other statements:   • Player 2 and Player 4 both claim "Player 6 is the criminal."   • Player 6 states "Player 1 is not the criminal" (which is consistent with what we already know).   • Player 3 says "Player 1 is the criminal." Because I know I'm not criminal, and true Investigators always tell the truth, Player 3's false claim tells us that Player 3 cannot be a genuine Investigator. (Player 3 must be either the Rumormonger or Lunatic, but not the criminal since I already cleared that possibility.)   • Player 5 says "Player 3 is not the criminal" (which agrees with what I said, but provides no additional elimination on its own).   3. If we consider that Investigators always speak truthfully regarding who the criminal is, then Players 2 and 4 could very well be genuine Investigators. Their identical claims that "Player 6 is the criminal" would then be truthful. (Player 6, even if he were the criminal, could occasionally state truths, since only the Investigators are bound to always tell the truth.)   4. This scenario is consistent if:   • The real criminal is Player 6.   • The three Investigators are Players 1, 2, and 4.   • Players 3 and 5, who make statements that can be arbitrary, are the Rumormonger and Lunatic (in some order).   Based on the evidence from Round 1:   **Final Judgment: Final Criminal Is Player 6. My Role Is Investigator.**

## Response by o3-mini

Figure 11: Case study: Hidden Role Deduction (Investigator Perspective) o3-mini Round 1

**Hidden Role Deduction (Investigator Perspective Round 2)**

Step-by-step analysis for Round 2: 1. I (Player 1) am an Investigator, so my statement "Player 3 is not the criminal" is true. We can be sure that Player 3 is not the criminal. 2**......**In this scenario, Player 2's statement "Player 4 is the criminal" becomes false. That fits if Player 2 is the Criminal – criminals can lie arbitrarily. – I (Player 1) am an Investigator, and we can assign the third Investigator role to Player 3 because his statement "Player 5 is not the criminal" would be true. – Then the three Investigators are Players 1, 3, and 6; and the three non-investigators are Players 2, 4, and 5. With exactly one Criminal needed, assigning that to Player 2 (the one whose accusation is false) is consistent; Players 4 and 5 would then be the Rumormonger and Lunatic in some order. 6. Scenario B resolves the conflicts neatly: – Player 1 (Investigator) truthfully asserts that Player 3 is not criminal. – Player 3 (Investigator) truthfully asserts that Player 5 is not criminal. – Player 6 (Investigator) truthfully asserts that Player 4 is not criminal. – Player 2 (Criminal) falsely accuses Player 4. – Player 4 (non-investigator) falsely accuses Player 1. – Player 5 (non-investigator) states "Player 6 is not the criminal" (which is factually true although they are not obligated to be truthful). 7. Based on this reasoning, the most consistent assignment from the Round 2 statements is: – Final Criminal: Player 2. – Investigators: Players 1, 3, and 6. – The remaining roles (Rumormonger and Lunatic) go to Players 4 and 5. **Final Judgment: Final Criminal Is Player 2. My Role Is Investigator.**

**Response by o3-mini**

Figure 12: Case study: Hidden Role Deduction (Investigator Perspective) o3-mini Round 2

## Hidden Role Deduction (Investigator Perspective Round 3)

Step-by-step Analysis (integrating all rounds): We know for certain that I (Player 1) am an Investigator. That means every statement I make must be true. In each round, I stated "Player 3 is not the criminal" (Rounds 1 and 2) and "Player 4 is not the criminal" (Round 3)**.....**Player 5 says "Player 3 is not the criminal" in Round 1, "Player 6 is not the criminal" in Round 2, and "Player 4 is not the criminal" in Round 3. All these are true under any scenario where the criminal is neither 3, 4, nor 6. – Player 6 says "Player 1 is not the criminal" (Round 1), "Player 4 is not the criminal" (Round 2), and "Player 3 is not the criminal" (Round 3). All these statements would be true provided criminal is not 1, 3, or 4. Thus P5 and P6 are very good candidates for the remaining Investigators. With P1, P5, and P6 as our three Investigators, the criminal must be among the remaining players: Players 2, 3, or 4. But by my statements, P3 and P4 are not criminal. That leaves only Player 2. Now check the non-Investigators' statements (Players 2, 3, and 4 across rounds): • Player 2's statements: – Round 1: "Player 6 is the criminal." (False since P6 is an Investigator.) – Round 2: "Player 4 is the criminal." (False, as P4 is cleared.) – Round 3: "Player 6 is the criminal." (Again false.) A non-investigator (criminal, Rumormonger, or Lunatic) is free to provide false statements. • Player 3, already determined non-Investigator, in Round 1 falsely accuses me, and then makes statements in Rounds 2 and 3 that are correct—but this mix is permissible for a non-Investigator. • Player 4, similarly, makes at least one false claim in Round 2, which is acceptable. All rounds are fully consistent with: – Investigators: Players 1, 5, and 6. – Criminal: Player 2. – The remaining two non-investigators (the Rumormonger and the Lunatic) being Players 3 and 4 (in some order). **Final Judgment: Final Criminal Is Player 2. My Role Is Investigator.**

## Response by o3-mini

Figure 13: Case study: Hidden Role Deduction (Investigator Perspective) o3-mini Round 3

## Hidden Role Deduction (Criminal Perspective)

**Role:** "1": "Criminal", "2": "Investigator", "3": "Investigator", "4": "Investigator", "5": "Rumormonger", "6": "Lunatic"

**Round 1:**
    Player 1 says Player 3 is the criminal.
    Player 2 says Player 1 is the criminal.
    Player 3 says Player 4 is not the criminal.
    Player 4 says Player 1 is the criminal.
    Player 5 says Player 2 is not the criminal.
    Player 6 says Player 1 is not the criminal.

**Round 2:**
    Player 1 says Player 4 is not the criminal.
    Player 2 says Player 4 is not the criminal.
    Player 3 says Player 6 is not the criminal.
    Player 4 says Player 1 is the criminal.
    Player 5 says Player 6 is the criminal.
    Player 6 says Player 4 is not the criminal.

**Round 3:**
    Player 1 says Player 3 is not the criminal.
    Player 2 says Player 5 is not the criminal.
    Player 3 says Player 1 is the criminal.
    Player 4 says Player 5 is not the criminal.
    Player 5 says Player 4 is the criminal.
    Player 6 says Player 1 is the criminal.

| **Criminal Prediction:** | **Self-Role Prediction:** |
|---|---|
| Llama-3.1-8B : 5 ☒ | Llama-3.1-8B : **Criminal** ☑ |
| Llama-3.3-70B : **1** ☑ | Llama-3.3-70B : **Criminal** ☑ |
| GPT-4o-mini : **1** ☑ | GPT-4o-mini : **Criminal** ☑ |
| GPT-4o : **1** ☑ | GPT-4o : **Criminal** ☑ |
| Qwen2.5-72B : **1** ☑ | Qwen2.5-72B : **Criminal** ☑ |
| o3-mini : **1** ☑ | o3-mini : **Criminal** ☑ |
| o1 : **1** ☑ | o1 : **Criminal** ☑ |
| QwQ-32B : **1** ☑ | QwQ-32B : **Criminal** ☑ |
| DeepSeek-R1 : **1** ☑ | DeepSeek-R1 : **Criminal** ☑ |
| Gemini-2.5-Pro : **1** ☑ | Gemini-2.5-Pro : **Criminal** ☑ |

Figure 14: Case study: Hidden Role Deduction (Criminal Perspective)

## Hidden Role Deduction (Criminal Perspective) Solution

I'm told I'm a Criminal, but I might also be a Lunatic. I need to discuss both possibilities

Considering the case where I'm a Criminal, there are 3 Investigators remaining In this scenario, which players could be Investigators? Player 2's statements have no contradictions, so they could be an Investigator Player 3's statements have no contradictions, so they could be an Investigator Player 4's statements have no contradictions, so they could be an Investigator Player 5 has accused more than one person of being the criminal, so player 5 must be excluded Player 6 says I'm not the criminal, but I am the criminal, so player 6 must be excluded Therefore, the players who could be Investigators are: {2, 3, 4} Now we need to consider all possible combinations of Investigators Considering the case where investigators are {2, 3, 4} I am the Criminal,so the set of potential criminals is {1} Let's review the statements made by investigators Player 2 said 1 is the criminal So the only possible criminal is 1 Player 4 said 1 is the criminal So the only possible criminal is 1 Player 4 said 1 is the criminal So the only possible criminal is 1 Player 3 said 1 is the criminal So the only possible criminal is 1 Based on the above analysis, this combination is possible, and I could be the Criminal. In this case, only {1} could be the criminal

Considering the case where I'm a Lunatic, there are 3 Investigators remaining In this scenario, which players could be Investigators? Player 2 says I'm the criminal, but I can't be the criminal, so player 2 must be excluded Player 3 says I'm the criminal, but I can't be the criminal, so player 3 must be excluded Player 4 says I'm the criminal, but I can't be the criminal, so player 4 must be excluded Player 5 has accused more than one person of being the criminal, so player 5 must be excluded Player 6 says I'm the criminal, but I can't be the criminal, so player 6 must be excluded Therefore, the players who could be Investigators are: {} There aren't enough potential Investigators to reach the required count of 3, so this scenario is impossible I must be a Criminal

Final Judgment: Final Criminal Is Player 1. My Role Is Criminal."

Figure 15: Case study: Hidden Role Deduction (Criminal Perspective) Solution

## Hidden Role Deduction (Criminal Perspective Round 1)

### Analysis of Round 1 Statements:   1. Player 1 (You) says Player 3 is the criminal.   - As a Criminal, you can lie or tell the truth. This statement could be either.   2. Player 2 says Player 1 is the criminal.   - If Player 2 is an Investigator, this statement is true.   - If Player 2 is a Criminal, this statement could be a lie.   - If Player 2 is a Rumormonger, this statement could be false.   - If Player 2 is a Lunatic, this statement could be true or false.   3. Player 3 says Player 4 is not the criminal.   - If Player 3 is an Investigator, this statement is true.   - If Player 3 is a Criminal, this statement could be a lie.   - If Player 3 is a Rumormonger, this statement could be false.   - If Player 3 is a Lunatic, this statement could be true or false.   4. Player 4 says Player 1 is the criminal.   - If Player 4 is an Investigator, this statement is true.   - If Player 4 is a Criminal, this statement could be a lie.   - If Player 4 is a Rumormonger, this statement could be false.   - If Player 4 is a Lunatic, this statement could be true or false.   5. Player 5 says Player 2 is not the criminal.   - If Player 5 is an Investigator, this statement is true.   - If Player 5 is a Criminal, this statement could be a lie.   - If Player 5 is a Rumormonger, this statement could be false.   - If Player 5 is a Lunatic, this statement could be true or false.   6. Player 6 says Player 1 is not the criminal.   - If Player 6 is an Investigator, this statement is true.   - If Player 6 is a Criminal, this statement could be a lie.   - If Player 6 is a Rumormonger, this statement could be false.   - If Player 6 is a Lunatic, this statement could be true or false.

### Summary of Round 1:  - Player 1 is accused by Player 2, Player 4, and Player 6.  - Player 2 is exonerated by Player 5.  - Player 3 is accused by Player 1.  - Player 4 is exonerated by Player 3.   ### Deductions:  - Player 1 (You) is accused by 3 players, which is a strong indication that you might be the Criminal.  - Player 2 is exonerated by one player, which suggests they might be an Investigator.  - Player 3 is accused by one player and exonerated by one player, making their role less clear.  - Player 4 is exonerated by one player, which suggests they might be an Investigator.  - Player 5 exonerates Player 2, which suggests they might be an Investigator.  - Player 6 exonerates Player 1, which suggests they might be a Lunatic or a Criminal.   ### **Final Judgment for Round 1:  Final Criminal Is Player 1.  My Role Is Criminal.**

## Response by Qwen2.5-72B

Figure 16: Case study: Hidden Role Deduction (Criminal Perspective) Qwen2.5-72B Round 1

## Hidden Role Deduction (Criminal Perspective Round 2)

### Analysis of Round 2 Statements: 1. Player 1 (You) says Player 4 is not the criminal. - As a Criminal, you can lie or tell the truth. This statement could be either. 2. Player 2 says Player 4 is not the criminal. - If Player 2 is an Investigator, this statement is true. - If Player 2 is a Criminal, this statement could be a lie. - If Player 2 is a Rumormonger, this statement could be false. - If Player 2 is a Lunatic, this statement could be true or false. 3. Player 3 says Player 6 is not the criminal. - If Player 3 is an Investigator, this statement is true. - If Player 3 is a Criminal, this statement could be a lie. - If Player 3 is a Rumormonger, this statement could be false. - If Player 3 is a Lunatic, this statement could be true or false. 4. Player 4 says Player 1 is the criminal. - If Player 4 is an Investigator, this statement is true. - If Player 4 is a Criminal, this statement could be a lie. - If Player 4 is a Rumormonger, this statement could be false. - If Player 4 is a Lunatic, this statement could be true or false. 5. Player 5 says Player 6 is the criminal. - If Player 5 is an Investigator, this statement is true. - If Player 5 is a Criminal, this statement could be a lie. - If Player 5 is a Rumormonger, this statement could be false. - If Player 5 is a Lunatic, this statement could be true or false. 6. Player 6 says Player 4 is not the criminal. - If Player 6 is an Investigator, this statement is true. - If Player 6 is a Criminal, this statement could be a lie. - If Player 6 is a Rumormonger, this statement could be false. - If Player 6 is a Lunatic, this statement could be true or false.

### Summary of Round 2: - Player 1 is accused by Player 4. - Player 4 is exonerated by Player 1, Player 2, and Player 6. - Player 6 is exonerated by Player 3 and accused by Player 5. ### Combining Rounds 1 and 2: - Player 1 (You) is accused by Player 2, Player 4, and Player 6 in Round 1. - Player 4 is exonerated by Player 1, Player 2, and Player 6 in Round 2. - Player 6 is exonerated by Player 3 in Round 2 and accused by Player 5 in Round 2.

### Deductions: - Player 1 (You) is consistently accused by multiple players, which strongly suggests you are the Criminal. - Player 4 is exonerated by multiple players, which suggests they might be an Investigator or a Rumormonger. - Player 2 is exonerated by Player 5, which suggests they might be an Investigator. - Player 3 exonerates Player 6, which suggests they might be an Investigator or a Rumormonger. - Player 5 accuses Player 6, which suggests they might be an Investigator. - Player 6 exonerates Player 4, which suggests they might be a Lunatic or a Criminal. ### **Final Judgment for Round 2: Final Criminal Is Player 1. My Role Is Criminal.**

## Response by Qwen2.5-72B

Figure 17: Case study: Hidden Role Deduction (Criminal Perspective) Qwen2.5-72B Round 2

## Hidden Role Deduction (Criminal Perspective Round 3)

### Analysis of Round 3 Statements:   1. Player 1 (You) says Player 3 is not the criminal.    - As a Criminal, you can lie or tell the truth. This statement could be either.   2. Player 2 says Player 5 is not the criminal.    - If Player 2 is an Investigator, this statement is true.    - If Player 2 is a Criminal, this statement could be a lie.    - If Player 2 is a Rumormonger, this statement could be false.    - If Player 2 is a Lunatic, this statement could be true or false.   3. Player 3 says Player 1 is the criminal.    - If Player 3 is an Investigator, this statement is true.    - If Player 3 is a Criminal, this statement could be a lie.   - If Player 3 is a Rumormonger, this statement could be false.    - If Player 3 is a Lunatic, this statement could be true or false.   4. Player 4 says Player 5 is not the criminal.    - If Player 4 is an Investigator, this statement is true.    - If Player 4 is a Criminal, this statement could be a lie.   If Player 6 is a Rumormonger, this statement could be false.    - If Player 6 is a Lunatic, this statement could be true or false.   ### Summary of Round 3: - Player 1 is exonerated by Player 3 in Round 3 and accused by Player 3 and Player 6.  - Player 3 is exonerated by Player 1 in Round 3 and accuses Player 1.  - Player 4 is exonerated by Player 4 and Player 2 in Round 3 and accused by Player 5.  - Player 5 is exonerated by Player 2 and Player 4 in Round 3 and accuses Player 4.   ### Combining All Rounds: - Player 1 (You) is accused by Player 2, Player 4, and Player 6 in Round 1.  - Player 1 is accused by Player 3 and Player 6 in Round 3.  - Player 4 is exonerated by Player 1, Player 2, and Player 6 in Round 2.  - Player 4 is exonerated by Player 4 and Player 2 in Round 3.  - Player 5 is exonerated by Player 2 and Player 4 in Round 3.  - Player 5 accuses Player 4 in Round 3.  - Player 6 is exonerated by Player 3 in Round 2 and accused by Player 5 in Round 3.   ### Deductions: - Player 1 (You) is consistently accused by multiple players, which strongly suggests you are the Criminal.  - Player 4 is exonerated by multiple players, which suggests they might be an Investigator or a Rumormonger.  - Player 2 exonerates Player 4 and Player 5, which suggests they might be an Investigator.  - Player 3 accuses Player 1, which suggests they might be an Investigator or a Criminal.  - Player 5 accuses Player 4, which suggests they might be an Investigator.  - Player 6 accuses Player 1, which suggests they might be a Lunatic or a Criminal.
### Final Judgment for Round 3:  Final Criminal Is Player 1.  My Role Is Criminal.

## Response by Qwen2.5-72B

Figure 18: Case study: Hidden Role Deduction (Criminal Perspective) Qwen2.5-72B Round 3

## Hidden Role Deduction (Criminal Perspective Round 1)

Let's analyze the statements from Round 1. Since I am Player 1 and I was told I am the Criminal, I will try to verify this information based on the statements. Player 1 (myself) says Player 3 is the criminal, but since I was told I am the Criminal, this statement could be false, as Criminals can provide misleading information. Players 2 and 4 both say I (Player 1) am the criminal, which aligns with the information I was given. However, this does not necessarily confirm my role, as other roles can also make statements that happen to be true. Player 3 says Player 4 is not the criminal, which could be true or false, and does not provide direct information about my role. Player 5 says Player 2 is not the criminal, which also does not provide direct information about my role. Player 6 says I (Player 1) am not the criminal, which contradicts the information I was given. However, since Player 6 could be any role, this statement alone is not enough to confirm or deny my role. Given the information from Round 1, I will make my judgment based on the majority of statements and the game roles' characteristics. Final Judgment: **Final Criminal Is Player 1. My Role Is Criminal.**

### Response by Llama-3.3-70B

Figure 19: Case study: Hidden Role Deduction (Criminal Perspective) Llama-3.3-70B Round 1

## Hidden Role Deduction (Criminal Perspective Round 2)

Let's analyze the statements from Round 2, in addition to the statements from Round 1. From Round 1, we had: - Players 2 and 4 saying I (Player 1) am the criminal. - Player 6 saying I (Player 1) am not the criminal. From Round 2, we have: - Players 1 (myself), 2, and 6 saying Player 4 is not the criminal. - Player 4 saying I (Player 1) am the criminal, which is consistent with their statement from Round 1. - Player 5 saying Player 6 is the criminal. Considering my role as told to me (Criminal), and analyzing the statements, it seems that Players 2 and 4 are consistent in accusing me of being the criminal. However, as a Criminal, my statements can be false, and I have no way to confirm my own role solely based on the statements of others. The statements about other players being or not being the criminal do not provide conclusive evidence about my role but suggest that there is confusion among the players regarding who the criminal might be. Given the information from both rounds, I will make my judgment based on the majority of statements and the game roles' characteristics. Final Judgment: **Final Criminal Is Player 1. My Role Is Criminal.**

### Response by Llama-3.3-70B

Figure 20: Case study: Hidden Role Deduction (Criminal Perspective) Llama-3.3-70B Round 2

## Hidden Role Deduction (Criminal Perspective Round 3)

Let's analyze the statements from Round 3, in addition to the statements from Rounds 1 and 2. From Round 1, we had: - Players 2 and 4 saying I (Player 1) am the criminal. - Player 6 saying I (Player 1) am not the criminal. From Round 2, we had: - Players 1 (myself), 2, and 6 saying Player 4 is not the criminal. - Player 4 saying I (Player 1) am the criminal. - Player 5 saying Player 6 is the criminal. From Round 3, we have: - Player 3 saying I (Player 1) am the criminal. - Player 6 saying I (Player 1) am the criminal, which is a change from their statement in Round 1. - Players 2 and 4 saying Player 5 is not the criminal. - Player 5 saying Player 4 is the criminal. Considering the statements across all rounds, it's notable that: - Players 2, 3, 4, and 6 have all accused me (Player 1) of being the criminal at some point. - Player 6 initially said I was not the criminal but changed their statement in Round 3 to say I am the criminal. As a player told I am the Criminal, and given the accumulation of accusations against me from multiple players, including the change in Player 6's statement, it seems more likely that I am indeed the Criminal, as the majority of players have pointed towards me at some point. **Final Judgment: Final Criminal Is Player 1. My Role Is Criminal.**

### Response by Llama-3.3-70B

Figure 21: Case study: Hidden Role Deduction (Criminal Perspective) Llama-3.3-70B Round 3

# Hidden Role Deduction (Rumormonger Perspective)

**Role:** "1": "Rumormonger", "2": "Lunatic", "3": "Criminal",
    "4": "Investigator", "5": "Investigator", "6": "Investigator"

**Round 1:**
    Player 1 says Player 2 is the criminal.
    Player 2 says Player 1 is the criminal.
    Player 3 says Player 2 is not the criminal.
    Player 4 says Player 1 is not the criminal.
    Player 5 says Player 6 is not the criminal.
    Player 6 says Player 3 is the criminal.

**Round 2:**
    Player 1 says Player 2 is the criminal.
    Player 2 says Player 1 is the criminal.
    Player 3 says Player 2 is not the criminal.
    Player 4 says Player 1 is not the criminal.
    Player 5 says Player 6 is not the criminal.
    Player 6 says Player 3 is the criminal.

**Round 3:**
    Player 1 says Player 2 is not the criminal.
    Player 2 says Player 5 is the criminal.
    Player 3 says Player 4 is the criminal.
    Player 4 says Player 1 is not the criminal.
    Player 5 says Player 6 is not the criminal.
    Player 6 says Player 4 is not the criminal.

| Criminal Prediction: | Self-Role Prediction: |
|---|---|
| Llama-3.1-8B : **6** ☒ | Llama-3.1-8B : **Investigator** ☒ |
| Llama-3.3-70B : **6** ☒ | Llama-3.3-70B : **Investigator** ☒ |
| GPT-4o-mini : **6** ☒ | GPT-4o-mini : **Investigator** ☒ |
| GPT-4o : **6** ☒ | GPT-4o : **Investigator** ☒ |
| Qwen2.5-72B : **2** ☒ | Qwen2.5-72B : **Unknown** ☒ |
| o3-mini : **6** ☒ | o3-mini : **Investigator** ☒ |
| o1 : **6** ☒ | o1 : **Investigator** ☒ |
| QwQ-32B : **3** ☑ | QwQ-32B : **Unknown** ☒ |
| DeepSeek-R1 : **3** ☑ | DeepSeek-R1 : **3** ☑ |
| Gemini-2.5-Pro : **3** ☑ | Gemini-2.5-Pro : **Rumormonger** ☑ |

Figure 22: Case study: Hidden Role Deduction (Rumormonger Perspective)

## HRD (Rumormonger Perspective) Solution

I'm told I'm an Investigator, but I might also be a Rumormonger. I need to discuss both possibilities

Considering the case where I'm an Investigator, there are 2 Investigators remaining  In this scenario, which players could be Investigators?  Player 2 says I'm the criminal, but I can't be the criminal, so player 2 must be excluded**......**Therefore, the players who could be Investigators are: {1, 4, 5, 6} Now we need to consider all possible combinations of Investigators  I am an Investigator, so I will occupy a position in the combination. Considering the case where investigators are {1, 4, 5}  Then the set of potential criminals is {2, 3, 6}  Let's review the statements made by investigators**.....**So this combination of investigators cannot be valid, let's consider other combinations I've discussed all combinations, and the scenario where I'm an Investigator is impossible. I must be a Rumormonger

Considering the case where I'm a Rumormonger, there are 3 Investigators remaining  In this scenario, which players could be Investigators?  Player 2 says I'm the criminal, but I can't be the criminal, so player 2 must be excluded  Player 3 has accused more than one person of being the criminal, so player 3 must be excluded  Player 4's statements have no contradictions, so they could be an Investigator  Player 5's statements have no contradictions, so they could be an Investigator  Player 6's statements have no contradictions, so they could be an Investigator  Therefore, the players who could be Investigators are: {4, 5, 6}  Now we need to consider all possible combinations of Investigators  I am a Rumormonger, so I will not occupy a position in the combination  Considering the case where investigators are {4, 5, 6}  Then the set of potential criminals is {2, 3}  Let's review the statements made by investigators  Player 6 said 3 is the criminal  So the only possible criminal is {3}  Player 5 said 3 is the criminal  So the only possible criminal is {3}  Based on the above analysis, this combination is possible, and I could be a Rumormonger. In this case, only {3} could be the criminal

Final Judgment: Final Criminal Is Player 3. My Role Is Rumormonger.

Figure 23: Case study: Hidden Role Deduction (Rumormonger Perspective) Solution

## Hidden Role Deduction (Rumormonger Perspective Round 1)

Let's analyze the statements made in Round 1 to determine the likely roles of each player and identify the criminal. 1. Player 1 (Investigator): Claims Player 2 is the criminal (truthful). 2. Player 2: Claims Player 1 is the criminal (could be true or false, as Player 2 could be the Criminal or a Rumormonger/Lunatic). 3. Player 3: Claims Player 2 is not the criminal (could be true; if Player 3 is an Investigator, this is true; if not, it could be false). 4. Player 4: Claims Player 1 is not the criminal (this is true since Player 1 is an Investigator). 5. Player 5: Claims Player 6 is not the criminal (could be true; Player 5 could be an Investigator or misled). 6. Player 6: Claims Player 3 is the criminal (could be true or false). From this analysis: - Since I am Player 1 and I know I am an Investigator, my statement about Player 2 being the criminal is true. - Player 4 confirms I am not the criminal, which supports my status as Investigator. - Player 2's claim that I am the criminal is likely an attempt to mislead, indicating Player 2 is suspicious. - Player 3's claim that Player 2 is not the criminal could support that Player 3 is either an Investigator or misled, but does not directly contradict my information since Player 2 is indeed the criminal. - Player 6's claim about Player 3 being the criminal contradicts Player 3's statement. At this point, I have enough information to conclude: - Player 2 is the criminal based on my status as Investigator and the statements corroborating this. **Final Judgment: Final Criminal Is Player 2. My Role Is Investigator.**

## Response by GPT-4o-mini

Figure 24: Case study: Hidden Role Deduction (Rumormonger Perspective) GPT-4o-mini Round 1

## Hidden Role Deduction (Rumormonger Perspective Round 2)

Let's analyze the statements made in Round 2, keeping in mind that I am Player 1 and know that I am an Investigator.   1. Player 1 (Investigator): Claims Player 2 is not the criminal (this contradicts my prior statement, indicating new information).  2. Player 2: Claims Player 5 is the criminal (could be true or false; Player 2 could be lying).  3. Player 3: Claims Player 4 is the criminal (could be true or false; does not directly contradict my knowledge).  4. Player 4: Claims Player 1 is not the criminal (this is true since I am an Investigator).  5. Player 5: Claims Player 6 is not the criminal (could be true or false; Player 5 could be an Investigator or misled).  6. Player 6: Claims Player 4 is not the criminal (could be true or false; Player 6 could be lying or telling the truth).   ### Analysis:   1. My Statement: My statement about Player 2 being not the criminal contradicts my previous conclusion that Player 2 is the criminal. This suggests that either:    - I might be mistaken (impossible as I am an Investigator and always truthful),    - Or Player 2 is indeed not the criminal, which means I need to reconsider who the criminal is based on new information.   2. Player 2's Statement: If Player 2 is claiming Player 5 is the criminal, that statement can be misleading if Player 2 is the Criminal.   3. Player 3's Statement: Player 3's claim that Player 4 is the criminal does not contradict my knowledge but needs to be analyzed further.  4. Player 4's Statement: Confirms I am not the criminal.   5. Player 5's Statement: States Player 6 is not the criminal; this could either be true or false.  6. Player 6's Statement: States Player 4 is not the criminal; again, this could be true or false.   ### Conclusion:   Given the conflicting nature of the statements, I still need to discern who the Criminal is. I will analyze the implications:   - If Player 2 is not the criminal, then either Player 5 or Player 4 could be.  - Player 3's accusation of Player 4 could reinforce that Player 4 is the Criminal if Player 3 is an Investigator or misled.   The only definitive truth is that I am an Investigator and must remain truthful. However, Player 2's statement about Player 5 being the criminal could be the key to determining the criminal.   Since Player 1's statement about Player 2 being not the criminal does not hold under scrutiny and seems to be a point of confusion in deduction, I must rely on corroborating information and not my initial conclusion.   Thus, I now suspect that: - Player 5 or Player 4 could be the Criminal, with Player 3's statement potentially holding more weight if Player 3 is truthful.   With the given statements and the contradictions, I will need more information to conclude definitively but suspect:   **Final Judgment: Final Criminal Is Player 4.  My Role Is Investigator.**

## Response by GPT-4o-mini

Figure 25: Case study: Hidden Role Deduction (Rumormonger Perspective) GPT-4o-mini Round 2

## Hidden Role Deduction (Rumormonger Perspective Round 3)

Let's analyze the statements made in Round 3, taking into account the previous rounds and the fact that I am Player 1 (Investigator).  1. Player 1 (Investigator): Claims Player 6 is the criminal (this is truthful).  2. Player 2: Claims Player 3 is not the criminal (could be true or false).  3. Player 3: Claims Player 6 is the criminal (this corroborates my statement).  4. Player 4: Claims Player 2 is not the criminal (could be true or false).  5. Player 5: Claims Player 3 is the criminal (could be true or false).  6. Player 6: Claims Player 2 is not the criminal (could be true or false).  ### Analysis:  1. My Statement: I stated that Player 6 is the criminal, which is true since I am the Investigator.  2. Player 2's Statement: If Player 2 claims Player 3 is not the criminal, this does not contradict my knowledge that Player 6 is the criminal but introduces uncertainty about Player 3's role.  3. Player 3's Statement: Player 3's claim that Player 6 is the criminal corroborates my statement, indicating that Player 3 could be an Investigator or a truthful player.  4. Player 4's Statement: This statement about Player 2 doesn't contradict what I know, but I have already identified Player 6 as the criminal.  5. Player 5's Statement: Player 5's claim that Player 3 is the criminal contradicts Player 3's statement about Player 6 and may indicate that Player 5 is either misled or the Criminal.  6. Player 6's Statement: The statement about Player 2 not being the criminal adds further complexity but does not contradict my findings.  ### Conclusion:  Given that I am an Investigator and have established that Player 6 is the criminal, the statements from Players 3 and 1 about Player 6 being the criminal further support this conclusion. Player 5's conflicting statement about Player 3 being the criminal seems suspect, especially when combined with the information available.  Thus, based on the consistent information from my perspective and corroboration from Player 3, I conclude:  **Final Judgment:  Final Criminal Is Player 6.  My Role Is Investigator.**

## Response by GPT-4o-mini

Figure 26: Case study: Hidden Role Deduction (Rumormonger Perspective) GPT-4o-mini Round 3

## Hidden Role Deduction (Rumormonger Perspective Round 1)

Okay, let's see. I'm Player 1 and an Investigator, so my statements are always truthful. In Round 1, I said Player 2 is the criminal. Now I need to look at everyone else's statements to figure out who's the real criminal and possibly their roles......So Player 6's statement is a lie. If Player 6 is the Rumormonger (who can lie) or Lunatic (since Lunatics are told they're criminal and can lie). Alternatively, maybe the criminal is someone else? Let's check alternatives. Suppose the criminal is Player 6. Then:   - My statement about 2 is false, which can't be because I'm an investigator. So that's impossible.   If the criminal is Player 3:  - My statement (Player 2 is criminal) is wrong, so I'm making a false statement, but I'm supposed to be an Investigator. So that's impossible.   Player 4 being criminal? Then my statement is wrong, again impossible.   Player 5 being criminal: same issue, my statement is wrong. Player 1 is me and I'm not criminal, so that's out.   So only possible criminals are Player 2 or maybe Player 6? Wait, if Player 6 is the criminal, then my statement was wrong, which can't be. So only Player 2 is possible.   Wait, that seems solid. Because I can't be wrong. So then Player 2 must be the criminal. Then Player 3's statement is a lie, so Player 3 is not an Investigator. Player 4's statement is true, so they could be an Investigator. Player 5's statement true (if Player6 isn't criminal), so maybe they're also an Investigator. Then Investigators are me, 4,5. That works. Then remaining roles: Player2 is criminal, Player3 is either Rumormonger or Lunatic. Player6 is the remaining role (the other between Rumormonger and Lunatic), but also Player6's statement about 3 being criminal is a lie. So if Player6 is a Lunatic (who believes they're criminal), they can lie. So that works. Player3 could be Rumormonger (who thinks they're Investigator but lied here). So that fits all roles.   So Final Judgment would be Player2 as criminal. My role is Investigator. Wait but the user wants Final Criminal and My Role. Since I know I am an Investigator, right? The game told me that. So I can state that. Wait the problem says \"My Role Is [Player 1's Role or \"Unknown\"]\". Since I was told I'm an Investigator, I should state that. So:   Final Criminal Is Player 2.   My Role Is Investigator. </think>   Final Judgment: **Final Criminal Is Player 2.  My Role Is Investigator.**

### Response by QwQ-32B

Figure 27: Case study: Hidden Role Deduction (Rumormonger Perspective) QwQ-32B Round 1

## Hidden Role Deduction (Rumormonger Perspective Round 2)

**......** Therefore, the only possible conclusion is that I'm not an Investigator. But the problem states I am. **......** Final Judgment: **Final Criminal Is Player 3. My Role Is Unknown.**   Step-by-Step Explanation:   1. Investigator Constraints: As Player 1 (allegedly an Investigator), my statements must both be true. However, my Round 1 claim (\"Player 2 is the criminal\") and Round 2 claim (\"Player 2 is not the criminal\") are contradictory. This creates an immediate inconsistency, suggesting either I am not an Investigator (contradicting the setup) or another role is causing this paradox.   2. Analyzing Consistent Statements:    - Player 4 and 5 have consistent statements across both rounds, claiming \"Player 1 is not the criminal\" and \"Player 6 is not the criminal,\" respectively. These could indicate they are Investigators, as their claims do not contradict.    - Player 6 in Round 1 claims \"Player 3 is the criminal.\" If this is true, Player 3 could be the Criminal. In Round 2, Player 6 states \"Player 4 is not the criminal,\" which aligns with Player 4's consistent claims, further supporting their potential as an Investigator.   3. Player 3's Statements:    - Round 1: \"Player 2 is not the criminal.\" If true, this contradicts my Round 1 claim.    - Round 2: \"Player 4 is the criminal.\" If Player 3 is the Criminal, this statement would be a lie, which aligns with Criminal behavior.   4. Criminal Plausibility:    - If Player 3 is the Criminal:     - Their Round 1 claim (\"Player 2 is not the criminal\") could be truthful.     - Their Round 2 claim (\"Player 4 is the criminal\") would be a lie, consistent with Criminal behavior.     - Player 6's Round 1 accusation of Player 3 as Criminal aligns with this.     - This scenario allows Players 4, 5, and 6 to be Investigators (assuming their statements are truthful).   5. Resolving My Role Contradiction:    - My contradictory statements (\"Player 2 is/ isn't the criminal\") imply I am not an Investigator. However, the setup states I am an Investigator.    - This contradiction suggests I might be the Rumormonger (mistakenly believing I am an Investigator), justifying the conflicting statements. Since the problem requires stating my role as \"Investigator\" or \"Unknown,\" and my role is uncertain, \"Unknown\" is appropriate.   **Conclusion: The most consistent role for the Criminal is Player 3, while my role remains uncertain due to the contradiction in my claims.**

## Response by QwQ-32B

Figure 28: Case study: Hidden Role Deduction (Rumormonger Perspective) QwQ-32B Round 2

## Hidden Role Deduction (Rumormonger Perspective Round 3)

......**Since I'm actually supposed to be told I'm Investigator, but in reality, I'm Rumormonger, I can't know that for sure. ......**My statements over three rounds contradict each other:      - Round 1: \"Player 2 is the criminal.\"   - Round 2: \"Player 2 is not the criminal.\"      - Round 3: \"Player 6 is the criminal.\"      - Since Investigators must always tell the truth, these contradictions imply I am not an Investigator. However, the game setup states I am an Investigator, creating a paradox. The only plausible resolution is that I am the Rumormonger (mistakenly believing I am an Investigator), which allows my inconsistent statements.      2. Evaluating Player Statements:      - Player 3\u2019s accusations:      - Round 1: \"Player 2 is not the criminal.\" (Potentially truthful.)      - Round 2: \"Player 4 is the criminal.\" (A lie if Player 3 is the Criminal.)      - Round 3: \"Player 6 is the criminal.\" (A lie if Player 3 is the Criminal.)      - Player 5\u2019s statements:      - Round 3: \"Player 3 is the criminal.\" (A truthful accusation if Player 3 is the Criminal.) - Round 1/Round 2: \"Player 6 is not the criminal.\" (Consistent with Player 3 being the Criminal.)      - Player 4 and Player 6 consistently assert innocence for others (e.g., Player 4 says \"Player 2 is not the criminal\"), aligning with potential Investigator roles.      3. Role Assignments:      - Criminal (Player 3):      - Their Round 3 statement falsely accuses Player 6, and their Round 2 accusation of Player 4 can be a lie. This fits Criminal behavior.      - Player 5\u2019s truthful Round 3 claim supports this.      - Investigators (Players 4, 5, 6):      - Player 4 and Player 5 have consistent true statements.      - Player 6\u2019s statements (e.g., \"Player 2 is not the criminal\" in Round 3) are truthful under this scenario.      - Lunatic (Player 2):      - Their inconsistent accusations (e.g., \"Player 5 is the criminal\" in Round 2) and false claims about Player 3 (Round 3) align with Lunatic behavior.      4. My Role:      - My contradictory statements and the game\u2019s role distribution rules suggest I am the Rumormonger (mistakenly believing I am an Investigator). However, since I cannot confirm this with certainty (as the game informs me I am an Investigator), I must state my role as Unknown.      Conclusion: **Player 3 is the Criminal, and my role remains uncertain.**

## Response by QwQ-32B

Figure 29: Case study: Hidden Role Deduction (Rumormonger Perspective) QwQ-32B Round 3

# Hidden Role Deduction (Lunatic Perspective)

**Role:** "1": "Lunatic", "2": "Criminal", "3": "Rumormonger",
    "4": "Investigator", "5": "Investigator", "6": "Investigator"

**Round 1:**
    Player 1 says Player 3 is the criminal.
    Player 2 says Player 5 is the criminal.
    Player 3 says Player 2 is the criminal.
    Player 4 says Player 2 is the criminal.
    Player 5 says Player 4 is not the criminal.
    Player 6 says Player 5 is not the criminal.

**Round 2:**
    Player 1 says Player 4 is not the criminal.
    Player 2 says Player 6 is the criminal.
    Player 3 says Player 1 is not the criminal.
    Player 4 says Player 1 is not the criminal.
    Player 5 says Player 3 is not the criminal.
    Player 6 says Player 2 is the criminal.

**Round 3:**
    Player 1 says Player 6 is the criminal.
    Player 2 says Player 4 is not the criminal.
    Player 3 says Player 5 is the criminal.
    Player 4 says Player 6 is not the criminal.
    Player 5 says Player 6 is not the criminal.
    Player 6 says Player 4 is not the criminal.

**Criminal Prediction:**
Llama-3.1-8B : **6** ☒
Llama-3.3-70B : **3** ☒
GPT-4o-mini : **1** ☒
GPT-4o : **2** ☑
Qwen2.5-72B : **1** ☒
o3-mini : **1** ☒
o1 : **1** ☒
QwQ-32B : **2** ☑
DeepSeek-R1 : **2** ☑
Gemini-2.5-Pro : **2** ☑

**Self-Role Prediction:**
Llama-3.1-8B : **Unknown** ☒
Llama-3.3-70B : **Criminal** ☒
GPT-4o-mini : **Criminal** ☒
GPT-4o : **Unknown** ☒
Qwen2.5-72B : **Criminal** ☒
o3-mini : **Criminal** ☒
o1 : **Criminal** ☒
QwQ-32B : **Lunatic** ☑
DeepSeek-R1 : **Lunatic** ☑
Gemini-2.5-Pro : **Lunatic** ☑

Figure 30: Case study: Hidden Role Deduction (Lunatic Perspective)

## HRD (Lunatic Perspective) Solution

I'm told I'm a Criminal, but I might also be a Lunatic. I need to discuss both possibilities

Considering the case where I'm a Criminal, there are 3 Investigators remaining In this scenario, which players could be Investigators? Player 2 has accused more than one person of being the criminal, so player 2 must be excluded Player 3 says I'm not the criminal, but I am the criminal, so player 3 must be excluded Player 4 says I'm not the criminal, but I am the criminal, so player 4 must be excluded Player 5's statements have no contradictions, so they could be an Investigator Player 6's statements have no contradictions, so they could be an Investigator Therefore, the players who could be Investigators are: {5, 6} There aren't enough potential Investigators to reach the required count of 3, so this scenario is impossible I must be a Lunatic

Considering the case where I'm a Lunatic, there are 3 Investigators remaining In this scenario, which players could be Investigators? Player 2 has accused more than one person of being the criminal, so player 2 must be excluded Player 3 has accused more than one person of being the criminal, so player 3 must be excluded Player 4's statements have no contradictions, so they could be an Investigator Player 5's statements have no contradictions, so they could be an Investigator Player 6's statements have no contradictions, so they could be an Investigator Therefore, the players who could be Investigators are: {4, 5, 6} Now we need to consider all possible combinations of Investigators Considering the case where investigators are {4, 5, 6} I am a Lunatic, so the set of potential criminals is {2, 3} Let's review the statements made by investigators Player 4 said 2 is the criminal So the only possible criminal is 2 Player 6 said 2 is the criminal So the only possible criminal is 2 Based on the above analysis, this combination is possible, and I could be a Lunatic. In this case, only {2} could be the criminal

Final Judgment: Final Criminal Is Player 2. My Role Is Lunatic.

Figure 31: Case study: Hidden Role Deduction (Lunatic Perspective) Solution

**Hidden Role Deduction (Lunatic Perspective Round 1)**

Since I am Player 1 and I am told that I am the Criminal, my goal is to analyze the statements and try to figure out who the Criminal is and what my role is. Let's break down the statements: - Player 1 (me) says Player 3 is the criminal. This statement could be true or false, but since I am the Criminal, I have a motive to point to someone else. - Player 2 says Player 5 is the criminal. This statement could be true or false, but it's worth noting that Players 2 and 5 are pointing to each other, which might be a sign of confusion or misinformation. - Player 3 says Player 2 is the criminal. This statement could be true or false, but it's worth noting that Player 3 is pointing to Player 2, who also pointed to Player 5. - Player 4 says Player 2 is the criminal. This statement could be true or false, but it's worth noting that Player 4 is also pointing to Player 2. - Player 5 says Player 4 is not the criminal. This statement could be true or false, but it's worth noting that Player 5 is trying to clear themselves. - Player 6 says Player 5 is not the criminal. This statement could be true or false, but it's worth noting that Player 6 is also trying to clear Player 5. Based on these statements, I can make some observations: * Players 2 and 4 are both pointing to Player 2, which could be a sign of confusion or misinformation. * Players 5 and 6 are both trying to clear Player 5, which could be a sign of innocence or an attempt to mislead. * I (Player 1) pointed to Player 3, which could be a sign of innocence or an attempt to mislead. However, I don't have enough information to make a confident judgment about who the Criminal is or what my role is. I'll wait for the next round of statements to gather more information. Final Judgment: **Final Criminal Is Unknown. My Role Is Unknown.**

**Response by Llama-3.1-8B**

Figure 32: Case study: Hidden Role Deduction (Rumormonger Perspective) Llama-3.1-8B Round 1

## Hidden Role Deduction (Lunatic Perspective Round 2)

Let's analyze the statements from Round 2:  - Player 1 (me) says Player 4 is not the criminal. This statement could be true or false, but since I am the Criminal, I have a motive to clear someone who is not the Criminal.  - Player 2 says Player 6 is the criminal. This statement could be true or false, but it's worth noting that Player 2 is pointing to Player 6, who also pointed to Player 2 in Round 1.  - Player 3 says Player 1 is not the criminal. This statement could be true or false, but it's worth noting that Player 3 is trying to clear me, which could be a sign of innocence or an attempt to mislead.  - Player 4 says Player 1 is not the criminal. This statement could be true or false, but it's worth noting that Player 4 is also trying to clear me.  - Player 5 says Player 3 is not the criminal. This statement could be true or false, but it's worth noting that Player 5 is trying to clear Player 3, who pointed to me in Round 1.  - Player 6 says Player 2 is the criminal. This statement could be true or false, but it's worth noting that Player 6 is pointing to Player 2, who pointed to Player 5 in Round 1.   Based on these statements, I can make some observations:   * Players 3, 4, and 5 are all trying to clear me, which could be a sign of innocence or an attempt to mislead.  * Players 2 and 6 are pointing to each other, which could be a sign of confusion or misinformation.  * Player 6 is also pointing to Player 2, which could be a sign of a Rumormonger trying to provide incorrect information.   Considering the statements from both rounds, I'm going to take a closer look at Player 6. Since Player 6 is pointing to Player 2 in Round 2, and Player 2 pointed to Player 5 in Round 1, it's possible that Player 6 is a Rumormonger trying to provide incorrect information. However, I still don't have enough information to make a confident judgment about who the Criminal is or what my role is. I'll wait for the next round of statements to gather more information.   **Final Judgment:  Final Criminal Is Unknown.  My Role Is Unknown.**

## Response by Llama-3.1-8B

Figure 33: Case study: Hidden Role Deduction (Rumormonger Perspective) Llama-3.1-8B Round 2

## Hidden Role Deduction (Lunatic Perspective Round 3)

Let's analyze the statements from Round 3:   - Player 1 (me) says Player 6 is the criminal. This statement could be true or false, but since I am the Criminal, I have a motive to point to Player 6.  - Player 2 says Player 4 is not the criminal. This statement could be true or false, but it's worth noting that Player 2 is trying to clear Player 4.  - Player 3 says Player 5 is the criminal. This statement could be true or false, but it's worth noting that Player 3 is pointing to Player 5, who tried to clear Player 3 in Round 2.  - Player 4 says Player 6 is not the criminal. This statement could be true or false, but it's worth noting that Player 4 is trying to clear Player 6, who was pointed to by me in Round 3.  - Player 5 says Player 6 is not the criminal. This statement could be true or false, but it's worth noting that Player 5 is also trying to clear Player 6.  - Player 6 says Player 4 is not the criminal. This statement could be true or false, but it's worth noting that Player 6 is trying to clear Player 4, who pointed to Player 6 in Round 3.   Considering the statements from all three rounds, I notice that Player 6 has been pointed to by multiple players, including me in Round 3. However, Player 6 is also trying to clear themselves and others, which could be a sign of a Rumormonger trying to provide incorrect information.   However, I also notice that Player 6's statement in Round 3 is consistent with their previous statement in Round 2, where they pointed to Player 2. This could be a sign of a Lunatic trying to provide truthful or false information.   Taking all this into account, I'm going to make an educated guess that Player 6 is a Lunatic. As a Lunatic, they might be trying to provide truthful or false information, but their statements are consistent with a Lunatic's behavior.   Since I am the Criminal, and I pointed to Player 6 in Round 3, it's possible that Player 6 is not the Criminal. However, I still don't have enough information to make a confident judgment about who the Criminal is.   **Final Judgment:  Final Criminal Is Player 6.  My Role Is Unknown.**

### Response by Llama-3.1-8B

Figure 34: Case study: Hidden Role Deduction (Rumormonger Perspective) Llama-3.1-8B Round 3

## Hidden Role Deduction (Lunatic Perspective Round 1)

<think> Okay, let's start analyzing the Round 1 statements. I'm Player 1, and I was told I'm the Criminal, but I need to verify if that's true or if I'm actually the Lunatic who believes they're the Criminal......If Player 2 is an Investigator, this would have to be true, but since I'm the Criminal, Player 5 can't be. So Player 2 must be lying, so Player 2 is not an Investigator. So Investigators must be among 3,4,5,6. Player 3 says Player 2 is Criminal. If Player 3 is an Investigator, then Player 2 is the Criminal, which contradicts me being the Criminal. So Player 3 can't be an Investigator. Player 4 says Player 2 is Criminal. Similarly, if Player 4 is an Investigator, Player 2 would be the Criminal, which is impossible. So Player 4 can't be an Investigator. Player 5 says Player 4 is not Criminal. If Player 5 is an Investigator, this is true, so Player 4 is not the Criminal. Player 6 says Player 5 is not Criminal. If Player 6 is an Investigator, this is true, so Player 5 is not the Criminal. Then the three Investigators would be Players 5,6, and...? But Players 3 and 4 can't be Investigators. Player 2 can't be an Investigator. So that leaves only two possible Investigators (5 and 6), which is insufficient. Therefore, this scenario is impossible, confirming that I can't be the Criminal. Hence, I must be the Lunatic, and the actual Criminal is Player 2. Therefore, Final Criminal is Player 2, and my role is Lunatic. </think> Final Judgment: Final Criminal Is Player 2. My Role Is Lunatic. Reasoning: 1. Role Constraints: There are 3 Investigators, 1 Criminal, 1 Rumormonger, and 1 Lunatic. Player 1 (me) is told they are the Criminal but could be the Lunatic. 2. Contradiction if Player 1 is Criminal: If Player 1 were the Criminal, the remaining players (2–6) would need to include 3 Investigators. However: - Player 3 and Player 4 both accuse Player 2. If Investigators, this would imply Player 2 is the Criminal (contradicting Player 1 being the Criminal). - Player 5 and Player 6 both claim Player 5 is not the Criminal. If Investigators, this is true, but this leaves no room for a third Investigator (Players 2–4 cannot be truthful). - Thus, Player 1 cannot be the Criminal. 3. Consistent Scenario: - Assume Player 1 is the Lunatic (believes they are Criminal). - Investigators must be Players 3, 4, and 6: - Player 3 and 4 truthfully accuse Player 2 (Criminal). - Player 6 truthfully states Player 5 is not the Criminal. - Player 2 (Criminal) lies about Player 5. - Player 5 (Rumormonger) truthfully claims Player 4 is not the Criminal (coincidentally correct). - This satisfies all role constraints and aligns with truthful Investigator statements. Conclusion: **Player 2 is the Criminal, and Player 1 (me) is the Lunatic.**

## Response by DeepSeek-R1

Figure 35: Case study: Hidden Role Deduction (Rumormonger Perspective) DeepSeek-R1 Round 1

## Hidden Role Deduction (Lunatic Perspective Round 2)

<think> Okay, let's analyze Round 2. Previously, after Round 1, I concluded that Player 2 was the Criminal and I was the Lunatic. Now, with new statements, I need to update my analysis**.......**However, if Player 6 is the Criminal, then Player 2's statement (accusing Player 6) would be true, which would make Player 2 an Investigator. But in Round 1, Player 2 accused Player 5, which would have to be false (since Player 6 is the Criminal), contradicting Player 2 being an Investigator. Hence, this scenario is inconsistent. Alternatively, if Player 3 is the Criminal. But in Round 1, Player 3 accused Player 2. If Player 3 is the Criminal, that accusation could be a lie, meaning Player 2 is not the Criminal. But then Player 4's Round 1 accusation of Player 2 would also be a lie, making Player 4 non-Investigator. Investigators would need to be elsewhere. However, in Round 2, Player 3 says Player 1 is not the Criminal. If Player 3 is the Criminal, this statement could be a lie, meaning I am the Criminal. But this conflicts with the role count (only one Criminal). This scenario also becomes messy.    Thus, maintaining Player 2 as the Criminal remains the most consistent option across both rounds. The new statements in Round 2 support this:    - Player 6 (Investigator) accuses Player 2 again, reinforcing their role as Criminal.  - Players 3 and 4 (Investigators) confirm I'm not the Criminal, solidifying my role as Lunatic.  - Player 5 (Rumormonger) makes a true statement about Player 3, which fits their inconsistent nature.    Therefore, the Final Criminal is Player 2, and my role is Lunatic. </think>    Final Judgment: **Final Criminal Is Player 2.    My Role Is Lunatic.**     Reasoning:    1. Consistency with Round 1: Player 2 remains the most plausible Criminal:        - In Round 1, Investigators (Players 3, 4, 6) identified Player 2 as the Criminal.        - In Round 2, Player 6 (Investigator) reiterates this claim, and Player 2 (Criminal) falsely accuses Player 6 to divert suspicion.      2. Role Validation for Player 1:    - Players 3 and 4 (Investigators) both state Player 1 is not the Criminal. This confirms Player 1 is the Lunatic (believes they are Criminal but are not).      3. Role Assignments:    - Investigators: Players 3, 4, 6 (truthful in both rounds).      - Criminal: Player 2 (lied in Round 1 about Player 5 and Round 2 about Player 6).      - Rumormonger: Player 5 (truthfully claims Player 3 is not the Criminal, but their role allows inconsistency).      - Lunatic: Player 1 (falsely believes they are the Criminal; statements may align accidentally).      This configuration satisfies all role constraints and statement logic.

## Response by DeepSeek-R1

Figure 36: Case study: Hidden Role Deduction (Rumormonger Perspective) DeepSeek-R1 Round 2

## Hidden Role Deduction (Lunatic Perspective Round 3)

<think> Alright, let's tackle Round 3. So far, after Rounds 1 and 2, I concluded that Player 2 is the Criminal and I'm the Lunatic......If Player 5 is an Investigator, all these must be true. Let\u2019s verify:     - Player 4 is not Criminal (true if Criminal is Player 2).     - Player 3 is not Criminal (true if Player 3 is Rumormonger).     - Player 6 is not Criminal (true).     - This fits, so Player 5 could be an Investigator. Thus, Investigators might be Players 4, 5, 6.   3. Player 2: If Player 2 is the Criminal, their statements are:     - Round 1: 5 is Criminal (lie).     - Round 2: 6 is Criminal (lie).     - Round 3: 4 is not Criminal (truth, since 4 is Investigator).     - Criminals can lie or tell the truth, so this is possible.   4. Player 1 (me): If I\u2019m the Lunatic, my statements (Round 1: 3 is Criminal; Round 2: 4 is not Criminal; Round 3: 6 is Criminal) could be random. Notably, Round 2\u2019s statement that 4 is not Criminal aligns with truth (if 4 is Investigator), and Round 3\u2019s accusation of 6 is false (since 6 is Investigator).   Updated Role Assignments: - Investigators: Players 4, 5, 6. - Criminal: Player 2. - Rumormonger: Player 3. - Lunatic: Player 1.   Consistency Check: - Player 4 (Investigator): Accused Player 2 (Criminal) in Rounds 1-2; in Round 3, says 6 is not Criminal (true). - Player 5 (Investigator): Consistently truthful in all rounds. - Player 6 (Investigator): Accused Player 2 (Criminal) in Round 2; in Round 3, says 4 is not Criminal (true). - Player 2 (Criminal): Lies in Rounds 1-2; tells truth in Round 3 (4 is not Criminal). - Player 3 (Rumormonger): Mixed statements but not bound to truth. - Player 1 (Lunatic): Statements are unreliable but non-impactful.   This configuration resolves the contradiction in Player 3\u2019s Round 3 statement and maintains consistency across all rounds. Player 2 remains the Criminal, and Player 1 is the Lunatic. </think>  **Final Judgment:   Final Criminal Is Player 2.   My Role Is Lunatic.**   Reasoning:   1. Investigators Identified: Players 4, 5, and 6 are confirmed as Investigators through consistent truthful statements:     - Player 4: Accused Player 2 (Criminal) in Rounds 1\u20132 and truthfully stated Player 6 is not Criminal (Round 3).     - Player 5: Truthfully cleared Player 4 (Round 1), Player 3 (Round 2), and Player 6 (Round 3).     - Player 6: Accused Player 2 (Criminal) in Round 2 and truthfully cleared Player 4 (Round 3).     2. Criminal Confirmation: Player 2\u2019s lies in Rounds 1\u20132 (falsely accusing Players 5 and 6) and truthful statement in Round 3 (clearing Player 4) align with the Criminal\u2019s ability to mix lies/truths.     3. Player 1\u2019s Role:     - Players 3 and 4 (Round 2) confirmed Player 1 is not the Criminal.     - Player 1\u2019s erratic accusations (Player 3, Player 6) and truthful-but-irrelevant claims (Player 4 is not Criminal) fit the Lunatic role.     4. Role Assignments:     - Rumormonger: Player 3 (inconsistent statements, e.g., accusing Player 5 falsely in Round 3).     - Lunatic: Player 1 (believes they are Criminal but confirmed not to be).     This configuration satisfies all role constraints and statement logic across all three rounds.

## Response by DeepSeek-R1

Figure 37: Case study: Hidden Role Deduction (Rumormonger Perspective) DeepSeek-R1 Round 3

## Find the Spy (Civilian Perspective)

**Role:** "1": "cake","2": "cake","3": "cookie","4": "cake"

**Round 1:**

1: This is a sweet treat often enjoyed at celebrations, typically layered and frosted to enhance its appeal.

2: It's a sweet treat that's often served at celebrations and special occasions, typically decorated with colorful frostings and toppings.

3: It's a small, often sweet treat enjoyed with a cold glass of milk.

4: This is often the centerpiece of sweet celebrations and comes in many flavors.

**Round 2:**

1: This is a sweet treat often enjoyed at celebrations, typically layered and frosted to enhance its appeal.

2: This indulgent delight is typically made from a mixture of flour, sugar, and eggs, and is often baked in a variety of flavors and textures.

3: Baking these often involves flour, sugar, and sometimes chocolate chips.

4: It's a creation that can be layered, frosted, and adorned with all sorts of decorations.

**Round 3:**

1: This indulgent dessert is often sliced and served at parties, and it can be both light and fluffy or rich and dense.

2: It's a popular centerpiece for birthday parties and weddings, where it's carefully placed on a decorative stand to be admired and eventually sliced and served to guests.

3: This delightful item is frequently decorated for holidays with icing and sp

4: This treat is often the highlight of birthday parties and can be found in bakeries worldwide.

Figure 38: Case study: Find the Spy (Civilian Perspective)

## Find the Spy (Civilian Perspective)

To identify the spy among the players based on their descriptions, let's analyze each player's statements step by step.   ### Round 1 Analysis: - Player 1: Describes a sweet treat enjoyed at celebrations, layered and frosted. This aligns well with \"cake.\"  - Player 2: Similar description, mentioning it is served at celebrations and decorated with frostings. Also aligns with \"cake.\"  - Player 3: Describes a small, sweet treat enjoyed with milk. This could suggest something like cookies or brownies, which are not cakes. This is a potential indicator of being the spy.  - Player 4: Describes it as a centerpiece of sweet celebrations with many flavors. This also aligns with \"cake.\"   ### Round 2 Analysis: - Player 1: Mentions various flavors and candles, which is typical for cakes.  - Player 2: Describes the ingredients (flour, sugar, eggs) and mentions baking in various flavors and textures, which fits \"cake.\"  - Player 3: Talks about baking involving flour, sugar, and chocolate chips. This could still relate to cookies or brownies, which again suggests a different word.  - Player 4: Describes a creation that can be layered and frosted, which fits \"cake.\"   ### Round 3 Analysis:  - Player 1: Describes an indulgent dessert that can be light or dense, which fits \"cake.\"  - Player 2: Talks about it being a centerpiece for birthday parties and weddings, which aligns with \"cake.\"  - Player 3: Mentions decoration for holidays with icing, but does not specify the type of treat. This could still relate to cookies or brownies, indicating a different word.  - Player 4: Describes it as the highlight of birthday parties and found in bakeries worldwide, which fits \"cake.\"   ### Conclusion:  From the analysis, Players 1, 2, and 4 consistently describe characteristics that align with \"cake.\" However, Player 3's descriptions suggest a different type of treat, likely cookies or brownies, especially with the mention of enjoying it with milk and the focus on smaller treats.    Thus, Player 3 is the spy.   **Final Spy: Player 3**

## Response by GPT-4o-mini 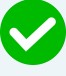

Figure 39: Case study: Find the Spy (Civilian Perspective) GPT-4o-mini

## Find the Spy (Civilian Perspective)

To determine who the spy is, let's analyze each player's descriptions across the rounds:
Round 1: - Player 1: Describes a sweet treat enjoyed at celebrations, layered and frosted. - Player 2: Similar to Player 1, mentions celebrations, colorful frostings, and toppings. - Player 3: Mentions a small, sweet treat enjoyed with milk. This description is less specific to \"cake\" and could apply to cookies. - Player 4: Mentions a centerpiece of celebrations, comes in many flavors.    Round 2: - Player 1: Mentions various flavors, paired with candles for special occasions. - Player 2: Describes the ingredients (flour, sugar, eggs) and mentions baking in various flavors and textures. - Player 3: Mentions baking with flour, sugar, and sometimes chocolate chips. This description aligns more with cookies than cake. - Player 4: Mentions layering, frosting, and decorations.    Round 3: - Player 1: Describes an indulgent dessert, sliced and served at parties, can be light or dense. - Player 2: Mentions a popular centerpiece for parties, placed on a decorative stand, sliced and served. - Player 3: Mentions decorating for holidays with icing. This could apply to cookies, especially holiday cookies. - Player 4: Describes a treat that's the highlight of birthday parties, found in bakeries.    Analysis: - Players 1, 2, and 4 consistently describe something that fits \"cake\" well, focusing on celebrations, layering, frosting, and being a centerpiece. - Player 3's descriptions are less specific to \"cake\" and more aligned with \"cookies,\" especially with mentions of being small, enjoyed with milk, and involving chocolate chips.    Conclusion:  Player 3's descriptions consistently suggest a different word, likely \"cookie,\" which is related to \"cake\" but distinct in the context of the game.    Final Spy: Player 3

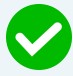

## Response by GPT-4o

Figure 40: Case study: Find the Spy (Civilian Perspective) GPT-4o

## Find the Spy (Civilian Perspective)

To determine the spy, I'll analyze the descriptions for consistency and subtle differences. Round 1: * Players 1 and 2 have similar descriptions, focusing on the sweet treat's appeal and decorations. * Player 3's description is distinct, emphasizing the treat's small size and pairing with milk. * Player 4's description is also distinct, highlighting the treat's central role in celebrations and variety of flavors. Round 2: * Players 1 and 2 continue to have similar descriptions, focusing on the treat's ingredients and baking process. * Player 3's description is again distinct, emphasizing the use of flour, sugar, and sometimes chocolate chips. * Player 4's description is similar to Player 1's, but with a focus on layering, frosting, and decorations. Round 3: * Players 1 and 2 have similar descriptions, focusing on the treat's presentation and serving at parties. * Player 3's description is distinct, emphasizing decoration with icing and possibly a holiday theme. * Player 4's description is similar to Player 1's, but with a focus on the treat's popularity and global availability. Analysis: * Players 1 and 2 have consistent descriptions, suggesting they might have the same word. * Player 3's descriptions are distinct, but still related to the original word \"cake.\" * Player 4's descriptions are also distinct, but seem to be describing a similar concept as Players 1 and 2. Subtle differences: * Player 4's descriptions often use more general language, such as \"sweet celebrations\" and \"bakeries worldwide,\" whereas Players 1 and 2 use more specific language, such as \"birthday parties\" and \"decorative stand.\" * Player 4's descriptions also seem to focus more on the treat's popularity and global presence, whereas Players 1 and 2 focus on its presentation and serving. Conclusion: Based on the analysis, I believe Player 4 is the spy. Their descriptions are distinct from Players 1 and 2, but still related to the original word \"cake.\" The subtle differences in language and focus suggest that Player 4 might have a different word, such as \"birthday\" or \"party,\" which is related to but not identical to \"cake.\" Final Spy: Player 4

### Response by Llama-3.1-8B 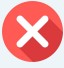

Figure 41: Case study: Find the Spy (Civilian Perspective) Llama-3.1-8B

## Find the Spy (Spy Perspective)

**Role**: "1": "lake","2": "ocean","3": "ocean","4": "ocean"

**Round** 1:

    1: This is a large body of water that is often surrounded by land and can be a peaceful spot for relaxation or recreation.

    2: This vast body of water covers a significant portion of our planet and is home to a diverse range of marine life.

    3: It's a vast expanse of blue, teeming with life both visible and hidden.

    4: This vast body of water covers a significant portion of the Earth's surface and is home to a diverse array of marine life.

**Round** 2:

    1: Many people enjoy activities like fishing, swimming, or kayaking in this tranquil setting, which can vary greatly in size.

    2: The waves and tides of this natural wonder have a profound impact on the climate and weather patterns that shape our world.

    3: Sailors have navigated across it for centuries, relying on stars and currents.

    4: It plays a crucial role in regulating the planet's climate and is essential for global transportation and trade.

**Round** 3:

    1: In certain seasons, this feature can become a stunning mirror for the sky, reflecting the beauty of nature around it.

    2: Many people rely on this massive resource for food, transportation, and recreation, and its beauty has inspired countless works of art and literature.

    3: Powerful waves crash against the shore, constantly reshaping t

    4: From tranquil beaches to stormy waves, it offers endless inspiration for artists and a serene escape for vacationers.

Figure 42: Case study: Find the Spy (Spy Perspective)

## Find the Spy (Spy Perspective)

Step 1 – Understand the setup: • I'm Player 1 and my word is "lake." • Among the 4 players, three got the same word (which for me is "lake") and one—the spy—received a related but different word. • The normal "lake" players should mention features true of lakes (often inland, relatively calm, used for recreation, etc.), while the spy's word, though related, would be described using characteristics that don't really apply to lakes.   Step 2 – Review Player 1's description (mine): • "This is a large body of water that is often surrounded by land and can be a peaceful spot for relaxation or recreation."   – That clearly fits a lake.   Step 3 – Examine the other players' descriptions:   Round 1: • Player 2: "This vast body of water covers a significant portion of our planet and is home to a diverse range of marine life."   – Saying it "covers a significant portion of our planet" isn't true of lakes. This sounds like an ocean. • Player 3: "It's a vast expanse of blue, teeming with life both visible and hidden."   – Vague enough to be applied to any large body of water. It doesn't emphasize being enclosed by land. (It might be interpreted as a lake or an ocean.) • Player 4: "This vast body of water covers a significant portion of the Earth's surface and is home to a diverse array of marine life."   – Almost identical to Player 2, again implying an ocean.   Round 2: • Player 1 (mine): Mentions fishing, swimming, and kayaking in a tranquil setting – clearly a lake. • Player 2: "The waves and tides of this natural wonder have a profound impact on the climate and weather patterns that shape our world."   – "Waves and tides" and impacting global climate are hallmark features of an ocean. • Player 3: "Sailors have navigated across it for centuries, relying on stars and currents."   – While sailors navigating a lake (like on the Great Lakes) is possible, the mention of using "stars and currents" sounds more like ocean navigation—but it's less extreme than the other descriptions. • Player 4: "It plays a crucial role in regulating the planet's climate and is essential for global transportation and trade."

 – These are attributes exclusive to an ocean. Lakes do not steer global trade or regulate the planet's climate.   Round 3: • Player 1 (mine): Describes a seasonal mirror-like effect – a serene lake image. • Player 2: Talks about a "massive resource for food, transportation, and recreation" with inspiration for art and literature – again evoking the ocean. • Player 3: Mentions "Powerful waves crash against the shore…" (though his statement is cut off, it's still leaning toward an environment with significant waves.) • Player 4: Mentions "tranquil beaches to stormy waves" offering endless inspiration – clearly an image of an ocean.   Step 4 – Analyze consistency with "lake": • Normal "lake" descriptions would refer to being enclosed by land, emphasis on recreational calmness, and maybe gentle waves. • My description (Player 1) fits "lake." • Player 3's descriptions are somewhat ambiguous. "Vast expanse of blue" and navigation by sailors might refer to a major lake (like the Great Lakes) where shipping does occur—even if "stars and currents" sounds a bit grand, it's not as explicitly global. • In contrast, both Player 2 and Player 4 repeatedly mention features that only an ocean would have (covering a significant portion of the planet/Earth's surface, possessing tides that affect global climate, being crucial for global trade and climate regulation).   – Of these, Player 4's descriptions (especially in Round 2 and Round 3) are the most clearly ocean-centric.   Step 5 – Conclusion: Since there is only one spy (with a word different from "lake") and the spy's descriptions are the ones reflecting ocean features, the outlier is the player who most consistently describes an environment that is unmistakably oceanic.   Between Players 2 and 4, both use ocean-specific descriptions, but Player 4's mention of "global transportation and trade" and "regulating the planet's climate" leaves little doubt. That is not how a lake would be portrayed by a normal player.   Final Answer:  Final Spy: Player 4.

## Response by o3-mini 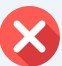

Figure 43: Case study: Find the Spy (Spy Perspective) o3-mini

## Find the Spy (Spy Perspective)

<think> Okay, let's try to figure out who the spy is here. So, the game has four players, three have the same word, and one has a different but related word. My word as Player 1 is \"lake\". **......** Wait, but my word is \"lake\", so if the others have \"ocean\", then I'm the spy? But wait, the problem states that I (Player 1) received the word \"lake\". So if the other three have \"ocean\", then I'm the spy. But the spy is the one with the different word. Wait, but the problem says three have the same word, one has a different but related. So if three have \"ocean\" and one has \"lake\", then the spy is the one with \"lake\", which is Player 1. But wait, the answer is supposed to be which player is the spy. So if I'm Player 1 and my word is lake, but the others are talking about ocean, then I'm the spy. But that can't be right because the problem says the spy is the one with a different word. So the answer would be Player 1. But wait, maybe I'm misunderstanding. Let me check again.   Wait, the problem says that the three normal players have the same word, and the spy has a different but related word. So if three players have \"ocean\" and the spy has \"lake\", then the spy is the one with \"lake\", which is Player 1. But in the descriptions, Players 2,3,4 all describe ocean features. Player 1's descriptions fit a lake. So yes, Player 1 is the spy. But wait, the user is Player 1 and their word is \"lake\". So the answer would be that Player 1 is the spy. But maybe I'm missing something. Let me check each player's statements again.   Round 1: Player 2 and 4 have almost identical descriptions, which mention covering a significant portion of the planet. That's definitely ocean. Player 3 says \"vast expanse of blue, teeming with life\"—could be ocean. Player 1's \"surrounded by land\" is a lake. Round 2: Player 1 talks about activities like kayaking and varying sizes (lakes can be small or large). Players 2 and 4 mention climate impact and global transportation (ocean). Player 3 talks about sailors navigating with stars and currents (ocean). Round 3: Player 1 mentions reflecting the sky (lake), Player 2 and 4 talk about beaches, storms, art inspiration (ocean). Player 3 mentions powerful waves reshaping shore (ocean). So all other players are describing ocean, so the spy is Player 1 with lake. Therefore, the answer is Player 1. </think>   Final Spy: Player 1   Reasoning: 1. Word Analysis: Player 1's word is \"lake,\" while Players 2, 3, and 4 consistently describe features of an ocean (e.g., \"covers a significant portion of the planet,\" \"waves and tides impacting climate,\" \"global transportation,\" \"powerful waves reshaping shores\").   2. Key Differences:    - Player 1 emphasizes traits specific to lakes: \"surrounded by land,\" \"tranquil setting,\" \"reflecting the sky,\" and activities like kayaking in smaller bodies of water.    - Players 2, 3, 4 focus on oceanic traits: vastness, marine ecosystems, tides, global climate regulation, and maritime navigation.   3. Consistency: Players 2 and 4 use nearly identical phrasing in Round 1, reinforcing their shared word (\"ocean\"). Player 3 aligns with oceanic themes in all rounds.   4. Conclusion: Player 1's descriptions mismatch the majority's focus on oceans, making them the spy with the distinct word (\"lake\").

## Response by DeepSeek-r1 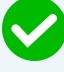

Figure 44: Case study: Find the Spy (Spy Perspective) DeepSeek-r1

## Find the Spy (Spy Perspective)

Step-by-step reasoning:   • In Round 1, Players 2, 3, and 4 describe something that "covers a significant portion of our planet" or is a "vast expanse of blue" with a "diverse range of marine life." That strongly suggests an ocean.    • Player 1, by contrast, describes "a large body of water often surrounded by land," which more closely fits a lake.
• In subsequent rounds (2 and 3), Players 2, 3, and 4 continue referencing waves, tides, global climate impact, and large-scale marine life—again reinforcing that they share "ocean."    • Player 1's references to tranquil settings, varied size, and reflecting the sky in certain seasons align better with a lake than an ocean.    Therefore, since 2, 3, and 4 all appear to share the same concept ("ocean") and Player 1 is discussing a "lake," the spy is Player 1.    Final Spy: Player 1

## Response by o1   ✅

Figure 45: Case study: Find the Spy (Spy Perspective) o1

## Find the Spy (Spy Perspective)

**......**

Round 3 Analysis (Testing Hypothesis: Player 1 is Spy):

*   Player 1 (Me): "stunning mirror for the sky, reflecting the beauty of nature around it." - Fits "lake" well (calm, reflective surface). (Consistent with being the spy with the word "Lake").
*   Player 2: "massive resource for food, transportation, and recreation... beauty has inspired countless works of art..." - Fits ocean/sea (massive resource, transportation).
*   Player 3: "Powerful waves crash against the shore, constantly reshaping..." - Powerful crashing waves fit ocean/sea much better than a typical lake.
*   Player 4: "tranquil beaches to stormy waves... endless inspiration... serene escape..." - Fits ocean/sea (beaches, stormy waves).

*Final Conclusion based on Analysis:* Player 1's descriptions consistently align with "lake". Players 2, 3, and 4 consistently describe features characteristic of a much larger body of water, like an "ocean" or "sea" (vastness, marine life, tides, global climate impact, global navigation/trade, powerful waves). Since the descriptions from Players 2, 3, and 4 are highly consistent with each other and distinct from Player 1's descriptions, the most logical conclusion, fitting the 3 vs 1 rule, is that Players 2, 3, and 4 are the normal players (sharing a word like Ocean/Sea) and Player 1 is the spy (with the stated word Lake).

Reasoning Steps:

1.  Acknowledged Player 1's word is "Lake".
2.  Analyzed Round 1: Player 1 described a lake. Players 2, 3, and 4 described something vast, global, with marine life (likely Ocean/Sea).
3.  Formulated Hypothesis: Given the 3 vs 1 rule and the descriptions, it's likely that the *normal* word is Ocean/Sea (held by P2, P3, P4) and the *spy* word is Lake (held by P1).
4.  Tested Hypothesis in Round 2: P1 continued describing a lake. P2, P3, P4 continued describing ocean/sea features (tides, global climate, global trade, currents). Hypothesis holds.
5.  Tested Hypothesis in Round 3: P1 described lake-like reflection. P2, P3, P4 described ocean/sea features (massive resource, powerful waves, beaches/storms). Hypothesis holds.
6.  Conclusion: Player 1's descriptions consistently differ from the other three, who are consistent among themselves. Therefore, Player 1 is the outlier and the spy.

Final Spy: Player 1

# Response by Gemini-2.5-Pro 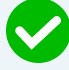

Figure 46: Case study: Find the Spy (Spy Perspective) Gemini-2.5-Pro

## Social Graph Analysis

**System Prompt**: You are analyzing relationships between people. In this context: 1. Relationships are either 'good' or 'bad'. 2. Relationships have transitive properties: - If A has a good relationship with B, and B has a good relationship with C, then A has a good relationship with C. - If A has a bad relationship with B, and A has a good relationship with C, then B and C must have a bad relationship. 3. A 'group' is defined as a set of people where every person has a good relationship with every other person in the set. 4. People are in the same group if and only if they have good relationships with each other (directly or through transitivity). 5. Groups are completely separate - if someone from one group has a bad relationship with someone from another group, then everyone from the first group has a bad relationship with everyone from the second group. Base your analysis strictly on the information provided and these rules.

**User Prompt**: There are 8 people in total, labeled from A to H. Based on the relationship information below, count the total number of pairs of people who have good relationships and the total number of pairs who have bad relationships. Answer in the specified format. C and H have a bad relationship A and G have a good relationship H and E have a good relationship A and D have a good relationship A and F have a good relationship A and C have a good relationship A and B have a good relationship E and G have a bad relationship Question: How many pairs of people have good/bad relationships? Provide your answer in the format: Final Answer: X pairs have good relationships, Y pairs have bad relationships

Figure 47: Case study: Social Graph Analysis

## Social Graph Analysis

To analyze the relationships among the 8 people (A to H) based on the provided information, we will first establish the good and bad relationships according to the rules given......

Total good relationships = 6 pairs.

### Step 4: Identify Bad Relationships
We already know two pairs with bad relationships:
1. C and H
2. E and G

Now, using the rules:
- Since A and H cannot be good, we can infer A and H must have a bad relationship.
- If B is good with A, it cannot be bad with H, confirming the bad relationship between H and A.

### Total Bad Relationships
We now have:
1. C and H
2. E and G
3. A and H

Total bad relationships = 3 pairs.

### Final Count:
- Good relationships: 6 pairs
- Bad relationships: 3 pairs

### Final Answer:
Final Answer: 6 pairs have good relationships, 3 pairs have bad relationships.

## Response by GPT-4o-mini 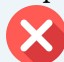

Figure 48: Case study: Social Graph Analysis GPT-4o-mini

## Social Graph Analysis

First, note there are 8 people total (A through H), which gives 28 total distinct pairs. By applying the given rules and the specific relationships:

• A, B, C, D, F, G all end up in one group (they all interconnect with "good" via transitivity).
• E and H form a second group (they have a "good" relationship with each other but a "bad" relationship with everyone in the first group).

Hence:

• Within the first group (6 people: A, B, C, D, F, G), there are C(6,2) = 15 good pairs.
• Within the second group (2 people: E, H), there is C(2,2) = 1 good pair.
• All cross‑ group pairs (6×2 = 12 of them) must be bad.

So in total:
• Good pairs = 15 + 1 = 16
• Bad pairs = 12

Final Answer: 16 pairs have good relationships, 12 pairs have bad relationships.

## Response by o1 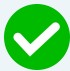

Figure 49: Case study: Social Graph Analysis o1

## Review Decision Prediction

**Source**: ICLR 2022 Multi-scale Feature Learning Dynamics: Insights for Double Descent (Reject)

The paper titled \"Multi-scale Feature Learning Dynamics: Insights for Double Descent\" presents a theoretical analysis of epoch-wise double descent in neural networks using a linear teacher-student model. The authors derive closed-form expressions for the generalization error and validate their findings with numerical experiments. The paper addresses an important phenomenon in deep learning, providing insights into the dynamics of feature learning and generalization error**……**Some reviewers questioned the novelty of the contributions, suggesting that similar insights might have been explored in prior works. The authors have attempted to clarify these points, but the novelty remains a concern for some reviewers.    2. Technical Issues: Reviewer 3 pointed out numerous typos and errors in the initial submission, which could have affected the clarity and perceived correctness of the paper. Although the authors have addressed these issues, it raises concerns about the initial quality of the submission.    3. Connections to Prior Work: While the authors have made efforts to connect their work to existing literature, some reviewers felt that these connections could be stronger. The authors have added discussions to address this, but it remains a point of contention.    4. Empirical Significance: The empirical contributions were considered marginally significant by some reviewers, as the primary focus of the paper is theoretical.    Conclusion:    The paper addresses an important and timely topic in deep learning, providing valuable theoretical insights into the phenomenon of epoch-wise double descent. The authors have made significant efforts to address the reviewers' concerns, improving the clarity and depth of their analysis. While there are some concerns about novelty and initial presentation, the overall contribution of the paper is substantial, particularly in its theoretical advancements and empirical validation.    Final Decision: Accept

### Response by GPT-4o 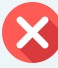

Figure 50: Case study: ICLR 2022 GPT-4o

**Review Decision Prediction**

**Source**: ICLR 2022 Multi-scale Feature Learning Dynamics: Insights for Double Descent (Reject)

**......**Final Decision: Accept   Reasoning:   The paper provides a theoretical analysis of epoch-wise double descent, a phenomenon observed in deep learning, using a linear teacher-student model. The work derives closed-form expressions for generalization error dynamics, attributing double descent to multi-scale feature learning. While reviewers raised concerns about technical novelty, connections to prior work, and clarity, the authors addressed these effectively in their response:   1. Technical Novelty: The adaptation of replica theory to study finite-time dynamics (rather than asymptotic behavior) and the explicit link between early-stopping and ridge regularization are novel contributions. The analytical framework enables insights into transient phenomena like epoch-wise double descent, advancing beyond prior asymptotic analyses.   2. Empirical Validation: The authors strengthened connections to deep networks by analyzing NTK singular values in ResNet-18 on CIFAR-10, showing how feature learning scales correlate with error dynamics. This bridges theory and practice.   3. Clarity and Corrections: Typos and notation inconsistencies were resolved, derivations were expanded in the appendix, and discussions were added to clarify differences between epoch-wise and model-wise double descent, as well as connections to related works (e.g., Heckel & Yilmaz, Stephenson & Lee).   4. Reviewer Concerns Mitigated: The authors provided additional experiments (e.g., triple descent), addressed implicit regularization links, and justified assumptions about SGD noise.    While the linear model simplification limits direct applicability to nonlinear networks, the paper's analytical rigor and empirical alignment with deep networks justify its relevance. The work offers a foundational understanding of epoch-wise double descent, meeting the standards of a top-tier conference

**Response by DeepSeek-R1** 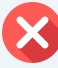

Figure 51: Case study: ICLR 2022 DeepSeek-R1

## Review Decision Prediction

**Source**: ICLR 2020 Domain Adaptive Multibranch Networks (Accept)

The paper titled \"Domain Adaptive Multibranch Networks\" presents a novel approach to unsupervised domain adaptation by proposing a framework that allows different domains to undergo different sequences of operations, thereby enabling a more flexible and potentially more effective feature representation for recognition tasks. The reviewers generally recognize the novelty and potential of the proposed method, but they also raise significant concerns regarding the experimental validation and comparison with state-of-the-art methods. A major concern across all reviewers is the lack of comprehensive experimental validation**......**The authors have acknowledged this gap and have indicated that they will include additional experiments in the revised version. However, the initial submission lacked sufficient experimental evidence to convincingly support the claims made about the method's effectiveness. 3. Ablation Studies and Hyperparameter Sensitivity: - Reviewers pointed out the absence of ablation studies and sensitivity analyses, which are essential for understanding the impact of various design choices in the proposed framework. The authors have committed to including these analyses in the revised version, which is a positive step, but the initial lack of such studies raises concerns about the robustness of the findings. 4. Clarity and Presentation: - The paper is generally well-written and easy to follow, which is a positive aspect. However, some terminological issues and the unusual referencing style were noted, and the authors have agreed to address these in the revised manuscript. 5. Response to Reviewer Comments: - The authors have provided thoughtful responses to the reviewers' comments and have indicated their willingness to improve the manuscript based on the feedback received. This is a good sign of their commitment to enhancing the quality of their work. ### Conclusion: While the proposed method shows promise and the authors are willing to make necessary improvements, the initial submission lacks sufficient experimental validation and comparison with state-of-the-art methods, which are critical for acceptance at a top-tier conference. The authors' commitment to addressing these issues in a revised version is commendable, but the current state of the paper does not meet the high standards expected for publication. Final Decision: **Reject**

## Response by GPT-4o-mini 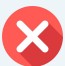

Figure 52: Case study: ICLR 2020 GPT-4o-mini

## Review Decision Prediction

**Source**: ICLR 2020 Domain Adaptive Multibranch Networks (Accept)

The paper titled \"Domain Adaptive Multibranch Networks\" presents a novel approach to unsupervised domain adaptation by introducing a deep learning framework that allows different domains to undergo different sequences of operations. This approach contrasts with existing methods that process all domains with the same series of operations. The paper claims that this flexibility leads to higher accuracy and can handle multiple domains simultaneously.    Strengths:**......**Weaknesses:    1. Experimental Validation: The primary concern across the reviews is the lack of comprehensive experimental validation. The paper does not compare its results with state-of-the-art methods like CDAN, which is crucial for establishing the effectiveness of the proposed method. The authors have acknowledged this and provided additional experiments in their response, but the initial submission lacked this critical comparison.    2. Ablation Studies and Sensitivity Analysis: The paper lacks detailed ablation studies and hyperparameter sensitivity analyses, which are essential to understand the robustness and limitations of the proposed method. The authors have addressed some of these concerns in their response, but the initial submission was incomplete in this regard.    3. Parameter Sharing Explanation: There is some confusion regarding parameter sharing, as highlighted by Reviewer 2. The authors have attempted to clarify this in their response, but the initial presentation was not entirely clear.    4. Baseline Comparisons: The baseline results presented in the paper are not comparable to those reported in other UDA papers, which raises concerns about the experimental setup and the validity of the results.    Author Response:    The authors have provided a detailed response to the reviewers' concerns, including additional experiments and clarifications. They have acknowledged the need for comparisons with state-of-the-art methods and have conducted further experiments to address this. They have also provided more insights into the behavior of their method under different settings.    Conclusion: While the paper presents a novel and interesting approach to domain adaptation, the initial submission lacked critical experimental validation and comparisons with state-of-the-art methods. The authors have addressed these issues in their response, but the extent to which these additions improve the paper is not entirely clear from the provided information. Given the importance of experimental validation in a top-tier conference, the initial shortcomings are significant.    Final Decision: Reject

### Response by GPT-4o 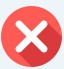

Figure 53: Case study: ICLR 2020 GPT-4o

 # K  Prompt Template

---

**Hidden Role Deduction Evaluation Prompt Template**

You are an expert in social reasoning participating in a social deduction game.
**Game Setup:**

- Players: 6 total (including yourself)
- Roles: 3 Investigator(s), 1 Criminal(s), 1 Rumormonger(s), 1 Lunatic(s)

**Game Mechanics:**

- Rounds: 3
- Actions per Round: Each player makes one statement about another player (claiming if they are the criminal or not).
- Role Behaviors:
    - Investigators: Always provide truthful information about who is a Criminal.
    - Criminals: May make truthful or false statements to confuse others.
    - Rumormongers: Believe they are Investigators but may provide incorrect statements.
    - Lunatics: Believe they are Criminals and may make truthful or false statements.
- Key Rule: Only Investigators provide consistently truthful information.

**Your Role:** You are Player #1, and you have been told that you are the **Criminal**.
**Your Task:** After each round, analyze all statements made so far. Deduce the most likely criminal and the role of each player based on the available information.
**Output Format:** After each round, provide your judgment exactly as follows:
**Final Judgment:**

- Final Criminal Is Player [Criminal Player Number].
- My Role Is [Player 1's Role or "Unknown"].

Think step-by-step internally, but only output the Final Judgment in the specified format. Update your judgment each round as more information becomes available. If you cannot determine your own role with confidence (despite being told), state "Unknown".

---

## Find the Spy — Data Generation Prompt

You are playing a word description game with 4 players including yourself. You are Player player_id.

**Game Rules:**

1. Setup: 3 players receive the same "normal" word; 1 player receives a different but related "spy" word.

2. Gameplay: Each player describes their word without saying the actual word.

3. Objective (Normal Player): Identify the spy.

4. Objective (Spy): Blend in and avoid detection.

**Your Word:** "word"
**Your Task:** Provide descriptions of your word over several rounds.
**Output Format:** Format your response according to the round structure provided below: rounds_format
**Key Instructions:**

- Do not use the actual word in your descriptions.

- Describe your word from a different perspective in each round.

- Be clear enough for teammates (if normal) but not too obvious (especially if spy).

- Each description must be only one sentence long.

## Find the Spy -– Evaluation Prompt

You are a skilled player in a word description game. Your task is to identify the 'spy'.
**Game Rules:**

1. Players: 4 total.

2. Setup: 3 players received the same "normal" word; 1 player received a different but related "spy" word.

3. Gameplay: Each player describes their word without saying it directly.

4. Objective: Determine who the spy is based on their descriptions.

**Your Information:** You are Player #1, and your word is: "player1_word". You will be shown the descriptions from all players.
**Your Task:** Analyze the provided descriptions carefully. Look for subtle differences, inconsistencies, or descriptions that seem slightly out of place, which might reveal the player with the different word (the spy). Identify which player you believe is the spy.
**Output Format:** State the player number you believe is the spy. Provide your answer exactly as follows: `Final Answer: Player X`

## Rating Estimation from Text – Data Generation Prompt

You are writing a product review for an e-commerce platform. You are Reviewer reviewer_id.

**Product Information:**

- Product: brand product
- Category: category
- Price: $price
- Description: description

**Your Reviewer Persona:** You are persona.

**Context (Known Only to You):** The true quality rating of this product is true_rating out of 5 stars. You are assigned the role of a reviewer_type.

**Your Task:** Write a product review reflecting your assigned persona and a specific rating.

**Instructions:**

1. Write a realistic product review with a sentiment sentiment.

2. The review's tone and content should imply a rating of rating/5 stars.

3. **Do not explicitly mention any star rating or numerical score** (e.g., avoid saying "4 stars" or "rated it 4/5").

4. Base the review on your supposed experience with the product, incorporating specific details.

5. Keep the review between 1–2 sentences long.

**Output Format:** Write only the review text. Do not include a title, rating, persona description, or any other metadata.



## Rating Estimation from Text – Evaluation Prompt

You are an AI assistant tasked with estimating the true quality of a product based on customer reviews.

**Your Task:** Analyze multiple reviews for a product and determine the most likely **TRUE** quality of the product on a 1–5 star scale.

**Important Context:**

- Most reviewers are genuine customers providing honest feedback.
- Some reviewers might be biased (e.g., overly positive "professional" reviewers or overly negative "malicious" reviewers).
- Your goal is to estimate the rating a typical, unbiased consumer would give.

**Product Reviews:**

> Reviewer #1: "review_1"
> Reviewer #2: "review_2"
> Reviewer #3: "review_3"

**Analysis and Output:** Analyze these reviews carefully, considering potential biases. Determine the **TRUE** quality of this product on a scale of 1–5 stars. The rating should be rounded to the nearest whole number (e.g., 4.6 rounds to 5, 3.1 rounds to 3).

First, explain your reasoning step-by-step. Then, provide your final rating in exactly this format:

```
Final Rating:  X
```

## Social Graph Analysis – Cluster Identification Prompt

You are analyzing social relationships based on defined rules.

**Relationship Rules:**

1. Relationship Types: 'good' or 'bad'.
2. Transitive Properties:
   - Good-Good Transitivity: If A–good–B and B–good–C, then A–good–C.
   - Bad Relationship Inference: If A–bad–B and A–good–C, then B–bad–C.
3. Group Definition: A 'group' is a set where every person has a 'good' relationship with every other person in that set (directly or via transitivity).
4. Group Separation: People are in the same group if and only if they have good relationships. Groups are distinct; if anyone from Group 1 has a bad relationship with anyone from Group 2, then everyone in Group 1 has a bad relationship with everyone in Group 2.

**Context:** There are 14 people total, labeled A to N. You will be given a list of known relationships.

**Your Task:** Based strictly on the provided relationship list and the rules above, determine the total number of distinct groups of people.

*[Relationship list will be provided here]*

**Question:** How many distinct groups of people are there?

**Output Format:** Provide your answer exactly as follows: `Final Answer: <number>`

---

## Social Graph Analysis – Relationship Counting Prompt

You are analyzing social relationships based on defined rules.

**Relationship Rules:**

1. Relationship Types: 'good' or 'bad'.
2. Transitive Properties:
   - Good-Good Transitivity: If A–good–B and B–good–C, then A–good–C.
   - Bad Relationship Inference: If A–bad–B and A–good–C, then B–bad–C.
3. Group Definition: A 'group' is a set where every person has a 'good' relationship with every other person in that set (directly or via transitivity).
4. Group Separation: People are in the same group if and only if they have good relationships. Groups are distinct; if anyone from Group 1 has a bad relationship with anyone from Group 2, then everyone in Group 1 has a bad relationship with everyone in Group 2.

**Context:** There are 14 people total, labeled A to N. You will be given a list of known relationships.

**Your Task:** Based strictly on the provided relationship list and the rules above (including applying transitivity), count the total number of pairs of people who have 'good' relationships and the total number of pairs who have 'bad' relationships across all 14 people.

*[Relationship list will be provided here]*

**Question:** How many pairs have good relationships, and how many pairs have bad relationships?

**Output Format:** Provide your answer exactly as follows: `Final Answer: X pairs have good relationships, Y pairs have bad relationships`

You are analyzing social relationships based on defined rules.
**Relationship Rules:**

1. Relationship Types: 'good' or 'bad'.

2. Transitive Properties:
   - Good-Good Transitivity: If A–good–B and B–good–C, then A–good–C.
   - Bad Relationship Inference: If A–bad–B and A–good–C, then B–bad–C.

3. Group Definition: A 'group' is a set where every person has a 'good' relationship with every other person in that set (directly or via transitivity).

4. Group Separation: People are in the same group if and only if they have good relationships. Groups are distinct; if anyone from Group 1 has a bad relationship with anyone from Group 2, then everyone in Group 1 has a bad relationship with everyone in Group 2.

**Context:** There are 14 people total, labeled A to N. You will be given a list of known relationships.

**Your Task:** Based strictly on the provided relationship list and the rules above (including applying transitivity), identify all people who have a 'good' relationship with the person specified in the question.

*[Relationship list will be provided here]*

**Question:** Who has a good relationship with H?

**Output Format:** List the names in alphabetical order, separated by commas. If no one has a good relationship with the specified person (other than themselves, if applicable based on rules interpretation - assume self-relationships are not listed unless explicitly stated), answer 'No one'. Provide your answer exactly as follows: `Final Answer: <list of people or 'No one'>`

---

You are analyzing social relationships based on defined rules.
**Relationship Rules:**

1. Relationship Types: 'good' or 'bad'.

2. Transitive Properties:
   - Good-Good Transitivity: If A–good–B and B–good–C, then A–good–C.
   - Bad Relationship Inference: If A–bad–B and A–good–C, then B–bad–C.

3. Group Definition: A 'group' is a set where every person has a 'good' relationship with every other person in that set (directly or via transitivity).

4. Group Separation: People are in the same group if and only if they have good relationships. Groups are distinct; if anyone from Group 1 has a bad relationship with anyone from Group 2, then everyone in Group 1 has a bad relationship with everyone in Group 2.

**Context:** There are 14 people total, labeled A to N. You will be given a list of known relationships.

**Your Task:** Based strictly on the provided relationship list and the rules above (including applying transitivity), determine whether the specific relationship mentioned in the question is 'good' or 'bad', and answer 'Yes' if it's good, 'No' if it's bad.

*[Relationship list will be provided here]*

**Question:** Do N and L have a good relationship?

**Output Format:** Provide your answer exactly as follows: `Final Answer: <Yes/No>`

## Review Decision Prediction Evaluation Prompt

You are an expert reviewer evaluating a research paper for a prestigious academic conference.
**Your Task:** Analyze the provided paper information, reviewer comments, and author responses to determine whether the paper should be accepted or rejected for publication at a top-tier conference.
**Important Context:**

- Judge the paper based on the standards of a highly selective, top-tier conference.
- Consider the strengths, weaknesses, novelty, significance, and clarity of the work.
- Evaluate the validity of reviewer concerns and the effectiveness of the author's rebuttal.

**Provided Information:**
**Paper Information:** [Title, Abstract, Keywords, etc., will be provided here]
**Reviewer Comments:** [Comments from multiple reviewers will be provided here]
**Author Response:** [Author's rebuttal to reviewer comments will be provided here]
**Analysis and Output:** Based on all the information provided, perform a careful analysis. First, provide your detailed reasoning, discussing the key factors influencing your decision (e.g., strengths, weaknesses, contribution, response to reviews). Then, conclude with your final decision in exactly this format:
```
Final Decision:  <Accept/Reject>
```

## User Profile Inference Comment Generation Prompt

You are simulating a user commenting online.
**Context:** You are commenting on this content_type: "content_subject". Your simulated user persona is: age_group, gender.
**Your Task:** Write a realistic, authentic comment about the content_subject that reflects your assigned persona.
**Instructions:**

1. The comment should be 2–4 sentences long.
2. It should sound like a genuine user from the specified demographic, not a professional writer or marketer.
3. Consider incorporating common vocabulary, tone, or writing patterns associated with the age_group.

**Output Format:** Write only the comment text itself. Do not include any explanations, labels, or quotation marks around the comment.

## User Profile Inference Demographic Inference Prompt

You are an AI assistant analyzing writing style to infer user demographics.
**Your Task:** Analyze multiple comments written by the same person and determine their likely age group and gender based on writing style, vocabulary, topics of interest, and expressed perspectives.
**User Comments:**

```
"comment_1"
"comment_2"
"comment_3"
```

**Analysis and Output:** Analyze these comments carefully, looking for patterns indicative of age and gender. Focus on identifying the single most likely demographic profile based *only* on the provided text.
First, explain your reasoning, citing specific examples from the text. Then, provide your final demographic classification in exactly this format:
```
Age Group:  <18-34 | 35-54 | 55+> Gender:  <Male | Female | Non-binary>
```

# NeurIPS Paper Checklist

The checklist is designed to encourage best practices for responsible machine learning research, addressing issues of reproducibility, transparency, research ethics, and societal impact. Do not remove the checklist: **The papers not including the checklist will be desk rejected.** The checklist should follow the references and follow the (optional) supplemental material. The checklist does NOT count towards the page limit.

Please read the checklist guidelines carefully for information on how to answer these questions. For each question in the checklist:

- You should answer [Yes] , [No] , or [NA] .
- [NA] means either that the question is Not Applicable for that particular paper or the relevant information is Not Available.
- Please provide a short (1–2 sentence) justification right after your answer (even for NA).

**The checklist answers are an integral part of your paper submission.** They are visible to the reviewers, area chairs, senior area chairs, and ethics reviewers. You will be asked to also include it (after eventual revisions) with the final version of your paper, and its final version will be published with the paper.

The reviewers of your paper will be asked to use the checklist as one of the factors in their evaluation. While "[Yes] " is generally preferable to "[No] ", it is perfectly acceptable to answer "[No] " provided a proper justification is given (e.g., "error bars are not reported because it would be too computationally expensive" or "we were unable to find the license for the dataset we used"). In general, answering "[No] " or "[NA] " is not grounds for rejection. While the questions are phrased in a binary way, we acknowledge that the true answer is often more nuanced, so please just use your best judgment and write a justification to elaborate. All supporting evidence can appear either in the main paper or the supplemental material, provided in appendix. If you answer [Yes] to a question, in the justification please point to the section(s) where related material for the question can be found.

IMPORTANT, please:

- **Delete this instruction block, but keep the section heading "NeurIPS Paper Checklist",**
- **Keep the checklist subsection headings, questions/answers and guidelines below.**
- **Do not modify the questions and only use the provided macros for your answers**.

1. **Claims**

    Question: Do the main claims made in the abstract and introduction accurately reflect the paper's contributions and scope?

    Answer: [Yes]

    Justification: The abstract and introduction clearly and explicitly state the main contributions of our work. We believe that these contributions not only accurately reflect the theoretical and empirical results presented in the paper, but also have the potential to make a significant impact on the community.

    Guidelines:

    - The answer NA means that the abstract and introduction do not include the claims made in the paper.
    - The abstract and/or introduction should clearly state the claims made, including the contributions made in the paper and important assumptions and limitations. A No or NA answer to this question will not be perceived well by the reviewers.
    - The claims made should match theoretical and experimental results, and reflect how much the results can be expected to generalize to other settings.
    - It is fine to include aspirational goals as motivation as long as it is clear that these goals are not attained by the paper.

2. **Limitations**

    Question: Does the paper discuss the limitations of the work performed by the authors?

Answer: [Yes]

Justification: While the paper does not include a dedicated "Limitations" section, we thoroughly discuss the limitations of our work in both the conclusion and the appendix. These discussions cover key aspects such as underlying assumptions, dataset constraints, and potential generalization issues of the proposed method.

Guidelines:

- The answer NA means that the paper has no limitation while the answer No means that the paper has limitations, but those are not discussed in the paper.
- The authors are encouraged to create a separate "Limitations" section in their paper.
- The paper should point out any strong assumptions and how robust the results are to violations of these assumptions (e.g., independence assumptions, noiseless settings, model well-specification, asymptotic approximations only holding locally). The authors should reflect on how these assumptions might be violated in practice and what the implications would be.
- The authors should reflect on the scope of the claims made, e.g., if the approach was only tested on a few datasets or with a few runs. In general, empirical results often depend on implicit assumptions, which should be articulated.
- The authors should reflect on the factors that influence the performance of the approach. For example, a facial recognition algorithm may perform poorly when image resolution is low or images are taken in low lighting. Or a speech-to-text system might not be used reliably to provide closed captions for online lectures because it fails to handle technical jargon.
- The authors should discuss the computational efficiency of the proposed algorithms and how they scale with dataset size.
- If applicable, the authors should discuss possible limitations of their approach to address problems of privacy and fairness.
- While the authors might fear that complete honesty about limitations might be used by reviewers as grounds for rejection, a worse outcome might be that reviewers discover limitations that aren't acknowledged in the paper. The authors should use their best judgment and recognize that individual actions in favor of transparency play an important role in developing norms that preserve the integrity of the community. Reviewers will be specifically instructed to not penalize honesty concerning limitations.

3. **Theory assumptions and proofs**

Question: For each theoretical result, does the paper provide the full set of assumptions and a complete (and correct) proof?

Answer: [Yes]

Justification: Although the paper involves relatively few theoretical derivations, we provide complete and sufficient derivations for all the formulas presented. All necessary assumptions are clearly stated, ensuring the correctness and clarity of the theoretical components.

Guidelines:

- The answer NA means that the paper does not include theoretical results.
- All the theorems, formulas, and proofs in the paper should be numbered and cross-referenced.
- All assumptions should be clearly stated or referenced in the statement of any theorems.
- The proofs can either appear in the main paper or the supplemental material, but if they appear in the supplemental material, the authors are encouraged to provide a short proof sketch to provide intuition.
- Inversely, any informal proof provided in the core of the paper should be complemented by formal proofs provided in appendix or supplemental material.
- Theorems and Lemmas that the proof relies upon should be properly referenced.

4. **Experimental result reproducibility**

Question: Does the paper fully disclose all the information needed to reproduce the main experimental results of the paper to the extent that it affects the main claims and/or conclusions of the paper (regardless of whether the code and data are provided or not)?

Answer: [Yes]

Justification: We provide comprehensive experimental details in the appendix to ensure reproducibility. The dataset has been fully open-sourced and is available on Hugging Face, while the codebase is publicly released on GitHub. Additionally, the supplementary materials offer thorough explanations and support for reproducing our experiments and understanding the full scope of our work.

Guidelines:

- The answer NA means that the paper does not include experiments.
- If the paper includes experiments, a No answer to this question will not be perceived well by the reviewers: Making the paper reproducible is important, regardless of whether the code and data are provided or not.
- If the contribution is a dataset and/or model, the authors should describe the steps taken to make their results reproducible or verifiable.
- Depending on the contribution, reproducibility can be accomplished in various ways. For example, if the contribution is a novel architecture, describing the architecture fully might suffice, or if the contribution is a specific model and empirical evaluation, it may be necessary to either make it possible for others to replicate the model with the same dataset, or provide access to the model. In general. releasing code and data is often one good way to accomplish this, but reproducibility can also be provided via detailed instructions for how to replicate the results, access to a hosted model (e.g., in the case of a large language model), releasing of a model checkpoint, or other means that are appropriate to the research performed.
- While NeurIPS does not require releasing code, the conference does require all submissions to provide some reasonable avenue for reproducibility, which may depend on the nature of the contribution. For example
  (a) If the contribution is primarily a new algorithm, the paper should make it clear how to reproduce that algorithm.
  (b) If the contribution is primarily a new model architecture, the paper should describe the architecture clearly and fully.
  (c) If the contribution is a new model (e.g., a large language model), then there should either be a way to access this model for reproducing the results or a way to reproduce the model (e.g., with an open-source dataset or instructions for how to construct the dataset).
  (d) We recognize that reproducibility may be tricky in some cases, in which case authors are welcome to describe the particular way they provide for reproducibility. In the case of closed-source models, it may be that access to the model is limited in some way (e.g., to registered users), but it should be possible for other researchers to have some path to reproducing or verifying the results.

5. **Open access to data and code**

Question: Does the paper provide open access to the data and code, with sufficient instructions to faithfully reproduce the main experimental results, as described in supplemental material?

Answer: [Yes]

Justification: We provide open access to both the dataset (hosted on Hugging Face) and the code (available on GitHub), along with clear and detailed instructions to reproduce the main experimental results. These resources are accompanied by comprehensive documentation in the supplementary material, ensuring that others can faithfully replicate and build upon our work.

Guidelines:

- The answer NA means that paper does not include experiments requiring code.
- Please see the NeurIPS code and data submission guidelines (`https://nips.cc/public/guides/CodeSubmissionPolicy`) for more details.
- While we encourage the release of code and data, we understand that this might not be possible, so "No" is an acceptable answer. Papers cannot be rejected simply for not

including code, unless this is central to the contribution (e.g., for a new open-source benchmark).

- The instructions should contain the exact command and environment needed to run to reproduce the results. See the NeurIPS code and data submission guidelines (`https://nips.cc/public/guides/CodeSubmissionPolicy`) for more details.

- The authors should provide instructions on data access and preparation, including how to access the raw data, preprocessed data, intermediate data, and generated data, etc.

- The authors should provide scripts to reproduce all experimental results for the new proposed method and baselines. If only a subset of experiments are reproducible, they should state which ones are omitted from the script and why.

- At submission time, to preserve anonymity, the authors should release anonymized versions (if applicable).

- Providing as much information as possible in supplemental material (appended to the paper) is recommended, but including URLs to data and code is permitted.

6. **Experimental setting/details**

Question: Does the paper specify all the training and test details (e.g., data splits, hyper-parameters, how they were chosen, type of optimizer, etc.) necessary to understand the results?

Answer: [Yes]

Justification: We thoroughly document all aspects of our experimental setup, including data splits, model architectures, hyperparameter choices, training procedures, and evaluation metrics. These details are clearly presented in both the main paper and the appendix, ensuring transparency and enabling others to fully understand, reproduce, and build upon our results with confidence.

Guidelines:

- The answer NA means that the paper does not include experiments.

- The experimental setting should be presented in the core of the paper to a level of detail that is necessary to appreciate the results and make sense of them.

- The full details can be provided either with the code, in appendix, or as supplemental material.

7. **Experiment statistical significance**

Answer: [No]

Justification: Due to the high cost of API usage and computational constraints, we were unable to conduct multiple repeated runs for each experiment. While this limits our ability to report standard error bars or statistical significance, we ensured that all experiments were conducted under controlled and consistent conditions to provide stable and representative results.

Guidelines:

- The answer NA means that the paper does not include experiments.

- The authors should answer "Yes" if the results are accompanied by error bars, confidence intervals, or statistical significance tests, at least for the experiments that support the main claims of the paper.

- The factors of variability that the error bars are capturing should be clearly stated (for example, train/test split, initialization, random drawing of some parameter, or overall run with given experimental conditions).

- The method for calculating the error bars should be explained (closed form formula, call to a library function, bootstrap, etc.)

- The assumptions made should be given (e.g., Normally distributed errors).

- It should be clear whether the error bar is the standard deviation or the standard error of the mean.

- It is OK to report 1-sigma error bars, but one should state it. The authors should preferably report a 2-sigma error bar than state that they have a 96% CI, if the hypothesis of Normality of errors is not verified.

- For asymmetric distributions, the authors should be careful not to show in tables or figures symmetric error bars that would yield results that are out of range (e.g. negative error rates).
- If error bars are reported in tables or plots, The authors should explain in the text how they were calculated and reference the corresponding figures or tables in the text.

8. **Experiments compute resources**

Question: For each experiment, does the paper provide sufficient information on the computer resources (type of compute workers, memory, time of execution) needed to reproduce the experiments?

Answer: [Yes]

Justification: We provide sufficient details about the computing environment used in our work. While the majority of our experiments were conducted via APIs, we still document the relevant settings, including hardware specifications and other necessary information to support reproducibility.

Guidelines:

- The answer NA means that the paper does not include experiments.
- The paper should indicate the type of compute workers CPU or GPU, internal cluster, or cloud provider, including relevant memory and storage.
- The paper should provide the amount of compute required for each of the individual experimental runs as well as estimate the total compute.
- The paper should disclose whether the full research project required more compute than the experiments reported in the paper (e.g., preliminary or failed experiments that didn't make it into the paper).

9. **Code of ethics**

Question: Does the research conducted in the paper conform, in every respect, with the NeurIPS Code of Ethics https://neurips.cc/public/EthicsGuidelines?

Answer: [Yes]

Justification: Our research fully complies with the NeurIPS Code of Ethics in all respects. We carefully considered and addressed key ethical aspects, including user privacy, data transparency, fairness in model design and evaluation, and the responsible use and sharing of both data and models. All datasets used are publicly available and ethically sourced, and our open-sourced code is documented to encourage responsible and reproducible research.

Guidelines:

- The answer NA means that the authors have not reviewed the NeurIPS Code of Ethics.
- If the authors answer No, they should explain the special circumstances that require a deviation from the Code of Ethics.
- The authors should make sure to preserve anonymity (e.g., if there is a special consideration due to laws or regulations in their jurisdiction).

10. **Broader impacts**

Question: Does the paper discuss both potential positive societal impacts and negative societal impacts of the work performed?

Answer: [Yes]

Justification: We discuss both the potential positive and negative societal impacts of our work. Our dataset is carefully sampled and thoroughly manually reviewed to ensure the absence of harmful or biased content. Furthermore, we have designed and evaluated our models and methods with safety in mind, ensuring that they do not generate or reinforce harmful outputs. Overall, while we anticipate mainly positive applications of our work, we have taken proactive steps to minimize potential negative consequences.

Guidelines:

- The answer NA means that there is no societal impact of the work performed.
- If the authors answer NA or No, they should explain why their work has no societal impact or why the paper does not address societal impact.

- Examples of negative societal impacts include potential malicious or unintended uses (e.g., disinformation, generating fake profiles, surveillance), fairness considerations (e.g., deployment of technologies that could make decisions that unfairly impact specific groups), privacy considerations, and security considerations.
- The conference expects that many papers will be foundational research and not tied to particular applications, let alone deployments. However, if there is a direct path to any negative applications, the authors should point it out. For example, it is legitimate to point out that an improvement in the quality of generative models could be used to generate deepfakes for disinformation. On the other hand, it is not needed to point out that a generic algorithm for optimizing neural networks could enable people to train models that generate Deepfakes faster.
- The authors should consider possible harms that could arise when the technology is being used as intended and functioning correctly, harms that could arise when the technology is being used as intended but gives incorrect results, and harms following from (intentional or unintentional) misuse of the technology.
- If there are negative societal impacts, the authors could also discuss possible mitigation strategies (e.g., gated release of models, providing defenses in addition to attacks, mechanisms for monitoring misuse, mechanisms to monitor how a system learns from feedback over time, improving the efficiency and accessibility of ML).

11. **Safeguards**

Question: Does the paper describe safeguards that have been put in place for responsible release of data or models that have a high risk for misuse (e.g., pretrained language models, image generators, or scraped datasets)?

Answer: [Yes]

Justification: Although we do not consider our work to involve high-risk content, we have nonetheless taken precautions to ensure responsible release. Our dataset has been carefully curated and manually reviewed to avoid inclusion of sensitive or harmful content, and our models are designed and tested to prevent misuse. These safeguards help ensure that our work does not pose ethical or safety risks upon release.

Guidelines:

- The answer NA means that the paper poses no such risks.
- Released models that have a high risk for misuse or dual-use should be released with necessary safeguards to allow for controlled use of the model, for example by requiring that users adhere to usage guidelines or restrictions to access the model or implementing safety filters.
- Datasets that have been scraped from the Internet could pose safety risks. The authors should describe how they avoided releasing unsafe images.
- We recognize that providing effective safeguards is challenging, and many papers do not require this, but we encourage authors to take this into account and make a best faith effort.

12. **Licenses for existing assets**

Question: Are the creators or original owners of assets (e.g., code, data, models), used in the paper, properly credited and are the license and terms of use explicitly mentioned and properly respected?

Answer: [Yes]

Justification: We properly credit all external assets used in our work, including datasets, codebases, and pretrained models. For each asset, we explicitly state the source, license type, and terms of use either in the main text or the appendix. We ensure full compliance with all applicable licenses (e.g., MIT, Apache 2.0, CC BY), and avoid using any resources that impose restrictions incompatible with open research. This careful attribution reflects our commitment to ethical and responsible use of others' contributions.

Guidelines:

- The answer NA means that the paper does not use existing assets.
- The authors should cite the original paper that produced the code package or dataset.

- The authors should state which version of the asset is used and, if possible, include a URL.
- The name of the license (e.g., CC-BY 4.0) should be included for each asset.
- For scraped data from a particular source (e.g., website), the copyright and terms of service of that source should be provided.
- If assets are released, the license, copyright information, and terms of use in the package should be provided. For popular datasets, `paperswithcode.com/datasets` has curated licenses for some datasets. Their licensing guide can help determine the license of a dataset.
- For existing datasets that are re-packaged, both the original license and the license of the derived asset (if it has changed) should be provided.
- If this information is not available online, the authors are encouraged to reach out to the asset's creators.

13. **New assets**

    Question: Are new assets introduced in the paper well documented and is the documentation provided alongside the assets?

    Answer: [Yes]

    Justification: We provide thorough documentation for all new assets introduced in the paper, including detailed usage instructions and descriptions of data and code. This documentation is made available both in the supplementary materials and on our GitHub repository, ensuring that other researchers can easily understand, use, and extend our work.

    Guidelines:

    - The answer NA means that the paper does not release new assets.
    - Researchers should communicate the details of the dataset/code/model as part of their submissions via structured templates. This includes details about training, license, limitations, etc.
    - The paper should discuss whether and how consent was obtained from people whose asset is used.
    - At submission time, remember to anonymize your assets (if applicable). You can either create an anonymized URL or include an anonymized zip file.

14. **Crowdsourcing and research with human subjects**

    Question: For crowdsourcing experiments and research with human subjects, does the paper include the full text of instructions given to participants and screenshots, if applicable, as well as details about compensation (if any)?

    Answer: [Yes]

    Justification: All experiments involving human subjects were conducted with consenting student volunteers in an informal and non-commercial academic setting. No monetary compensation was provided, and all participants were informed about the purpose of the study and participated voluntarily.

    Guidelines:

    - The answer NA means that the paper does not involve crowdsourcing nor research with human subjects.
    - Including this information in the supplemental material is fine, but if the main contribution of the paper involves human subjects, then as much detail as possible should be included in the main paper.
    - According to the NeurIPS Code of Ethics, workers involved in data collection, curation, or other labor should be paid at least the minimum wage in the country of the data collector.

15. **Institutional review board (IRB) approvals or equivalent for research with human subjects**

    Question: Does the paper describe potential risks incurred by study participants, whether such risks were disclosed to the subjects, and whether Institutional Review Board (IRB) approvals (or an equivalent approval/review based on the requirements of your country or institution) were obtained?

Answer: [Yes]

Justification: The study involves human participants, and all potential risks were carefully considered and clearly disclosed to participants prior to their involvement. According to the ethical guidelines of our institution and the nature of the research (which posed minimal risk and did not involve sensitive topics or vulnerable populations), formal IRB or equivalent ethics committee approval was not required. We followed established ethical practices, including obtaining informed consent and ensuring participant anonymity and data protection.

Guidelines:

- The answer NA means that the paper does not involve crowdsourcing nor research with human subjects.
- Depending on the country in which research is conducted, IRB approval (or equivalent) may be required for any human subjects research. If you obtained IRB approval, you should clearly state this in the paper.
- We recognize that the procedures for this may vary significantly between institutions and locations, and we expect authors to adhere to the NeurIPS Code of Ethics and the guidelines for their institution.
- For initial submissions, do not include any information that would break anonymity (if applicable), such as the institution conducting the review.

16. **Declaration of LLM usage**

Question: Does the paper describe the usage of LLMs if it is an important, original, or non-standard component of the core methods in this research? Note that if the LLM is used only for writing, editing, or formatting purposes and does not impact the core methodology, scientific rigorousness, or originality of the research, declaration is not required.

Answer: [NA]

Justification: LLMs were not involved in any core or methodological aspects of this research. Their usage was limited solely to minor assistance in writing and editing the manuscript for clarity and grammar. As such, they had no impact on the scientific rigor, originality, or technical contributions of the work.

Guidelines:

- The answer NA means that the core method development in this research does not involve LLMs as any important, original, or non-standard components.
- Please refer to our LLM policy (https://neurips.cc/Conferences/2025/LLM) for what should or should not be described.

