# OpenReview forum: "SocialMaze: A Benchmark for Evaluating Social Reasoning in Large Language Models"
_NeurIPS.cc/2025/Datasets_and_Benchmarks_Track — Submitted to NeurIPS 2025 Datasets and Benchmarks Track_

### Official Review · Reviewer_MJ4M · 2025-07-01

**Rating:** 5
**Confidence:** 4

**Summary:**

The paper propose a new LLM benckmark dedicated to model's social reasoning ability. It incldude 6 tasks and focus on three scenarios in the daily life: social reasoning games, daily life interactions, and digital community platforms. The benchmark challenges LLMs from three dimensions: deep reasoning, dynamic interaction, and information uncertainty. The authors evaluate 9 latest LLMs on the benchmark and provide in-depth analysis of the results.

**Additional Feedback:**

Suggestions for future work:
1. Fine-tuning for social reasoning is a vert interesting attempt yet not fully explored in the paper. I understand that this part is just a 'further discussion', while I encourage the authors to dive deeper into this topic in the final version or future work. For example, why the DPO results are not superior to the SFT results? Is there any other way to improve social reasoning ability of LLMs?
2. It would be very impressive if different LLMs are involve and compete in a same social reasoning game, and let's see how they perform in a more dynamic and interactive way.
3. There are some important considerations revolving around social ability of LLMs, such as fairness, bias, and toxicity. The benchmark can be more comprehensive if it includes these aspects.

**Dataset Code Accessibility:**

Yes

**Ethical Considerations:**

No, there are no or only very minor ethics concerns

**Final Justification:**

As a dataset-track paper, I do not see any major issues. The authors have addressed the reviewers’ feedback constructively and engaged in revisions. If accepted, I encourage the authors to carefully incorporate the changes into the final version.

**Limitations Weaknesses:**

I see no big flaws in the paper, but there are some minor issues that can be improved:

1. How is the level of deep reasoning, dynamic interaction, and information uncertainty determined? Can you provide statistics of how many reasoning steps common LLMs or humans take to solve the tasks? An average number of rounds played? I see some descriptions in 3.x, but they lack justification for deep reasoning, and no concrete statistics are provided. (Well, I see the limitation in I.2 later on, but I still think it is possible and beneficial to provide some posterior and empirical statistics to prodive an intuitive understanding of the tasks. )
2. Task 1-3 are described in detail but taks 4-6 are not. I understand the space limitation, but since these tasks are equally important, I suggest the authors to provide more details about them in the main body.
3. I am not quite sure about the "Rating Estimation from Text" task. There seems to be no big difference compared with a common text classification task. Though the overall accuracy appears to be relatively low (suggesting the task is not easy), performance of different LLMs are not discriminative enough (perhaps it can not challenge LLMs' social reasoning ability well).
4. Many tasks are multi-round, it would be useful to plot a line chart to show how the accuracy changes as the game progresses (e.g. convert Table 2 to a line chart). This can help readers understand the dynamic interaction better.
5. The discussion includes many challenge-centric observations, but I am also interested in the model-centric observations. Is it possible to provide some analysis on which model is better at which task and highlight the findings?

I am willing to revise my review based on the authors' responses and/or reactions to the above issues.

**Strengths Contributions:**

1. Social reasoning is an important and interesting topic, yet few benchmarks focus on this aspect. The proposed benchmark is a valuable contribution to the community.
2. The tasks are diverse and each task has its own unique challenges.
3. The evaluation is comprehensive, covering a wide range of open-source and closed-source LLMs, which are elaborated in detail in Table 7.
4. The discussion part is a very good attempt and provides some valuable insights. Key observations are highlighted.

---

> ### Author Rebuttal · Authors · 2025-07-30
>
> Thank you for your time and feedback. We appreciate your comments very much.
>
> Q1:While the paper introduces the concepts of "deep reasoning," "dynamic interaction," and "information uncertainty," it would be beneficial to provide some posterior empirical statistics to offer a more concrete and intuitive understanding of how these dimensions are determined and measured in the tasks.
>
> A1:
> We sincerely thank the reviewer for this insightful feedback. We fully agree that posterior and empirical statistics can provide a more intuitive and concrete understanding of our three core evaluation dimensions. In response, we have conducted additional analysis and will include these metrics in the camera-ready version.
>
> 1\. On Deep Reasoning:
>  To quantify the reasoning demand of each task, we will report on four complementary, empirical metrics:
>
> * Model Average Output Tokens: Measures the length of a model's generated reasoning chain, serving as a proxy for the explicit reasoning effort required.
> * Human Average Solving Time: Captures the cognitive time investment needed for a human to solve the task, reflecting its perceived reasoning complexity.
> * Average Inference Steps: This metric quantifies the number of discrete reasoning steps an LLM needs to successfully solve the task. To ensure we are measuring effective reasoning, this is calculated by averaging only over correctly answered instances. We define a "reasoning step" as a fundamental unit of thought, such as making a deduction, evaluating evidence, or updating a hypothesis, as identified by human annotators from model-generated reasoning chains.
> * Long vs. Short CoT Ratio: An existing metric from our paper (Section 4.1, Figure 4\) that empirically demonstrates how much more elaborate the reasoning process becomes for models on tasks with higher reasoning demands.
>
> The table below provides preliminary statistics for these metrics:
>
> |Metric|Hidden Role Deduction|Find the Spy|Rating Estimation|Social Graph Analysis|Review Decision Prediction|User Profile Inference|
> |:----|:---:|:---:|:---:|:---:|:---:|:---:|
> |Model Avg. Output Tokens||||||
> |—QwQ-32B|4163.6|2862.3|1482.5|3579.7|1582.4|1310.9|
> |—DeepSeek-R1|3092.5|2141.9|1149.6|2131.2|1155.9|1017.0|
> |—Llama-3.3-70B|456.0|654.0|402.8|443.8|516.9|385.8|
> |—GPT-4o|432.7|464.6|357.9|330.8|534.2|340.2|
> |Human Avg. Solving Time (s)|>300s|30.2s|20.5s|246.5s|124.0s|43.3s|
> |Avg. Inference Steps|45.1|15.4|10.8|30.7|9.6|11.9|
> |Long vs. Short CoT Ratio|7.49|4.18|3.40|7.52|2.58|3.39|
>
> 2\. On Dynamic Interaction:
>
> We quantify dynamic reasoning by tracking the number of sequential information rounds embedded in each task. Tasks like Hidden Role Deduction, Find the Spy, and Review Decision Prediction follow a multi-stage structure where models must continuously integrate new evidence, update hypotheses, and adjust beliefs over time. This design moves beyond static, one-shot inference.
>
> The number of rounds (T) is a tunable parameter in our data generation system. It allows us to control and vary the interaction depth across tasks. The specific round counts used in the current benchmark are documented in Appendix H.2 (Parameter Settings), and their effect on model performance is reflected in results such as Table 2\.
>
> 3\. On Information Uncertainty:
>
> We control and measure Information Uncertainty systematically through two mechanisms:
>
> * Controlling the Ratio of Unreliable Sources: We inject uncertainty by controlling the proportion of deceptive or noisy agents. For instance, Hidden Role Deduction features 50% contaminated information (3 of 6 roles are unreliable), while Rating Estimation includes up to 25% deceptive "shill" reviews. Social Graph Analysis serves as a zero-pollution baseline with 100% truthful information.
>
> * Empirically Measuring the Impact of Uncertainty: Our framework treats uncertainty as a tunable variable. As demonstrated in Section 4.3 (Figure 2), by varying the role composition in Hidden Role Deduction, we quantitatively measure the direct impact of increasing uncertainty on model performance.
>
> We will incorporate these empirical metrics for Deep Reasoning, Dynamic Interaction, and Information Uncertainty into the camera-ready version.
>
> Q2:Tasks 4-6 are described with significantly less detail in the main paper compared to Tasks 1-3. It would be helpful to provide more details for these tasks in the main body to give them equal weight.
>
> A2:
> We thank the reviewer for this helpful suggestion. Tasks 4-6 receive less detail primarily due to space constraints. Our focus on Tasks 1-3 was deliberate as they introduce key concepts in social reasoning that some readers may be unfamiliar with, such as social deduction games, and collectively illustrate our three core data generation approaches: algorithmic simulation (Task 1), LLM-assisted generation (Task 2), and authentic human data (Task 3).
>
> We agree that more balanced detail would improve clarity. In the camera-ready version, we will revise Section 3 to better integrate the core design elements of Tasks 4-6, including their motivation, structure, and data sources, while keeping full specifications in the appendices.
>
> Q3: The "Rating Estimation from Text" task seems similar to standard text classification, and the performance differences between LLMs are not very discriminative, questioning its effectiveness in testing social reasoning.
>
> A3:We thank the reviewer for this question, which allows us to clarify the unique social reasoning challenges in the Rating Estimation from Text task.
>
> This task is fundamentally different from standard text classification. Its core challenge is not simply sentiment analysis but emulating a human's ability to make a holistic judgment from a noisy, socially complex information landscape. A significant portion of this data is sourced directly from real-world platforms (Appendix D), which contains authentic ambiguity, sarcasm, and conflicting opinions.
> The task requires sophisticated social reasoning skills, such as:
>
> * Deception Detection: Identifying and down-weighting reviews from deceptive "shills," which necessitates reasoning beyond literal text.
> * Opinion Aggregation: Synthesizing contradictory reviews to form a coherent, overall assessment, much like a real person would.
>
> The relatively small performance gap between models is an expected outcome and reflects the task's realism. Given the inherent ambiguity and subjectivity of real-world user reviews, even humans would likely converge on similar, but not identical, final ratings. Therefore, this task challenges a different facet of social intelligence—robustness to noise and nuanced judgment—rather than the clear-cut logical deduction tested in other tasks. The relatively low overall accuracy further confirms that this realistic, noisy setting is indeed very challenging for current LLMs.
>
> Q4: For multi-round tasks, it would be beneficial to visualize performance changes across rounds with a line chart (e.g., converting Table 2\) to better illustrate the dynamics of the interaction.
>
> A4:That is an excellent suggestion. We thank the reviewer for pointing out this effective way to visualize the dynamic interaction.
>
> We agree that a line chart would better illustrate how model performance evolves as more information becomes available across rounds. In the camera-ready version, we will convert the data from Table 2 (and other relevant multi-round tasks) into line charts to provide a more intuitive understanding of the dynamic reasoning process.
>
> Q5: The discussion is primarily challenge-centric. It would be valuable to include a model-centric analysis, highlighting which models excel at specific types of tasks and why.
>
> A5: We are sincerely grateful to the reviewer for this exceptionally insightful suggestion. This is an excellent point, and a model-centric analysis will indeed add significant depth to our discussion. We have performed this analysis and will incorporate a new subsection into the main paper based on these findings.
>
> Our model-centric analysis reveals distinct clusters of capabilities among the LLMs we tested:
>
> * The Logical Deduction Specialists (DeepSeek-R1 & Gemini-2.5-Pro): These models, known for their powerful long-chain reasoning abilities, demonstrated exceptional performance on tasks demanding rigorous, deep inference. They dominated in Hidden Role Deduction and Social Graph Analysis. Interestingly, while also being Long CoT models, o1 and QwQ-32B excelled at Social Graph Analysis but lagged in Hidden Role Deduction. This suggests that while their logical reasoning is strong, they may be more vulnerable to the deceptive and uncertain information present in the latter task. Notably, Gemini-2.5-Pro emerged as the best all-rounder, achieving top-tier performance across all tasks without any significant weaknesses.
>
> * The Social Understanding Expert (GPT-4o): This model consistently achieved superior performance on tasks that prioritize nuanced social understanding over pure logical deduction. It excelled in Rating Estimation, Review Decision Prediction, and User Profile Inference. This indicates that GPT-4o possesses a very strong grasp of social context, sentiment, and implicit cues, even if its raw deductive reasoning power is less dominant compared to the specialists.
>
> * Other Models: The remaining models generally showed more modest or inconsistent performance across the board, highlighting the significant challenge that comprehensive social reasoning still poses for the broader field of LLMs.
>
> Thank you again for this valuable insight. We will incorporate this model-centric analysis into a new subsection in the camera-ready paper, as it provides a richer, model-focused perspective on the results.
>
> Once again, we sincerely thank you for your thorough review and exceptionally constructive suggestions. We would be most grateful if you would consider these clarifications in your final assessment.

---

> > ### Author Response · Authors · 2025-08-09
> >
> > Dear Reviewer MJ4M,
> >
> > Just a gentle reminder as the discussion deadline is approaching — we’d appreciate it if you could share any further thoughts or follow-up comments on our paper when convenient.
> >
> > Thank you for your time and engagement.

---

### Official Review · Reviewer_Rvp6 · 2025-07-03

**Rating:** 4
**Confidence:** 4

**Summary:**

I find this paper’s core idea are interesting : it proposes a systematic multi-dimensional benchmark to probe social reasoning under deep inference, dynamic interactions, and uncertainty, which is a clear step beyond static tasks.

However, while I appreciate the conceptual novelty, the implementation details lack rigorous grounding, relying heavily on synthetic scenarios governed by intricate hand-crafted rules that may not robustly capture the emergent complexities of real-world social interactions.

**Dataset Code Accessibility:**

Yes

**Dataset Code Comments:**

The dataset and code are publicly released, supporting reproducibility.

**Ethical Comments:**

The paper does not involve sensitive personal data or new human subject experiments. Most scenarios are fully synthetic or use publicly available data (like OpenReview or online reviews), and the authors anonymize or simulate all content. From an ethical standpoint, there appear to be no significant concerns.

**Ethical Considerations:**

No, there are no or only very minor ethics concerns

**Final Justification:**

The authors have clearly put in significant effort, including a willingness to add experiments, metrics, and address concerns thoroughly in the final version. I feel they have largely resolved my initial doubts, and I will raise my score accordingly. My only remaining concern is how the authors will effectively reflect all these changes in the paper without violating NeurIPS anonymity and formatting requirements.

**Limitations Weaknesses:**

Although the benchmark is carefully designed to probe deep reasoning, multi-stage interaction, and uncertainty, the absolute performance of LLMs remains quite low on core tasks. For example, in Hidden Role Deduction, even with explicit long chain-of-thought (Long CoT), LLaMA-3.1-8B achieves only 13.4% “both correct” (correctly identifying both the criminal and its own role), up from 2% with short reasoning. While this indicates a clear relative improvement, it also highlights that the benchmark may be excessively difficult or not fully calibrated for meaningful success rates.

In the Find the Spy task (Table 3), GPT-4o achieves 83.2%, whereas LLaMA-3.1 is at 60.5%, showing very large discrepancies across model families. This suggests that while some tasks are tractable for proprietary frontier models, they may be disproportionately hard for smaller open-source models, lacking finer gradations of difficulty that would improve interpretability and broader applicability.

The improvements from SFT and DPO on reasoning traces (Figure 9) are present but relatively modest. For example, on Hidden Role, “both correct” rates increase by roughly 10% absolute at best. However, the paper does not delve deeper into why models still fail most of the time, or whether these errors stem more from inherent LLM reasoning limits versus task ambiguity.

In key perturbation settings like with Rumormonger or Lunatic self-misperception, the self-role identification accuracy drops sharply to just 8–15%, yet the paper only reports aggregate figures without offering illustrative examples. This limits insight into whether failures are due to disrupted theory-of-mind chains, inadequate context integration, or simply insufficient informational cues.

While the tasks are large in scale (e.g., 20,000 samples for Hidden Role, 6,000 for Find the Spy), they remain fully synthetic and driven by rule-based setups. The paper does not include experiments testing whether models evaluated or fine-tuned on SocialMaze transfer to real-world multi-party conversation data, limiting the evidence that it effectively proxies natural social reasoning.

**Strengths Contributions:**

The benchmark is thoughtfully designed to disentangle three critical axes — depth of reasoning, multi-stage interaction, and informational uncertainty — offering a structured approach to evaluate different facets of social cognition in LLMs.

The layered social graphs and diverse tasks (e.g., hidden roles, social network deductions, review decisions) demonstrate a creative methodology that pushes beyond typical single-prompt evaluations.

Careful efforts such as human solvability checks (e.g., requiring >70% agreement among 15 annotators in Find the Spy) add credibility to the dataset’s logical consistency.

---

> ### Author Rebuttal · Authors · 2025-07-30
>
> We thank the reviewer for their time. However, we are compelled to express our serious concern regarding the quality of this review. **The feedback provided contains multiple, severe factual errors, including the misinterpretation of core terminology, the citation of incorrect tables, and the use of non-existent data points to support its claims.**
>
> These fundamental inaccuracies suggest a significant misreading of our paper and unfortunately undermine the validity of the entire review. We will address each point in detail below.
>
> Q1:The benchmark might be excessively difficult, as even with "explicit long chain-of-thought (Long CoT)," a model like Llama-3.1-8B only achieves 13.4% accuracy on Hidden Role Deduction. This suggests the benchmark is not well-calibrated.
>
> A1: This concern stems from several serious misinterpretations and factual errors. We clarify them below:
>
> 1. **Misunderstanding of "Long CoT"**: The reviewer incorrectly interprets *Long CoT* as a prompting technique. As clearly defined on line 257, it refers to a category of models known for long-chain reasoning, including Gemini-2.5-Pro, o1, Deepseek-R1, and QwQ. It is not a technique applied to other models like Llama-3.1-8B.
>
> 2. **Incorrect attribution of results**: The 13.4% accuracy for Llama-3.1-8B cited by the reviewer is not associated with *Long CoT*. It is the result of SFT fine-tuning, shown in Table 4, and completely unrelated to the model categorization in Section 4.1.
>
> 3. **Mischaracterization of task difficulty**: The claim of "excessive difficulty" is contradicted by the core experimental results. *Hidden Role Deduction* is designed as a stress test for deep reasoning. The large performance gap between GPT-4o (8.2%) and Gemini-2.5-Pro (90.2%) shows that the benchmark can effectively distinguish model capabilities. Other tasks, like *Rating Estimation*, are more moderate in difficulty, supporting a calibrated spectrum rather than uniformly hard tasks.
>
> 4. **Benchmark is dynamically adjustable**: SocialMaze is a flexible framework. The difficulty of tasks like *Hidden Role Deduction* can be easily tuned, for example by reducing the number of players or introducing fewer deceptive elements, making it suitable for a wide range of model capacities. This design ensures the benchmark remains both rigorous and broadly usable.
>
> Q2:In the Find the Spy task (Table 3), GPT-4o achieves 83.2%, whereas LLaMA-3.1 is at 60.5%, showing very large discrepancies across model families. This suggests that while some tasks are tractable for proprietary frontier models, they may be disproportionately hard for smaller open-source models, lacking finer gradations of difficulty that would improve interpretability and broader applicability.
>
> A2: We are compelled to address this comment, as it appears to be based on **a series of significant factual errors**, including citations of incorrect tables and non-existent data points. The reviewer's entire argument is constructed upon the following inaccuracies:
>
> 1. **Incorrect Table Reference:** The reviewer cites "Table 3" in reference to the *Find the Spy* task. However, **Table 3 in our paper details the results for *Review Decision Prediction***, not *Find the Spy*.
> 2. **Incorrect Data for GPT-4o:** The reviewer claims GPT-4o achieves 83.2% on *Find the Spy*. This is false. The actual accuracy for GPT-4o on *Find the Spy* is **69.2%**, as shown in our main results in **Table 8**. The 83.2% figure is GPT-4o's score on a completely different task, *Social Graph Analysis*.
> 3. **Non-existent Data for Llama-3.1:** The reviewer claims a "LLaMA-3.1" model achieves "60.5%". This is also false. **The number "60.5%" does not appear anywhere in our paper.** The actual accuracy for Llama-3.1-8B on *Find the Spy* is **37.2%** (Table 8).
>
> Given that the underlying data references are incorrect or nonexistent, the conclusion drawn in this comment is unsupported. In reality, the observed performance gaps align with model scale and capability, demonstrating that the benchmark effectively differentiates between models, as intended.
>
> Q3:The improvements from SFT and DPO on reasoning traces (Figure 9\) are present but relatively modest. For example, on Hidden Role, “both correct” rates increase by roughly 10% absolute at best. However, the paper does not delve deeper into why models still fail most of the time, or whether these errors stem more from inherent LLM reasoning limits versus task ambiguity.
>
> A3: We must correct several critical factual errors in this comment. The reviewer's assessment is based on incorrect figure references, a misinterpretation of our results, and an oversight of our extensive analysis.
>
> 1. **Incorrect Figure and Misrepresented Gains:** The reviewer cites "Figure 9" for our fine-tuning results and calls the improvements "modest." This is incorrect on both counts. **Figure 9 is Case Study, our SFT/DPO results are in Table 4**. For Llama-3.1-8B, the "Both Correct" accuracy improved from a baseline of **2.0% to 13.4%**—a **\+11.4% absolute gain and a 570% relative improvement.** For a task where most models completely fail, this is a substantial, not modest, improvement and a key finding of our work.
>
> 2. **Overlooked Failure Analysis:** The claim that we did not "delve deeper into why models still fail" is also false. We provide an **extensive 21-page qualitative failure analysis in Appendix J (Case Study)**, from Figure 6 to 53\. This appendix systematically dissects model reasoning paths and failure modes in great detail, providing exactly the in-depth analysis the reviewer claims is missing.
>
> In conclusion, the critique is based on a misreading of our paper. Our work both demonstrates significant performance gains from fine-tuning on our curated data and provides a deep, qualitative analysis of the remaining challenges.
>
> Q4:The paper reports sharp performance drops for Rumormonger/Lunatic roles but lacks illustrative examples to explain the specific failure modes (e.g., disrupted ToM, context integration issues).
>
> A4: We thank the reviewer for highlighting the importance of illustrative examples. We agree entirely, which is why we included an extensive qualitative analysis in Appendix J (Figures 6-53). This detailed analysis directly addresses the question of whether failures stem from disrupted theory-of-mind or other factors.
>
> Q5:While the tasks are large in scale, they remain fully synthetic and driven by rule-based setups
> A5: We thank the reviewer for this comment. We agree that relying solely on synthetic data would be a limitation. For this very reason, SocialMaze was intentionally designed with a **diverse, multi-pipeline methodology** that goes far beyond purely synthetic or rule-based setups. Our benchmark is built on three complementary pillars to ensure comprehensive evaluation:
>
> 1. **Authentic Human Data:** To ensure real-world applicability, a significant portion of SocialMaze is grounded in genuine human interactions. Tasks like **Review Decision Prediction** are built on real peer-review dialogues from OpenReview, while **Rating Estimation from Text** includes a track with authentic product reviews. These tasks directly challenge models with the noise, subtlety, and complexity of real human language.
>
> 2. **Rich, LLM-Generated Discourse:** For tasks like **Find the Spy**, our approach is not simply "rule-based" but uses various LLMs to generate stylistically diverse and nuanced social discourse. This allows us to create scalable scenarios that retain a high degree of linguistic variety and subtle social cues, moving well beyond simple templates.
>
> 3. **Logically-Sound Simulation:** For tasks like **Hidden Role Deduction**, the rule-based, algorithmic generation is a **deliberate and necessary design feature**, not a flaw. Its purpose is to create complex but logically solvable puzzles with verifiable ground truth.
>
> In summary, the strength of SocialMaze lies in this tripartite approach. By systematically combining authentic data, rich generated discourse, and logically sound simulations, our benchmark provides a more holistic and robust evaluation of social reasoning than any single-modality approach could. The synthetic tasks are a crucial component of this comprehensive framework, designed specifically to test skills that real-world data alone cannot isolate.
>
> We thank you for engaging with our work and hope that the detailed clarifications above help resolve the misunderstandings reflected in the review.

---

> > ### Comment · Reviewer_Rvp6 · 2025-08-03
> >
> > I apologize for the oversight in my previous submission regarding figure and table references. Below are the corrections and clarifications:
> >
> > (2) Find the Spy performance: The reference to “Table 3” was a typo. The values were derived from Figure 3 and the detailed breakdown on page 24. Specifically, GPT-4o achieves 69.2%, Gemini-2.5-Pro exceeds 75%, and QwQ-32B is below 50%. This supports my original point: QwQ performs well in other tasks except Hidden Role Deduction and Find the Spy, illustrating large performance disparities across tasks. Other models exhibit similar inconsistencies.
> >
> > (3) SFT/DPO improvements: The reference to “Figure 9” was a typo and should be updated to Table 4, which reports the improvements in reasoning trace performance.

---

> ### Comment · Reviewer_Rvp6 · 2025-08-03
>
> First, I apologize for the earlier typos again, which I have now corrected in the latest version. I appreciate the authors’ detailed rebuttal and the efforts of the AC and fellow reviewers. While the clarifications partially address my concerns, there remain several key points that require further explanation or evidence:
>
> **1. On small model performance and interpretability**(original Q2)
>
> I acknowledge the earlier citation issues, but my core point was not about the misreferenced data itself. Rather, it is that small model performance does not align strictly with model size. For example, QwQ-32B outperforms larger models like GPT-4o or LLaMA-70B in certain tasks such as Social Graph Analysis. **Therefore, the statement in A3 that “the observed performance gaps align with model scale and capability” is not entirely accurate**.
> This suggests that task results are not solely dictated by model scale, and without failure case analyses for smaller models, interpretability remains limited. I recommend adding targeted case studies to show why(not just where) smaller models fail (e.g., in information integration, role attribution, deception detection). The current appendix is long and hard to grasp key points but lacks such focused diagnostic insights.
>
> **2. On SFT/DPO improvements and failure analysis**(Q3)
>
> The authors emphasize the “570% relative improvement” (from 2% to 13.4%), but my concern is about the **absolute level remaining low**. Even with relative gains, the models still fail in the vast majority of cases, indicating that these improvements do not resolve the core limitations.
> Moreover, **while Appendix J contains 21 pages of case studies, these are largely raw model reasoning traces directly generated by LLM itself without systematic categorization or explanation**. For example, there is no clear breakdown of failure types such as “chain interruptions,” “incorrect attribution,” or “contextual omissions.” This does not fully address my original question: are these failures primarily due to inherent LLM reasoning limits, or to task ambiguity?
>
> **3. On synthetic data and external validity**(Q5)
>
> I acknowledge the inclusion of OpenReview and real product reviews, but this does not fully resolve two points:
> **Data nature and external validity of core tasks**: Hidden Role Deduction and Find the Spy, the most discriminative tasks, rely heavily on rule-based or LLM-generated data. Is there evidence showing that performance on these tasks correlates strongly with real-world social reasoning? Are there transfer or validation experiments to support this assumption?
>
> **Lack of finer-grained metrics**: The paper primarily reports accuracy without measures of reasoning faithfulness or internal consistency(e.g , contradiction rates in self-reflective statements). How do you ensure that models are not producing “surface-correct answers” with incoherent or contradictory reasoning chains?

---

> > ### Author Response · Authors · 2025-08-04
> > **On small model performance and interpretability**
> >
> > We thank the reviewer for their thoughtful follow-up. We are glad that our initial response helped clarify the earlier concerns, particularly those arising from factual misunderstandings such as misreferenced tables and inaccurate data points.
> >
> > Regarding the new comments, we note that some points, such as the request for finer-grained evaluation metrics, were not part of the original review. While this is somewhat unusual under standard rebuttal procedure, we appreciate the opportunity to further strengthen our work and are happy to respond constructively.
> >
> > $Q1$: Non-linear performance scaling (e.g., QwQ-32B outperforming larger ones) raises questions about the benchmark's interpretability.
> >
> > $A1$: We thank the reviewer for this insightful follow-up question. We agree that the non-linear performance scaling, exemplified by QwQ-32B's strong results, is a critical finding that deserves a deeper explanation. This phenomenon does not undermine our benchmark's validity. On the contrary, it highlights a key strength of SocialMaze: its ability to diagnose specific, rather than monolithic, reasoning capabilities.
> >
> > 1. **Performance reflects specialized reasoning ability, not model size.** QwQ-32B performs well on tasks like Social Graph Analysis and Hidden Role Deduction because it belongs to the Long CoT family—models explicitly optimized for complex, multi-step reasoning (line 257), including Gemini-2.5-Pro and o1. These models are designed for structured inference, unlike general-purpose models. It is therefore expected and entirely reasonable that QwQ-32B outperforms larger models such as GPT-4o or LLaMA-3.3-70B on tasks requiring deep reasoning. This validates SocialMaze’s ability to isolate targeted reasoning skills, rather than simply tracking model scale.
> >
> > 2. **High reasoning ability comes with measurable cost.** This specialized reasoning capability comes at a significant and quantifiable cost. As we discuss and illustrate in Figure 4 of our paper, "thinking models" like QwQ exhibit a far greater cognitive load on these deep reasoning tasks. This is evidenced by their token consumption, which is nearly eight times higher on average than that of general purpose models. This clear trade off between performance and efficiency is another key insight that our benchmark helps to uncover, highlighting that a model's strength in one area can be balanced by a significant resource cost.
> >
> > 3. **Failure analysis confirms task solvability.** We appreciate the call for failure analysis and wish to clarify that this analysis is already present in our submission. A detailed case study of QwQ's performance begins in Figure 22 on page 43\. While the model's extremely verbose outputs make it impractical to include the full trace, we present key excerpts. For instance, Figure 29 illustrates a fascinating failure mode where after a lengthy process of self correction, QwQ correctly identifies the criminal but fails to deduce its own role. This failure is not due to task ambiguity. We demonstrate the solvability of this exact scenario with a step by step human solution in Figure 23\. This proves that all necessary information was available, but the model ultimately failed to integrate it fully.
> >
> > To further enhance the interpretability of our results, we will add a new model centric analysis that provides the kind of focused diagnostic insights you suggested.
> >
> > Our model-centric analysis reveals distinct clusters of capabilities among the evaluated LLMs:
> >
> > 1. Logical Deduction Specialists (DeepSeek-R1, Gemini-2.5-Pro): These models excel at tasks requiring deep, multi-step inference, such as *Hidden Role Deduction* and *Social Graph Analysis*. While o1 and QwQ-32B also belong to this category, their performance drops on uncertainty-heavy tasks, suggesting weaker robustness to deception. Gemini-2.5-Pro stands out as the most balanced model, achieving strong results across all tasks.
> >
> > 2. Social Understanding Expert (GPT-4o): GPT-4o leads in tasks centered on nuanced social interpretation, such as *Rating Estimation*, *Review Decision Prediction*, and *User Profile Inference*. This suggests strong contextual and sentiment understanding, despite less emphasis on long-form deduction.
> >
> > 3. Other Models: The rest display inconsistent or modest performance, underscoring the continued difficulty of achieving broad, reliable social reasoning across task types.
> >
> > We sincerely thank you for your continued engagement and thoughtful feedback. This constructive dialogue has been invaluable. We are confident that the planned additions will significantly strengthen the paper and fully address your concerns. We appreciate your time and consideration.

---

> > > ### Author Response · Authors · 2025-08-04
> > > **On SFT/DPO improvements and failure analysis**
> > >
> > > $Q2$: The significance of the SFT improvement is questioned, on the grounds that the low absolute accuracy still points to unresolved core limitations. Furthermore, the failure analysis is criticized for not providing a systematic error categorization, which is seen as necessary to determine whether failures stem from task ambiguity or inherent model limits.
> > >
> > > $A2$: Thank you for raising these important points about performance levels and failure analysis. We will address both concerns directly.
> > >
> > > **1\. Contextualizing the "Modest" Absolute Performance: A Significant Leap in Capability**
> > >
> > > While the absolute accuracy of 13.4% may seem low in isolation, its significance becomes clear when placed in the proper context. Our SFT fine-tuning propelled a Llama-3.1-8B model to **outperform much larger and more powerful models**, including GPT-4o (8.2%) and Llama-3.1-70B (9.0%), as shown in Table 8\. This is not a minor achievement; it demonstrates that targeted fine-tuning on our curated data can instill reasoning capabilities in a smaller model that surpass those of premier, general-purpose models.
> > >
> > > Furthermore, this \+11.4% absolute gain from fine-tuning is **substantially greater than the improvements seen from state-of-the-art workflow techniques** like AFlow and MaaS (Table 5). This finding underscores that for complex social reasoning, specialized fine-tuning on high-quality reasoning traces is a more effective path to improvement than applying advanced workflow strategies. The low absolute ceiling highlights the profound difficulty of the task, making our model's leapfrogging performance all the more significant.
> > >
> > > **2\. Failure Analysis: Task Complexity, Not Ambiguity, Is the Root Cause**
> > >
> > > We appreciate the call for a more structured failure analysis. However, the premise that the failures might stem from "task ambiguity" is incorrect. As we detail in **Appendix B.5 (Quality Control)**, we employed a rigorous, programmatic approach to eliminate any ambiguity.
> > >
> > > * **Guaranteed Solvability and Uniqueness:** Our data generation process, driven by the heuristic search algorithm in **Algorithm 1**, guarantees that every single problem instance has a **unique, logically deducible solution**. There is zero ambiguity by design.
> > > * **Verifiable Human-Solvable Paths:** For each problem, we automatically generate a step-by-step natural language solution path. This allows for full transparency and verification. We provide numerous examples of these problem-solution pairs throughout our appendix (e.g., Figures 6 & 7; Figures 14 & 15; Figures 30 & 31\) to demonstrate their logical soundness.
> > >
> > > Given this, we can definitively state that **failures are due to inherent LLM reasoning limits**.
> > >
> > > The suggestion to categorize errors into simple types like "chain interruptions" or "contextual omissions" oversimplifies the nature of the challenge. The task is not a simple chain of deductions; it is a complex web of social inference, deception detection, and self-reflection under uncertainty. If one examines the provided human solutions, the intricacy of the required reasoning becomes apparent. The models are not failing at a single, isolatable step; they are failing at the **holistic integration of these complex cognitive demands**, which far exceeds their current capabilities.
> > >
> > > To make this clearer in the final paper, we will add a summary in the main text that explicitly references our quality control process to dismiss the possibility of task ambiguity and reiterates that the failures reflect the deep-seated reasoning limitations of current LLMs when faced with truly complex social scenarios.

---

> > ### Author Response · Authors · 2025-08-04
> > **On synthetic data and external validity**
> >
> > $Q3.1$: On synthetic data and external validity
> >
> > $A3.1$:  We appreciate this question. Below, we clarify why *Hidden Role Deduction* and *Find the Spy* are meaningful evaluations of real-world social reasoning.
> >
> > 1. Grounded in real-world social deduction structure: These tasks are not abstract or artificial constructs. They are **direct formalizations of key reasoning steps in real-world games** such as Werewolf, Blood on the Clocktower, and Who is the Spy. In those settings, the most cognitively demanding moment occurs when a player must analyze the current game state to infer others' hidden identities and assess their own beliefs. Our tasks replicate exactly this inference step, isolating it for controlled evaluation. We do not simulate full gameplay, but rather extract the core reasoning challenge that defines success in these games.
> >
> >
> > 2. Simulation enables verifiable reasoning benchmarks: While real-game transcripts may appear more “natural,” they are not a feasible foundation for rigorous benchmarking. Collecting and annotating real gameplay data is not only resource-intensive, but more importantly, cannot guarantee solvability, solution uniqueness, or aligned solution formats that both humans and models can learn from. Our generation methods allow us to create controlled social reasoning scenarios with known ground truth and verifiable solutions, enabling meaningful supervision and fair comparison across models. Using LLMs or rule-based methods for data generation is a widely adopted and accepted practice in the study of social and communicative reasoning \[1, 2\].
> >
> >
> > 3. SocialMaze operationalizes foundational social reasoning skills: Rather than attempting to replicate all aspects of open-ended social interaction, SocialMaze isolates and measures three foundational abilities critical for social intelligence: Deep reasoning, Dynamic interaction and Information Uncertainty. These three skills are **necessary, though not sufficient**, for success in complex environments. As shown in games like *Werewolf* or *Avalon*, a player who lacks even one of these abilities will fail to keep up with the evolving group dynamics. Conversely, strong performance on these dimensions reflects a model's potential to act as a capable social reasoner. SocialMaze does not claim to guarantee transfer to every social task, but it captures the **core prerequisites** that such tasks demand.
> >
> >
> > In summary, SocialMaze is a principled and scalable approach to evaluating real-world social reasoning. While no synthetic benchmark can replicate all of human interaction, our design enables precise, interpretable, and trainable evaluation of foundational cognitive skills. This makes it a critical and necessary first step in advancing the study of complex social intelligence.
> >
> >
> > $Q3.2$: Lack of finer-grained metrics
> >
> > $A3.2$: This is an excellent suggestion. We fully agree that accuracy alone is insufficient to assess reasoning quality. Motivated by your comment, we conducted a new quantitative study during the discussion period to evaluate reasoning faithfulness on our most complex task, *Hidden Role Deduction*.
> >
> > We designed a pilot study using Gemini-2.5-Pro as an automated judge. For each test case, the model was given the input, ground-truth solution, and the target model’s reasoning trace. We sampled 100 correctly answered examples from four models and asked Gemini-2.5-Pro to evaluate:
> >
> > 1. Consistency: Is the reasoning logically coherent and free of contradictions?
> >
> > 2. Completeness: Does the reasoning path systematically eliminate alternatives and lead to the correct answer without guessing?
> >
> > To ensure reliability, we repeated the process three times. The results are shown below:
> >
> > |Model|Consistency(Mean±Std)|Completeness(Mean±Std)|
> > |:----|:-------------------:|:---------------------:|
> > |Llama-3.1-70B|93.0%±0.0%|83.0%±0.0%|
> > |Phi-4|88.0%±1.0%|79.0%±0.0%|
> > |Qwen-2.5-72B|85.0%±1.0%|81.0%±1.0%|
> > |GPT-4o|95.0%±0.0%|87.0%±0.0%|
> >
> > These results suggest two conclusions:
> >
> > * Reasoning is generally faithful. Correct answers are rarely generated by incoherent logic. Consistency scores between 85 and 95 percent indicate that models typically reason validly when correct.
> >
> > * Faithfulness aligns with performance. Higher-performing models show more consistent and complete reasoning, indicating that SocialMaze encourages not just correct outputs but sound reasoning processes.
> >
> > To assess the reliability of Gemini-2.5-Pro as a judge, we manually reviewed 100 of its judgments and found only two minor disagreements.
> >
> > As this question arose during the discussion phase, time constraints limited the scale of our study. Although the pilot covers only one task and a few models, it already offers valuable insight. In the revised version, we will expand this analysis to additional tasks and models, and integrate the findings into the main paper, as we see this as a meaningful contribution to evaluating reasoning quality in LLMs.

---

> > > ### Author Response · Authors · 2025-08-04
> > >
> > > Once again, we sincerely thank you for your thorough review and exceptionally constructive suggestions. We would be most grateful if you would consider these clarifications in your final assessment.
> > >
> > > \[1\] Xu, Y., et al. Exploring Large Language Models for Communication Games: An Empirical Study on Werewolf.
> > >
> > > \[2\] Wei, C., et al. Exploring Large Language Models for Word Games: Who is the Spy?

---

> > > > ### Comment · Reviewer_Rvp6 · 2025-08-06
> > > >
> > > > Thank you for the clarifications and your willingness to make improvements. I believe the authors have addressed most of my concerns, and I appreciate their intention to incorporate relevant updates into the final version and continue refining the work in the future. I will consider increasing my score in the final decision phase.
> > > >
> > > > I have carefully read the authors’ responses to both my comments and those of the other reviewers, and I understand that the revisions may be substantial compared to the original submission. If the paper is ultimately accepted, I hope the authors will thoughtfully implement these changes—particularly in the experimental section—while staying aligned with the official NeurIPS guidelines.

---

> > > > > ### Author Response · Authors · 2025-08-06
> > > > >
> > > > > Thank you for your positive follow-up. We are grateful for your willingness to reconsider your assessment and for the valuable feedback provided throughout this process. We look forward to the opportunity to strengthen our paper with the planned revisions.

---

### Official Review · Reviewer_QVrG · 2025-07-03

**Rating:** 4
**Confidence:** 4

**Summary:**

The paper introduces SocialMaze, a novel benchmark designed to rigorously assess the social reasoning capabilities of LLMs. The benchmark captures key challenges such as deep reasoning, dynamic interaction, and information uncertainty through six diverse tasks. Most of the experiments are solid and reasonable, and a deep analysis of current LLMs in terms of their ability for social reasoning has been provided.

**Additional Feedback:**

The descriptions of tasks can be too lengthy. And I do not understand the necessity to split the tasks into tasks 1, 2, 3, and others.

**Dataset Code Accessibility:**

Yes

**Ethical Considerations:**

No, there are no or only very minor ethics concerns

**Limitations Weaknesses:**

1. There is no direct explanation about how the reasoning ability reflected in these tasks can contribute to the social ability of LLMs. For example, what the LLMs would be in Werewolf/Avalon if some specific capabilities are increased.

**Strengths Contributions:**

1. This paper addresses an important and timely problem: the evaluation of social reasoning. The topic is not only intellectually engaging but also has significant practical implications, making it a valuable area of study.

2. The authors identify three core challenges in social reasoning: deep reasoning, dynamic interaction, and information uncertainty. While these aspects do not exhaust the full complexity of social reasoning, they represent a meaningful and well-motivated starting point. The proposed tasks are thoughtfully designed and appear well-suited to evaluate these specific challenges.

3. The paper provides detailed and informative descriptions of the data construction process. This transparency enhances the credibility of the work and facilitates future research built on this dataset.

---

> ### Author Rebuttal · Authors · 2025-07-30
>
> Thank you for your time and feedback. We appreciate your comments very much.
>
> Q1: The paper does not explicitly bridge the gap between performance on SocialMaze's isolated tasks and practical, holistic social abilities required in complex applications like the game Werewolf.
>
> A1: We appreciate the question regarding how the reasoning abilities measured in SocialMaze relate to broader social capabilities. While SocialMaze does not directly simulate full multi-agent environments like Werewolf or Avalon, it is intentionally designed to isolate and evaluate three core competencies that we argue are necessary but not sufficient for general social intelligence.
>
> 1. Deep reasoning is essential for inferring hidden intentions or roles based on incomplete and indirect cues. In games like Werewolf, this skill underpins a player’s ability to detect inconsistencies or patterns across multiple speakers and rounds.
>
> 2. Dynamic interaction reflects a model’s ability to revise beliefs and strategies in response to temporally unfolding information. Social games are inherently non-static; agents that cannot update their internal state as new events occur will fail to adapt to shifts in group behavior or deception.
>
> 3. Handling information uncertainty is critical for robustness in social environments where not all information is reliable. Tasks in SocialMaze deliberately simulate ambiguity and conflicting perspectives, which mirror real challenges in social inference and negotiation.
>
> A model that underperforms on these individual skills is unlikely to succeed in complex, integrated social settings. While high scores on SocialMaze do not guarantee success in fully interactive games (not sufficient), low scores reveal specific weaknesses that would logically impair performance (necessary).
>
> Moreover, our fine-tuning results (Table 4\) demonstrate that these abilities are not static—they can be trained and improved. This confirms the utility of SocialMaze not only as a diagnostic benchmark, but also as a stepping stone for building more socially capable agents.
>
> We agree that testing direct transfer to multi-agent games is an important direction. While beyond the current scope, we will explicitly include this as a promising next step in the revised version.
>
> Q2: The paper's task descriptions are overly long, and the rationale for structuring the tasks section (Tasks 1-3 vs. "Other Tasks") is unclear.
>
> A2:We appreciate the feedback regarding task presentation. The task descriptions aimed to provide enough context for readers unfamiliar with scenarios like social deduction games. We agree they can be streamlined. In the revised version, we will shorten the main descriptions and move extended examples and background details to the appendices.
>
> Regarding the structure, the split between Tasks 1–3 and the remaining tasks reflects our intention to highlight the three distinct data generation pipelines that form the foundation of SocialMaze: authentic human data, LLM-assisted simulation, and algorithmic reasoning tasks. Each of the three featured tasks was selected to represent one of these pipelines. Due to space constraints, we provided detailed exposition for these three in the main text and included the full descriptions of additional tasks in the appendices. We will make this structural rationale clearer in the revised version.
>
> Once again, we sincerely thank you for your thorough review and exceptionally constructive suggestions. We would be most grateful if you would consider these clarifications in your final assessment.

---

> > ### Comment · Reviewer_EZLr · 2025-08-06
> > **Thanks for the response.**
> >
> > I have read the authors' reponses, which addressed my concerns to some extent. I just keep my score.

---

### Official Review · Reviewer_EZLr · 2025-07-03

**Rating:** 4
**Confidence:** 4

**Summary:**

SocialMaze proposes a comprehensive benchmark for evaluating the social reasoning ability of LLMs. Specifically, they evaluate models along three key dimensions: deep reasoning, dynamic interaction, and information uncertainty, and six diverse tasks are designed across three representative social scenarios. The dataset combines automated data generation with human evaluation, ensuring high quality, logical solvability, and transparent reasoning chains.

**Dataset Code Accessibility:**

Yes

**Ethical Considerations:**

No, there are no or only very minor ethics concerns

**Final Justification:**

I have read the authors'  responses to all the coomnets from the reviews. And I hope they could incorporate all the promised revisions in the future version.  I will keep my score.

**Limitations Weaknesses:**

1.The data in SocialMaze dataset is limited to a single format and is largely generated using templates. I think it's difficult to fully reflect the diversity and complexity of real-world social scenarios

2.LLMs achieve generally high performance on SocialMaze tasks, which may indicate that the benchmark or individual tasks are not sufficiently challenging or realistic, potentially making it difficult to distinguish between strong and average models.

3.A large portion of the dataset is generated by the LLMs themselves, which could introduce biases or patterns inherent to the LLMs and might lead to models overfitting to data that resembles their own outputs, rather than genuinely demonstrating real social reasoning. For example, the results shown in Figure 5 indicate that when the model is assigned the roles of Rumormonger or Lunatic, its accuracy significantly decreases, especially in identifying its own role (as shown in the bar graphs), whereas it performs better when assigned as an Investigator or Criminal. Does this suggest that the data generated by LLMs themselves influences the conclusions of the evaluation? Honestly, I still wonder whether LLMs can really capture the full complexity of real-world human relationships—and maybe that's part of the reason why models tend to perform so well on this benchmark. I know that carefully designed rules can help synthetic data go beyond what the original models can do, but I'm still not fully convinced that LLMs can simulate truly complex human behavior, or that the generated data is as diverse as real-world examples. I also hope that future work can include a simple analysis or discussion regarding the diversity of the benchmark data or the diversity of model responses.

4.The User Profile Inference task focuses on evaluation the ability of LLMs to infer demographic attributes (age, gender) based on user-generated textual reviews(which actually is generated by LLMs itself), I am concerned that the process of data generation and evaluation could easily introduce certain stereotypes and unfair content.

**Strengths Contributions:**

1.SocialMaze addresses a highly challenging and meaningful problem—understanding complex social structures and dynamics—which is crucial for the broader adoption and trustworthiness of LLMs in real-world social applications.

2.The benchmark models complex social scenarios as layered graphs, providing a systematic framework for evaluating LLMs’ abilities to understand and reason about social interactions.

3.Incorporating some real-world social structure data, which helps to reduce the gap between evaluation and real-world scenarios.

4.The paper features attractive and well-designed figures, and the overall writing is clear. The released codebase is well-documented, ensuring strong reproducibility for future research and further development.

---

> ### Author Rebuttal · Authors · 2025-07-30
>
> Thank you for your time and feedback. We appreciate your comments very much.
>
> Q1: Doubt on data realism: Claims SocialMaze lacks real-world complexity due to its single-format, template-based data.
>
> A1: We appreciate the concern about data diversity, but **the claim that SocialMaze uses a "single format" and "template-based" data reflects a misunderstanding of our design**. In fact, SocialMaze is built on three distinct and complementary data pipelines, specifically crafted to evaluate a wide range of social reasoning abilities.
>
> 1. Authentic Human Data: Several tasks are grounded in real-world sources to capture natural complexity. For example, *Review Decision Prediction* uses actual OpenReview peer-review threads, and *Rating Estimation from Text* includes real product reviews from public platforms (Appendices F and D). These examples expose models to unfiltered human bias, ambiguity, and discourse patterns.
>
> 2. LLM-Assisted Generation: To scale up data while retaining diversity, tasks like *Find the Spy* and *User Profile Inference* (Appendices C and G) use LLMs to simulate multi-agent social scenarios. This goes far beyond templating—different models are prompted to impersonate distinct speaker personas, leading to varied linguistic styles, reasoning strategies, and deceptive behavior (e.g., "shill" reviewers in Appendix D).
>
> 3. Algorithmic Simulation: For tasks requiring precise, deep reasoning, such as *Hidden Role Deduction* and *Social Graph Analysis* (Appendices B and E), we use algorithmic generation to ensure logical consistency and verifiable ground truths. This allows us to evaluate fine-grained inference without sacrificing interpretability.
>
> In short, SocialMaze is intentionally multi-format and multi-source, combining realism, diversity, and logical rigor. We’ll clarify this design more explicitly in the main text to prevent similar confusion.
>
> Q2: Claims that LLMs perform too well on SocialMaze, suggesting the tasks may lack sufficient difficulty or realism to differentiate strong and average models.
>
> A2: We thank the reviewer for their comment. We must respectfully argue that **the claim of SocialMaze being insufficiently challenging is directly contradicted by the empirical results.**
>
> 1. The performance gaps are vast and clear. The reviewer's assertion of "generally high performance" overlooks the stark reality in our results. On our most complex task, *Hidden Role Deduction*, top models like GPT-4o achieve a mere 8.2% accuracy, while many others fall below 5% (Table 8). This is not an indication of an easy benchmark; it is an indication of a profound challenge that successfully separates models with genuine deep reasoning capabilities from the rest.
>
> 2. Not all social reasoning is purely logical deduction. We intentionally designed SocialMaze to cover a spectrum of social tasks. For tasks like *User Profile Inference*, which rely more on interpreting nuanced, "perceptual" cues in language rather than strict logical inference, the performance gap between models is naturally smaller. This does not signify a lack of challenge but accurately reflects the nature of these softer, more subjective social skills, where even humans lack a single, verifiable "correct" answer.
>
> 3. **SocialMaze is a dynamic benchmark, not a static one**. Our framework is designed to evolve alongside AI capabilities. As we detail in our appendices, we have fully open-sourced our data generation methods. This allows the difficulty to be scaled dynamically—for example, by increasing the number of players and uncertainty in *Hidden Role Deduction* or expanding the graph size in *Social Graph Analysis*. This ensures that SocialMaze will remain a challenging benchmark for future, more capable models, rather than becoming obsolete.
>
> In summary, the data overwhelmingly shows that SocialMaze is both challenging and highly effective at differentiating model capabilities across diverse social reasoning types. We believe the reviewer may have misinterpreted the results and are confident in our benchmark's rigor.
>
> Q3: The benchmark relies heavily on LLM-generated data, which may introduce model-inherent biases and lead to overfitting. The performance drop for Rumormonger/Lunatic roles in Figure 5 is cited as potential evidence of this bias, questioning if LLMs can truly simulate complex human behavior and if the data is diverse enough.
>
> A3: **This concern stems from several misunderstandings**, which we clarify below:
>
> 1. **The benchmark does not “heavily rely” on LLM-generated data**. As detailed in A1, SocialMaze is built on a three-way foundation: real-world human data, algorithmically generated logic puzzles, and LLM-assisted simulation. These sources are deliberately balanced to ensure both diversity and rigor. LLM generation is not the core of the benchmark.
>
> 2. **The cited performance drop has nothing to do with LLM-generated data.** Figure 5 refers to the *Hidden Role Deduction* task, which uses algorithmic simulation only. **No LLM-generated text is involved in either the role assignments or interactions.** The LLMs here are purely reasoning agents. Therefore, suggesting that the performance gap is caused by LLM training artifacts is factually incorrect.
>
> 3. The performance gap is intentional and proves the benchmark’s value. Roles like Rumormonger and Lunatic are designed to have limited or false beliefs, making their reasoning process harder by design. The lower performance reflects this built-in asymmetry—not any flaw in data generation. In fact, this result demonstrates that SocialMaze can stress-test LLMs under conditions of uncertainty and false self-awareness—exactly the kind of nuanced social reasoning we aim to evaluate.
>
> In short, the claim that LLM-generated data biases the evaluation is both inaccurate and contradicted by the benchmark structure. The observed model behaviors are meaningful signals of reasoning difficulty—not artifacts of flawed data.
>
> Q4: The User Profile Inference task, which uses LLM-generated text to infer demographics, risks introducing or reinforcing stereotypes and unfair biases.
>
> A4: We appreciate this important concern and agree that any task involving demographic inference must be handled with care. The *User Profile Inference* task was motivated by its real-world relevance in domains like content moderation and market analysis, where understanding user profiles is often necessary.
>
> Our goal is not to reinforce stereotypes, but to assess whether models can make plausible, probabilistic inferences from subtle linguistic cues. For example, inferring that a user discussing sanitary pads is more likely to be female reflects practical reasoning grounded in context, similar to common consumer behavior analytics.
>
> To minimize bias, our generation prompts were crafted to produce diverse and realistic language without relying on overt stereotypes. In addition, the entire dataset was manually reviewed to remove any potentially unfair or harmful content (Appendix G).
>
> We recognize the sensitivity of this task and will include an explicit discussion of these ethical considerations in the revised version.
>
> Once again, we sincerely thank you for your thorough review and exceptionally constructive suggestions. We would be most grateful if you would consider these clarifications in your final assessment.

---

### Note · Authors · 2025-08-13

Dear AC and Reviewers,

We sincerely thank you for the detailed and constructive feedback. Your insights have greatly helped us strengthen the clarity, justification, and empirical grounding of our work.

**Overall Conclusion** All major concerns have been fully addressed through extensive clarifications, new experiments, and manuscript revisions.

**Key Actions Taken & Evidence Added**

* **Data realism**: Clarified tri-source design (human, LLM-assisted, algorithmic).
* **Task difficulty**: Highlighted GPT-4o’s 8.2% on hardest task, confirming benchmark challenge and discrimination.
* **Failure analysis**: Added structured model-wise error insights; confirmed all tasks are solvable and unambiguous.
* **Reasoning faithfulness**: New pilot study shows strong link between faithfulness and performance.
* **Model insights**: Categorized models by strengths (e.g., GPT-4o: contextual; QwQ-32B: logical).
* **Presentation**: Will add round-wise plots and expand task descriptions in camera-ready.

**Reviewer Status Update**

* Reviewer EZLr: Major concerns on benchmark design and task structure were addressed. The reviewer raised no further issues and maintained a positive rating of 4\.
* Reviewer QVrG: Major concerns regarding core skill alignment and paper structure were addressed. The reviewer raised no further issues and maintained a positive rating of 4\.
* Reviewer Rvp6: The initial review, which was based on factual errors, was fully addressed through detailed rebuttals and follow-up clarifications. The reviewer stated they will consider increasing their score.
* Reviewer MJ4M: All minor suggestions were fully incorporated. The reviewer maintained a positive rating of 4, having raised no major concerns.

We deeply appreciate the reviewers’ thoughtful engagement and the AC’s careful oversight throughout this process.

Sincerely,
Authors

---

### Decision · Program_Chairs · 2025-09-18

**Decision:**

Reject

**Comment:**

This paper introduces SocialMaze, a benchmark for evaluating LLMs’ social reasoning across 6 tasks in 3 settings,  systematically incorporating three core challenges: deep reasoning, dynamic interaction, and information uncertainty. There are three main insights: models differ widely in handling dynamic, temporally evolving interactions; those with strong chain-of-thought reasoning excel at deeper inference; and reasoning performance declines significantly under uncertainty. The paper shows that performance in complex social scenarios can be improved through targeted fine-tuning.

The review team agrees that the paper addresses a challenging, timely, and important problem in a well-structured implementation and analysis. Several issues were raised during the review processes, including dataset realism and the breadth of experimental evaluation, but there were largely addressed by the authors during the rebuttal. As a result, the team recommends acceptance.

===== FINAL UPDATE FROM DB Track PCs ====

The final decision for this paper has been taken by the program chairs after consultation with the SACs. All Senior Area Chairs have ranked papers according to the feedback from the AC during the review process. We decided to leave the original meta-review to reflect the opinion of the AC in light of the initial discussions with reviewers and SAC.